# Non-Asymptotic Length Generalization

**Thomas Chen** [1]   **Tengyu Ma** [1]   **Zhiyuan Li** [2]

## Abstract

Length generalization is the ability of a learning algorithm to learn a hypothesis which generalizes to longer inputs than the inputs in the training set. In this paper, we provide provable guarantees of length generalization for various classes of functions in an idealized setting. First, we formalize the framework of non-asymptotic length generalization, which requires a computable upper bound for the minimum input length that guarantees length generalization, as a function of the complexity of ground-truth function under some given complexity measure. We refer to this minimum input length to length generalize as length complexity. We show the Minimum-Complexity Interpolator learning algorithm achieves optimal length complexity. We further show that whether a function class admits non-asymptotic length generalization is equivalent to the decidability of its language equivalence problem, which implies that there is no computable upper bound for the length complexity of Context-Free Grammars. On the positive side, we show that the length complexity of Deterministic Finite Automata is $2n - 2$ where $n$ is the number of states of the ground-truth automaton. Our main results are upper bounds of length complexity for a subset of a transformer-related function class called C-RASP (Yang & Chiang, 2024). We show that the length complexity of 1-layer C-RASP functions is $O(T^2)$ when the ground-truth function has precision $T$, and that the length complexity of 2-layer C-RASP functions is $O(T^{O(K)})$ when the ground-truth function has precision $T$ and $K$ heads.

## 1. Introduction

The generalization of a trained model from shorter inputs seen during training time to longer inputs seen at inference time is a phenomenon called length generalization. The question of when length generalization is possible is an important question in the area of language modeling and reasoning with Large Language Models (LLMs) (Brown et al., 2020). Real-world language understanding often requires handling longer contexts that exceed training-time input lengths, such as long Chain-of-Thought for reasoning problems, multi-turn conversations, lengthy documents, or complex code. Many factors limit the ability to train on long sequences directly, such as the increasing computational cost and memory requirement of training on longer sequences and the fact that long sequences are rare. Successful length generalization does not only allow efficient training for good long-context performance, but also is a natural test on whether the model has robustly learned the underlying language or reasoning patterns.

Prior works study length generalization in a controlled setting where Transformers (Vaswani et al., 2017) are trained on algorithmic tasks from scratch. In this context, length generalization has been empirically observed for certain algorithmic tasks but not others, and it appears that transformers cannot length generalize for most tasks. Tasks that have been tested are arithmetic tasks like integer addition (Nogueira et al., 2021; Nye et al., 2021; Anil et al., 2022; Zhou et al., 2024), formal-language tasks like PARITY (Anil et al., 2022), algorithmic tasks in the Chomsky Hierarchy (Shaw et al., 2021; Delétang et al., 2023; Ruoss et al., 2023), copying tasks, tasks involving a sequence of integers such as sorting or finding the mode (Zhou et al., 2023), and deducing the end-assignment of a variable in a block of code (Anil et al., 2022). Some of these tasks do not exhibit length generalization in their vanilla form, but do exhibit length generalization if certain modifications are made to the learning setup. These modifications include modifying the input and output format (Zhou et al., 2023; 2024), adding positional embeddings (Jelassi et al., 2023), and adding access to a scratchpad (Nye et al., 2021).

On the theoretical side, (Gold, 1967) provides a foundational result in the theory of length generalization, which is originally stated in a more broad context of language iden-

[1]Department of Computer Science, Stanford University, Stanford, USA [2]Toyota Technological Institute at Chicago, Chicago, USA. Correspondence to: Thomas Chen <tchen@cs.stanford.edu>.

*Proceedings of the 42nd International Conference on Machine Learning*, Vancouver, Canada. PMLR 267, 2025. Copyright 2025 by the author(s).

| Function Class | Complexity of Ground-Truth Function | Length of Training Data Sufficient to Generalize |
|---|---|---|
| DFAs | number of states, $c$ | $2c - 2$ (Proposition 2.7) |
| CFGs | description length, $c$ | no computable bound in $c$ exists (Proposition 3.5) |
| C-RASP[1] | precision, $T$ | $O(T^2)$ (Theorem 5.5) |
| C-RASP[2] | precision, $T$, and number of heads, $K$ | $O(T^{O(K)})$ (Theorem 5.6) |

*Table 1.* Summary of results: upper bounds on minimum length of binary strings in training data which suffices for length generalization.

tification. Let $\Sigma$ be a finite alphabet, let a hypothesis be a mapping from all strings $\Sigma^*$ to $\{0, 1\}$, and let $\mathcal{F}$ be the hypothesis class. Gold (1967) roughly says that there is a learning algorithm $\mathcal{A}$ such that for all hypotheses in $\mathcal{F}$, $\mathcal{A}$ will eventually learn the ground-truth hypothesis, when the training set inputted to $\mathcal{A}$ contains (string, label) pairs for all the strings up to a sufficiently large length. Gold's result holds for all hypothesis classes $\mathcal{F}$ satisfying the mild assumption that $\mathcal{F}$ can be enumerated by a Turing machine, including Regular languages, Context-Free languages, decision problems that can be solved by finite-precision transformers with or without Chain-of-Thought, etc. The learning algorithm $\mathcal{A}$ simply returns the first hypothesis in the enumeration of $\mathcal{F}$ that correctly labels all training examples. When this enumeration lists the functions in $\mathcal{F}$ in order of increasing complexity for some complexity measure, then we call this learning algorithm the Minimum-Complexity Interpolator, denoted by $\mathcal{A}_{\mathsf{mci}}$.

The gap between the wide range of function classes which Gold's theory predicts length generalization for and the limited classes of functions that transformers empirically length generalize on presents an opportunity to develop a more predictive theory of length generalization. There are many potential reasons for this gap, such as the discrepancy between the learning algorithm $\mathcal{A}_{\mathsf{mci}}$ and the gradient-based learning algorithms used in practice. However, a more fundamental issue of Gold's result is that the guarantee is inherently *asymptotic* — it does not provide any information on the minimum input length required to achieve length generalization. It is completely possible that even if gradient-based training methods like SGD are able to find the minimum-complexity interpolating hypothesis, length generalization still only happens when the training set contains impractically long length inputs. In this case, it may appear empirically that length generalization does not occur at all, despite the asymptotic theoretical guarantee.

Intuitively, the minimum input length required to achieve length generalization should be a function of the complexity of the ground truth hypothesis, and better length generalization is expected for simpler hypotheses. To both get a more useful length generalization guarantee and better understand the limit of length generalization, the goal of this paper is to answer the following question:

*What is the minimum input length required to*

*achieve length generalization as a function of complexity of ground-truth hypothesis, assuming we have infinite computational resources?*

**Our Contributions:** This paper provides a more fine-grained analysis of length generalization, by providing a *non-asymptotic* guarantee on the minimum input length required to achieve length generalization, as a function of complexity of ground-truth. More specifically, we make the following contributions:

- In Section 2, we introduce the framework of *non-asymptotic length generalization* (Definition 2.6), which requires a non-asymptotic upper bound for the minimum input length that guarantees length generalization, given the complexity of ground-truth function under some given complexity measure $\mathcal{C}$. The latter we call the length complexity under $\mathcal{C}$ (Definition 3.1). We show that the Minimum-Complexity Interpolator learning algorithm $\mathcal{A}_{\mathsf{mci}}$, instantiated with complexity measure $\mathcal{C}$, is the optimal with respect to the length complexity under $\mathcal{C}$. As a concrete example, we show that $\mathcal{A}_{\mathsf{mci}}$ only needs inputs up to length $2c-2$ to learn any ground-truth Deterministic Finite Automata (DFA) of $c$ states.

- In Section 3, we show whether a hypothesis class admits non-asymptotic length generalization is equivalent to whether the problem of deciding equivalence of two finite descriptions of hypotheses is decidable. As a consequence, Context-Free Grammars (CFGs) do not admit non-asymptotic length-generalization, though they admit (asymptotic) length-generalization in the limit.

- In Section 5, we prove non-asymptotic length generalization for a subset of the transformer-related hypothesis class, C-RASP (Yang & Chiang, 2024). C-RASP is a superset of functions expressible by transformers. Variants of the RASP function class, RASP-L and C-RASP, have been shown to have a good predictive power on the length generalization performance of transformers (Zhou et al., 2023; Huang et al., 2024). We study two subclasses of C-RASP: C-RASP[1] (Theorem 5.5) and C-RASP[2] (Theorem 5.6), which are of depth 1 and 2, respectively.

## 2. Non-Asymptotic Length Generalization

**Notation.** Fix the alphabet $\Sigma = \{0, 1\}$ throughout the paper. Define computable (recursive) functions as functions which can be computed by a Turing Machine (TM),

which halts on all inputs. A function $f : \mathbb{N} \to \mathbb{N}$ is *computably bounded* if there exists a computable function $g : \mathbb{N} \to \mathbb{N}$ such that $f(n) \leq g(n)$ for all $n \in \mathbb{N}$. Let $\langle M \rangle \in \{0,1\}^*$ denote the binary string encoding of $M$. Let $[N] := \{1, 2, 3, \ldots, N-1, N\}$. Denote $\{0,1\}^n$ as the set of binary strings of length $n$, $\{0,1\}^{\leq n} := \bigcup_{j=0}^{n} \{0,1\}^j$, and $\{0,1\}^* := \bigcup_{j \geq 0} \{0,1\}^j$. $\mathbb{1}[\cdot]$ denotes the indicator function and $\mathrm{cl}(A)$ as the closure of set $A \subset \mathbb{R}^d$.

**Representation of Functions.** We are interested in learning subsets of computable functions mapping $\{0,1\}^*$ to $\{0,1\}$, denoted by $\mathcal{F}$. Because a learning algorithm can only return a finite description of the hypothesis it selects, we need to define the representation of the hypothesis and the corresponding encoding system below in Definition 2.1.

**Definition 2.1** (Encoding System). An *encoding system* is a TM $\mathcal{R}$ which on input of a finite string $p$ (which can be thought as the code or description of a function), outputs the TM description of the computable function $f : \{0,1\}^* \to \{0,1\}$ represented by $p$. Here we denote the description of the TM as $\langle \mathcal{R}(p) \rangle$ and the function $f$ as $\mathcal{R}(p)$. We use $\mathcal{F}^{\mathcal{R}}$ to denote the function class implicitly defined by the encoding system $\mathcal{R}$, i.e. $\mathcal{F}^{\mathcal{R}} = \{\mathcal{R}(p) : p \in \{0,1\}^*\}$.

Often the standard encoding system is only defined on valid inputs. For convenience we define the encoding system $\mathcal{R}$ for all inputs in $\{0,1\}^*$ and map invalid inputs to the empty language. As examples of encoding systems, DFAs and CFGs are two encoding systems for regular languages and context-free languages, respectively.

**Definition 2.2.** An $n$-state Deterministic Finite Automaton (DFA) is a tuple $M = (Q = [n], \Sigma = \{0,1\}, \delta, q_0, F)$ where $Q = [n]$ is the set of states, $\Sigma = \{0,1\}$ is the input alphabet, $\delta : Q \times \Sigma \to Q$ is the transition function, $q_0 \in Q$ is the start state, and $F \subseteq Q$ is the set of accepting states. The encoding system $\mathcal{R}_{\mathsf{DFA}}$ is a TM that reads $\langle M \rangle$ and outputs the description of a TM that simulates $M$. For any input string $x \in \{0,1\}^*$, the language characterized by $M$ is where $\mathcal{R}_{\mathsf{DFA}}(\langle M \rangle)(x) = 1$ if and only if $\delta^*(q_0, x) \in F$, where $\delta^* : Q \times \{0,1\}^* \to Q$ is the natural extension of $\delta$ to strings defined recursively as $\delta^*(q, \epsilon) = q$ and $\delta^*(q, sa) = \delta(\delta^*(q,s), a)$ for any $q \in Q$, $s \in \{0,1\}^*$, and $a \in \{0,1\}$.

**Definition 2.3.** A Context-Free Grammar (CFG) is a tuple $G = (N, T = \{0,1\}, P, S)$ where $N$ is a finite set of non-terminal symbols, $T = \{0,1\}$ is the terminal alphabet, $P$ is a finite set of production rules of the form $A \to \alpha$ where $A \in N$ and $\alpha \in (N \cup T)^*$, and $S \in N$ is the start symbol. Let $\langle G \rangle \in \{0,1\}^*$ denote the binary string encoding of $G$. The encoding system $\mathcal{R}_{\mathsf{CFG}}$ is a TM that reads $\langle G \rangle$ and outputs the description of a TM that simulates $G$. For any input string $x \in \{0,1\}^*$, the language characterized by $G$ is $\mathcal{R}_{\mathsf{CFG}}(\langle G \rangle)(x) = 1$ if and only if $x$ can be derived from $S$ by applying a finite sequence of production rules from $P$.

Finally, a Linear CFG is a CFG $G = (N, T = \{0,1\}, P, S)$ where each production rule in $P$ has at most one nonterminal symbol on its right-hand side. The encoding system for Linear CFGs is $\mathcal{R}_{\mathsf{L\text{-}CFG}}$, where $\mathcal{R}_{\mathsf{L\text{-}CFG}}(p) = \mathcal{R}_{\mathsf{CFG}}(p)$ for $p$ representing Linear CFGs, and otherwise $\mathcal{R}_{\mathsf{L\text{-}CFG}}(p)$ returns the TM with the empty language.

**Learning Setup.** With the ground truth function being denoted $f^*$ or $\mathcal{R}(p)$, we define the labeled training dataset consisting all data up to length $N$ as below, for all $N \in \mathbb{N}$: $D_N(f^*) := \{(x, f^*(x)) : x \in \{0,1\}^*, |x| \leq N\}$.

With $\mathcal{R}$ as the encoding system, an adversary picks any ground-truth function $f^* \in \mathcal{F}^{\mathcal{R}}$. A learning algorithm is a TM $\mathcal{A}$ which takes as input a training set $D_N(f^*)$ and outputs some $\hat{p} \in \{0,1\}^*$. We say a learning algorithm $\mathcal{A}$ *length-generalizably learns* a function $f^*$ at input length $N$ w.r.t. encoding system $\mathcal{R}$ iff $\mathcal{R}(\mathcal{A}(D_N(f^*))) = f^*$.

To give context to our notion of non-asymptotic length generalization, we describe Gold (1967)'s asymptotic notion of learnability in Appendix B.

**Complexity Measure.** We are interested in understanding the minimum length of training data which suffices for length generalization. To this end, a complexity measure for the functions in $\mathcal{F}$ is necessary. We assume a complexity measure $\mathcal{C} : \{0,1\}^* \to \mathbb{N}$, which assigns a complexity to each representation $p$, and define the complexity of the function $f$ as the minimum complexity of any representation of $f$ in $\mathcal{R}$, namely $\mathcal{C}^{\mathcal{R}}(f) := \min_{p \in \{0,1\}^*, \mathcal{R}(p) = f} \mathcal{C}(p)$. We may drop the superscript $\mathcal{R}$ and just use $\mathcal{C}$ for function complexity when $\mathcal{R}$ is clear from the context. We will make the following mild assumption that $\mathcal{C}$ is reasonably simple throughout the paper, unless otherwise stated.

**Assumption 2.4.** $\mathcal{C}$ is computable and there exists a TM $E$ which enumerates programs $p \in \{0,1\}^*$ in non-decreasing order of $\mathcal{C}(p)$. In particular, the range of $E$ is $\{0,1\}^*$ and $\forall i \leq i', \mathcal{C}(E(i)) \leq \mathcal{C}(E(i'))$. In addition, $\mathcal{C}$ is such that for each $c \in \mathbb{N}$, $|\{p \in \{0,1\}^* : \mathcal{C}(p) \leq c\}| < \infty$.

This assumption is easily satisfied by a standard choice of $\mathcal{C}$, namely where $\mathcal{C}(p)$ returns the length of $p \in \{0,1\}^*$. There is little loss in generality in just thinking of $\mathcal{C}$ as this standard complexity measure. Having a general $\mathcal{C}$ provides some extra flexibility and makes the results easier to understand.

**Definition 2.5** (Complexity Measures for DFAs and CFGs). The complexity measure $\mathcal{C}_{\mathsf{DFA}}$ for DFAs maps a DFA $M = (Q = [n], \Sigma, \delta, q_0, F)$ (represented by $\langle M \rangle$) to $n$, the number of states in $Q$. The number of states of any DFA is within a logarithmic factor of the length of its representation in bits ($n \log n$). For CFGs, given a CFG $G = (N, T, P, S)$, let $|P|$ be the total length of the production rules in $P$. The complexity measure of CFG $G$ is $\mathcal{C}_{\mathsf{CFG}}(\langle G \rangle) = |P| + |N| + |T|$, which is within a constant

factor of the length of $\langle G \rangle$ in bits.

Now we define non-asymptotic length generalization.

**Definition 2.6** (Non-Asymptotic Length Generalization). A function class $\mathcal{F} \subseteq \mathcal{F}^{\mathcal{R}}$ admits *non-asymptotic length generalization* w.r.t. encoding system $\mathcal{R}$ and complexity measure $\mathcal{C}$ if there exists a learning algorithm $\mathcal{A}$ and a computable function $\hat{N}_{\mathcal{A}}^{\mathcal{R},\mathcal{F}} : \mathbb{N} \to \mathbb{N}$ such that for all $f^* \in \mathcal{F}$ and for all $N' \geq \hat{N}_{\mathcal{A}}^{\mathcal{R},\mathcal{F}}(\mathcal{C}^{\mathcal{R}}(f^*))$, $\mathcal{A}$ length-generalizably learns $f^*$ at input length $N'$.

**Length Complexity.** For notation simplicity, we define the length complexity of a function $f^*$ for a learning algorithm $\mathcal{A}$ w.r.t. encoding system $\mathcal{R}$, $N_{\mathcal{A}}^{\mathcal{R}}(f^*)$, as the minimum length of training data which suffices for the learning algorithm $\mathcal{A}$ to length generalize on $f^*$: $N_{\mathcal{A}}^{\mathcal{R}}(f^*) := \min\{N \geq 0 : \forall n \geq N, \mathcal{R}(\mathcal{A}(D_n(f^*))) = f^*\}$

This quantity is $\infty$ if there is no such $N$ where $\forall n \geq N$, $\mathcal{R}(\mathcal{A}(D_n(f^*))) = f^*$. We also define the length complexity of functions in $\mathcal{F} \subset \mathcal{F}^{\mathcal{R}}$ up to complexity $c$ for a learning algorithm $\mathcal{A}$ w.r.t. encoding system $\mathcal{R}$, $N_{\mathcal{A}}^{\mathcal{R},\mathcal{F}}(c)$, as the maximum length complexity of any function in $\mathcal{F} \subset \mathcal{F}^{\mathcal{R}}$ with complexity at most $c$, given as $N_{\mathcal{A}}^{\mathcal{R},\mathcal{F}}(c) := \max_{f^* \in \mathcal{F} \text{ s.t. } \mathcal{C}^{\mathcal{R}}(f^*) \leq c} N_{\mathcal{A}}^{\mathcal{R}}(f^*) = \max_{p \in \{0,1\}^* \text{ s.t. } \mathcal{C}(p) \leq c \wedge \mathcal{R}(p) \in \mathcal{F}} N_{\mathcal{A}}^{\mathcal{R}}(\mathcal{R}(p))$. We denote $N_{\mathcal{A}}^{\mathcal{R},\mathcal{F}^{\mathcal{R}}}(c)$ by $N_{\mathcal{A}}^{\mathcal{R}}(c)$ for convenience.

As a concrete example, Proposition 2.7 shows that DFAs admits non-asymptotic length generalization w.r.t. the standard encoding system $\mathcal{R}_{\mathsf{DFA}}$ and complexity measure $\mathcal{C}_{\mathsf{DFA}}$ in Definition 2.5. Its proof is in Appendix D.

**Proposition 2.7** (Non-Asymptotic Length Generalization for DFAs). *Let $\mathcal{R}_{\mathsf{DFA}}$ be the DFA encoding system defined in Definition 2.2, and let $\mathcal{C}_{\mathsf{DFA}}$ be the number of states in DFA. Regular languages $\mathcal{F}^{\mathcal{R}}$ admits non-asymptotic length generalization w.r.t. encoding system $\mathcal{R}_{\mathsf{DFA}}$ and complexity measure $\mathcal{C}_{\mathsf{DFA}}$. More specifically, there exists a learning algorithm $\mathcal{A}$ such that $N_{\mathcal{A}}^{\mathcal{R}_{\mathsf{DFA}}}(c) \leq 2c - 2$ for all $c \in \mathbb{N}$.*

## 3. Characterization of Definition 2.6

In Section 3, we characterize the conditions under which a function class admits non-asymptotic length generalization w.r.t. an encoding system $\mathcal{R}$ and complexity measure $\mathcal{C}$ satisfying Assumption 2.4. First, we introduce Definition 3.1, an algorithm-independent version of length complexity, which coincides with the optimal algorithm-dependent length complexity, over all learning algorithms $\mathcal{A}$.

**Definition 3.1** (Length Complexity of Function Class). Given a function class $\mathcal{F}$, we define the length complexity of $\mathcal{F}$ as the minimum input length that can distinguish any two functions in $\mathcal{F}$: $N(\mathcal{F}) := \min\{n \in \mathbb{N} : \forall f \neq f' \in \mathcal{F}, \exists x \in \{0,1\}^{\leq n} \text{ s.t. } f(x) \neq f'(x)\}$.

---

**Algorithm 1** Minimum-Complexity Interpolator ($\mathcal{A}_{\mathsf{mci}}^{\mathcal{R},\mathcal{C}}$)

**Hyperparameters:** Complexity Measure $\mathcal{C}$, encoding system $\mathcal{R}$
**Input:** Finite Training Dataset $S \subseteq \{(x,y) \mid x \in \{0,1\}^*, y \in \{0,1\}\}$.
**Output:** $\arg\min_{p \in \{0,1\}^* : \forall(x,y) \in D_N(f^*), y = \mathcal{R}(p)(x)} \mathcal{C}(p)$

---

### 3.1. Optimality of Minimum Complexity Interpolator

The Minimum-Complexity Interpolator (Algorithm 1) is the main learning algorithm which we study in this work. Although it would be ideal to study a learning algorithm closer to what is used empirically to train transformers, we study the Minimum-Complexity Interpolator to abstract away the complex training dynamics of gradient-based methods like SGD, which are very non-trivial even for training 2 layer neural networks (Mahankali et al., 2023). We also acknowledge the limitation that the Minimum-Complexity Interpolator is computationally intractable for general function classes and encoding systems. For instance, it was shown that the problem of finding the minimum-state DFA for a regular language is NP-hard (Pitt & Warmuth, 1993).

We denote the Minimum-Complexity Interpolator learning algorithm by $\mathcal{A}_{\mathsf{mci}}^{\mathcal{R},\mathcal{C}}$ for short when the encoding system is $\mathcal{R}$ and complexity measure is $\mathcal{C}$. When the complexity measure $\mathcal{C}$ is the length of the program, $\mathcal{A}_{\mathsf{mci}}^{\mathcal{R},\mathcal{C}}$ is just the famous Minimum Description Length (MDL) algorithm (Rissanen, 1978). $\mathcal{A}_{\mathsf{mci}}^{\mathcal{R},\mathcal{C}}$ has the nice property that it is the best possible algorithm over all learning algorithms in minimizing $N_{\mathcal{A}}^{\mathcal{R},\mathcal{F}}(c)$, in the following sense.

**Theorem 3.2** (Optimality of Minimum-Complexity Interpolator). *Given any encoding system $\mathcal{R}$ and complexity measure $\mathcal{C}$, for all $c \in \mathbb{N}$, it holds that $N_{\mathcal{A}_{\mathsf{mci}}^{\mathcal{R},c}}^{\mathcal{R}}(c) = \min_{\mathcal{A}} N_{\mathcal{A}}^{\mathcal{R}}(c) = N(\mathcal{F}_c^{\mathcal{R}})$. As a consequence, the following three statements are equivalent:*

- *Function class $\mathcal{F}^{\mathcal{R}}$ admits non-asymptotic length generalization;*
- *Function class $\mathcal{F}^{\mathcal{R}}$ admits non-asymptotic length generalization, via learning algorithm $\mathcal{A}_{\mathsf{mci}}^{\mathcal{R},\mathcal{C}}$;*
- *For all $c \in \mathbb{N}$, length complexity of $\mathcal{F}_c^{\mathcal{R}}$, $N(\mathcal{F}_c^{\mathcal{R}})$, is computably bounded in $c$.*

The proof of Theorem 3.2 is in Appendix C. Though the above optimality result of $\mathcal{A}_{\mathsf{mci}}^{\mathcal{R},\mathcal{C}}$ holds for all complexity measures $\mathcal{C}$, $\mathcal{A}_{\mathsf{mci}}^{\mathcal{R},\mathcal{C}}$ may not be computable without restrictions on the complexity measure. For example, it is well-known that Kolmogorov complexity is not computable (Li & Vitányi, 2008). Lemma 3.3 shows that $\mathcal{A}_{\mathsf{mci}}^{\mathcal{R},\mathcal{C}}$ is indeed computable under Assumption 2.4.

**Lemma 3.3.** *Under Assumption 2.4, Algorithm 1 is computable and thus a valid learning algorithm.*

The proof of Lemma 3.3 is in Appendix C. In the remaining sections, we may omit $\mathcal{R}$ and $\mathcal{C}$ in the superscripts of $\mathcal{A}^{\mathcal{R},\mathcal{C}}_{\mathrm{mci}}$ when they are clear from context for convenience.

### 3.2. Equivalence to Decidability of Language Equivalence Problem

The main goal of this paper is to seek concrete upper bounds for the length complexity $N^{\mathcal{R}}_{\mathcal{A}}(c)$ of various function classes. Surprisingly, such upper bounds are not always computable. In this section, we show that there exist computable upper bounds on the length complexity if and only if the *Language Equivalence Problem* for encoding system $\mathcal{R}$ is decidable, where the *Language Equivalence Problem* for encoding system $\mathcal{R}$ is the computational problem where given any $p, q \in \{0, 1\}^*$, determine whether $\mathcal{R}(p) = \mathcal{R}(q)$.

**Lemma 3.4.** *For any encoding system $\mathcal{R}$ and complexity measure $\mathcal{C}$ satisfying Assumption 2.4, the Language Equivalence problem for $\mathcal{R}$ is decidable if and only if length complexity of $\mathcal{F}^{\mathcal{R}}_c$, $N(\mathcal{F}^{\mathcal{R}}_c)$, is computably bounded in c. Thus it is also equivalent to the property that $\mathcal{F}^{\mathcal{R}}$ admits non-asymptotic length generalization.*

As a consequence, we have the following impossibility result for non-asymptotic length generalization for a special case of CFGs: Linear CFGs. Comparing Proposition B.2 and Proposition 3.5, we see that CFG serves as a concrete example for the separation between length generalization in the limit and non-asymptotic length generalization.

**Proposition 3.5** ((Linear) CFGs only admit length generalization in the limit). *Recall $\mathcal{R}_{L\text{-}CFG}$ is the encoding system for Linear CFGs defined in Definition 2.3 and $\mathcal{C}_{CFG}(\langle G \rangle)$ is the complexity measure that maps CFG $G = (N, T, P, S = \{0, 1\})$ to $|N| + |T| + |P|$. Then for any learning algorithm $\mathcal{A}$, the length complexity, $N^{\mathcal{R}_{L\text{-}CFG}}_{\mathcal{A}}$, is not computably bounded. That is, Linear CFGs do not admit non-asymptotic length generalization (w.r.t. standard CFG encoding system $\mathcal{R}_{CFG}$), and neither does the set of all CFGs.*

The proofs of Lemma 3.4 and Proposition 3.5 are in Appendix C and Appendix D, respectively. In Appendix C.3, we show equivalence between non-asymptotic length generalization to a variant of "finite identification" proposed in Gold (1967). The results are summarized in Figure 1.

## 4. Related Work

Solomonoff (1964) proposed Bayesian-inference-based algorithms which when given a sequence of symbols, predict the next symbol according to the posterior distribution computed from the Solomonoff-Levin prior distribution. Gold (1967) introduced Identification-in-the-Limit as an asymptotic notion of learnability and proved that many classes of functions can be learned in this sense.

Zhou et al. (2023) propose the RASP-L Conjecture, which says that whether a transformer length generalizes on a particular ground-truth function $f_*$ is well predicted by whether $f_*$ has a short RASP-L description. Huang et al. (2024) formulate the problem of length generalization–and the RASP-L Conjecture–formally as a version of Identification-in-the-Limit and prove asymptotic results for identifying languages expressible by Limit Transformers. None of the works above provide non-asymptotic guarantees. We distinguish this work from previous work by providing non-asymptotic bounds on the length of the training data required in order to guarantee that the learner outputs a single hypothesis which exhibits perfect length generalization.

(Weiss et al., 2021) (Zhou et al., 2023), (Yang & Chiang, 2024), and (Shaw et al., 2024) study programming languages which capture the set of functions which transformers can express (like RASP).

Regarding theoretical works for length generalization not related to transformers, Marsden et al. (2024) prove length generalization for learning linear dynamical systems with SGD. Abbe et al. (2024) prove out-of-domain generalization for boolean functions of a fixed input size.

## 5. Main Results: Non-Asymptotic Length Generalization of C-RASP

### 5.1. Recap: Definition of C-RASP

Our main results pertain to a class of functions called C-RASP. C-RASP is a variant of RASP that, with alphabet $\Sigma = \{0, 1\}$, defines a class of functions from $\{0, 1\}^*$ to $\{0, 1\}$, where only certain sequence-to-sequence operations are permitted. C-RASP was shown to be a subset to the class of functions expressible by transformers with infinite-precision activations and a superset to that expressible by finite-precision transformers (Yang & Chiang, 2024).

**Definition 5.1** (C-RASP, (Yang & Chiang, 2024)). A sequence is an element of $\mathbb{N}^*$. A C-RASP program over alphabet $\Sigma = \{0, 1\}$ is defined as a series of $n < \infty$ C-RASP operations. We denote the $h^{(1)}, \ldots, h^{(n)}$ as the output sequences of each C-RASP operation. Denote $x \in \{0, 1\}^*$ as the input sequence to the C-RASP program. Denote $h^{(i)}_j$ as the $j$th element of the sequence $h^{(i)}$, where $i \in [n]$ and $j \in [|x|]$. Table 2 shows the list of allowed operations in C-RASP. The partial-sum operator $\mathtt{ps} : \{0, 1\}^* \to \mathbb{N}^*$ is defined as: $\forall h \in \{0, 1\}^*, \forall j \in [|x|], \mathtt{ps}(h)_j = \sum_{l \in [j]} h_l$. The entire C-RASP program is a mapping $\{0, 1\}^* \to \{0, 1\}^*$. We take the last bit of the output sequence as the output bit, yielding a function $\{0, 1\}^*$ to $\{0, 1\}$.

There are several natural parameters that contribute to the complexity of C-RASP functions. First, we study the impact of the following notion of precision of the parameters of

C-RASP functions on the difficulty of length generalization.

**Definition 5.2** ($p$-precision). An integer of absolute value at most $p$ is of $p$-precision. A rational number between $[0, 1]$ is of $p$-precision if in simplest form, where the numerator and denominator are relatively prime, its denominator is at most $p$ in magnitude. A tuple of rational numbers in $[0, 1]$ is precision $p$ if the least common denominator of its entries is at most $p$ in magnitude.

This notion of precision makes sense for C-RASP since C-RASP only allows integer combinations of previously computed variables instead of arbitrary linear combinations.

We also study the impact of depth of C-RASP programs on the difficulty of length generalization where depth is, loosely, the maximum number of sequentially-applied ps operations along any part of the C-RASP program. We study depth-1 and depth-2 programs, which we will also refer to as 1-layer and 2-layer programs. We will first prove length generalization results pertaining to the following subset of 1-layer C-RASP, which we call C-RASP$^1$. Note that for convenience, we use functions $f \in \mathcal{F}^{\mathcal{R}}$ and descriptions $p \in \{0, 1\}^*$ interchangeably in the following definitions.

**Definition 5.3** (C-RASP$^1$). With integer $T$, let C-RASP$^{1,T}$ denote the set of programs of the following form. Each program $f$ has parameters $a, b, d \in [-T, T]$, $a > 0$. For any $n > 0$, on input $x \in \{0, 1\}^n$, $f$ computes: $f(x) = \mathbb{1}[a \cdot \mathtt{ps}(x)_n - b \cdot n - d > 0]$. Then, C-RASP$^1 = \bigcup_{T \geq 1}$ C-RASP$^{1,T}$. Given a function $f \in$ C-RASP$^1$, the complexity measure $\mathcal{C}(f) = \max(|a|, |b|, |d|)$, the precision of $f$'s parameters in the sense of Definition 5.2.

Here, the encoding system $\mathcal{R}$ maps a string $p$ to a C-RASP$^1$ function. The string $p$ is interpreted by $\mathcal{R}$ to encode 3 parameters, each taking $\Theta(\log T)$ bits to encode. Invalid encodings are mapped to a default C-RASP$^1$ function. Thus, the complexity measure proposed above, which takes in $p$ and returns the maximum precision of its parameters, returns an integer which is roughly exponential in the length of $p$.

Our main length generalization result will apply to a subset of 2-layer C-RASP, which we call C-RASP$^2$. For 2 layer programs, the width $K$ of the first layer becomes a natural parameter to study.

**Definition 5.4** (C-RASP$^2$). With integers $T$ and $1 \leq K \leq T^2$, let C-RASP$^{2,K,T}$ be the set of programs of the following form. Each program $f$ has parameters $0 < z \leq T$, $\forall i \in [K]$, $a^{(i)}, b^{(i)}, \lambda_i \in \{-T, \ldots, T\}$, with $a^{(i)} > 0$. We require that for all $i \in [K]$, $\frac{b^{(i)}}{a^{(i)}} \in (0, 1)$ and is distinct from $\frac{b^{(i')}}{a^{(i')}}$ for $i' \neq i$. We also require $\sum_{i \in [K]} \lambda_i > z$.

For any $n > 0$, on input $x \in \{0, 1\}^n$, the first layer computes the values of $K$ heads, $\{h^{(i)}\}_{i \in [K]}$, on the $n$ prefixes of $x$: $\{\{x_1\}, \{x_1, x_2\}, \ldots, \{x_1, \ldots, x_n\}\}$ as follows: $\forall j \in [n], \forall i \in [K], h_j^{(i)} = \mathbb{1}[\mathtt{ps}(x)_j > \frac{b^{(i)}}{a^{(i)}} j]$. Subscript $j$

indicates the value of a quantity on the $j$th prefix of $x$.

The second layer computes the output, which is the $n$th bit of the final sequence: $f(x) = \mathbb{1}[\sum_{i \in [K]} \lambda_i \mathtt{ps}(h^{(i)})_n > z \cdot n]$.

Then, C-RASP$^2 = \bigcup_{1 \leq T, 1 \leq K \leq T^2}$ C-RASP$^{2,K,T}$. Given a function $f \in$ C-RASP$^2$, let $K(f)$ be the number of heads, $h^{(i)}$, in the first layer of $f$ and let $T(f) := \max(\max_{i \in [K(f)]} |a^{(i)}|, \max_{i \in [K(f)]} |b^{(i)}|, \max_{i \in [K(f)]} |\lambda_i|, |z|)$. The complexity measure is $\mathcal{C}(f) = T(f)^{K(f)}$, the precision of the function's parameters to the power of the number of heads.

We refer to functions in C-RASP$^{2,K,T}$ as 2 layer, $K$-head, $T$-precision programs, where each intermediate variable $h^{(i)}, i \in [K]$ is called a head.

Here, the encoding system $\mathcal{R}_{\text{C-RASP}^2}$ maps a string $p$ to a C-RASP$^2$ function. The string $p$ is interpreted by $\mathcal{R}_{\text{C-RASP}^2}$ to encode $\Theta(K)$ parameters, each taking $\Theta(\log T)$ bits to encode, so that the entire encoding of a C-RASP$^{2,K,T}$ function is roughly $\Theta(K \log T)$ bits long. Invalid encodings are mapped to a default C-RASP$^2$ function. Thus, the complexity measure $\mathcal{C}(f) = T(f)^{K(f)} = \exp(K(f) \log T(f))$ returns an integer which is roughly exponential in $|p|$.

## 5.2. Non-Asymptotic Length Generalization of C-RASP$^1$ and C-RASP$^2$

The following result states that in order to identify the ground-truth function from the set C-RASP$^1$ where the precision of parameters is at most $T$, it suffices for the Minimum-Complexity Interpolator to receive (string, label) pairs for all strings of length at most $O(T^2)$.

**Theorem 5.5** (C-RASP$^1$ Length Generalization). *Let $\mathcal{F} = $ C-RASP$^1$ and $\mathcal{C}(f) = \max(|a|, |b|, |d|)$, defined in Definition 5.3. Then $\forall T \in \mathbb{N}$, we have $N_{\mathcal{A}_{mci}}(T) \leq O(T^2)$. That is, the Minimum-Complexity Interpolator, with complexity $\mathcal{C}$ and function class $\mathcal{F}$, can length generalize given inputs of length $O(T^2)$ when the ground-truth has complexity $T$.*

The following is our main length generalization result. It states that in order to identify the ground-truth function from the set C-RASP$^2$ where the precision of parameters is at most $T$ and number of heads is at most $K$, it suffices for the Minimum-Complexity Interpolator to receive (string, label) pairs for all strings of length at most $O(T^{O(K)})$.

**Theorem 5.6** (C-RASP$^2$ Length Generalization, Main Result). *Let $\mathcal{F} = $ C-RASP$^2$ and $\mathcal{C}(f) = T(f)^{K(f)}$, defined in Definition 5.4. If the ground-truth function $f_*$ has $T(f_*) \leq T$ and $K(f_*) \leq K$, then the Minimum-Complexity Interpolator, with complexity $\mathcal{C}$ and function class $\mathcal{F}$, can length generalize given inputs of length $O(T^{O(K)})$.*

By Theorem 5.6, C-RASP$^{2,K,T}$ can be learned in a length generalizable way with inputs of length $O(T^{O(K)})$. By

Theorem 5.5, C-RASP$^{1,T}$ can be learned in a length generalizable way with inputs of length $O(T^2)$. Our upper bound on the minimum length required to learn C-RASP$^{2,K,T}$ is much larger than that of C-RASP$^{1,T}$. We don't have a lower bound on the length of inputs required to identify C-RASP$^{2,K,T}$, but our best guess is that one of the form $\Omega(T^K)$ exists in the regime $K \le \Theta(\log T)$.

The proof of Theorem 5.5 can be found in Appendix E.1. Below we sketch the proof of Theorem 5.6.

## 6. Proof Sketch of Theorem 5.6.

Theorem 5.6 follows as a Corollary to the following, stronger Theorem 6.1, which says that the Minimum Complexity Interpolator with complexity measure $\mathcal{C}(f) = T(f)^{K(f)}$ will length generalize if it receives inputs of length $N_{\mathcal{A}_{mci}}(\alpha) \le O(\alpha^{O(1)})$, when the ground-truth function has complexity at most $\alpha$.

**Theorem 6.1.** *Let $\mathcal{F} = $ C-RASP$^2$ and $\mathcal{C}(f) = T(f)^{K(f)}$, as in Definition 5.4. Then $\forall \alpha \in \mathbb{N}$, $N_{\mathcal{A}_{mci}}(\alpha) \le O(\alpha^{O(1)})$.*

Theorem 6.1 is proved in Appendix E.2. The proof that Theorem 5.6 follows as a Corollary to Theorem 6.1 is simple, and deferred to Appendix A for brevity. Below, we sketch the proof Lemma 6.2, a weaker version of Theorem 6.1. Although weaker, our proof sketch of Lemma 6.2 will still illustrate the key ideas of the proof of Theorem 6.1.

**Lemma 6.2.** *Let $\mathcal{F} = $ C-RASP$^2$ and $\mathcal{C}(f) = T(f)^{K(f)}$, as in Definition 5.4. Then $\forall \alpha \in \mathbb{N}$, $N_{\mathcal{A}_{mci}}(\alpha) \le O(\alpha^{O(\log^2 \alpha)})$.*

To prove Lemma 6.2, it suffices to prove Lemma 6.3. We will upper bound, for any $T$ and $K$, the minimum length of inputs required to distinguish any two unequal C-RASP$^2$ functions that have at most $K$ heads and $T$ precision.

**Lemma 6.3** (Length Bound on C-RASP$^{2,K,T}$). *For any $1 \le T, 1 \le K \le T^2$, for all $f, f' \in $ C-RASP$^{2,K,T}$ such that $f \ne f'$, there exists a string $x_* \in \{0,1\}^*$ of length at most $O(T^{O(K^2)})$ such that $f(x_*) \ne f'(x_*)$.*

The proof that Lemma 6.2 is a Corollary to Lemma 6.3 is simple and is deferred to Appendix A. The rest of the proof sketch describes how to prove Lemma 6.3. To do this, given two arbitrary $f, f' \in $ C-RASP$^{2,K,T}$ that are not equal, we show the existence of a *short* string $x_*$ which distinguishes $f$ and $f'$, in the sense that $f(x_*) \ne f'(x_*)$.

### 6.1. Key Definitions.

Suppose $f, f' \in $ C-RASP$^{2,K,T}$, with parameters $(a^{(i)})_{i \in [K]}, (b^{(i)})_{i \in [K]}, (\lambda_i)_{i \in [K]}, z$ and $((a^{(i)})')_{i \in [K]}, ((b^{(i)})')_{i \in [K]}, (\lambda_i')_{i \in [K]}, z'$, respectively.

Suppose the set of unique numbers $R := \{\frac{b^{(i)}}{a^{(i)}}\}_{i \in [K]} \cup \{\frac{(b^{(i)})'}{(a^{(i)})'}\}_{i \in [K]} \subset (0,1)$ between the first layer of $f$ and $f'$

has size $k$, where $K \le k \le 2K$. We will refer to these numbers as "slopes." We will denote $R = \{s_j\}_{j \in [k]} \subset (0,1)$, where $s_1$ is the largest slope and $s_k$ is the smallest slope, and the slopes are sorted in descending order so that $s_1 > \ldots > s_k$. For $i \in [K]$, let ord$(1,i) : [K] \to [k]$ be the index within $R$ of the $i$th slope of $f$, $\frac{b^{(i)}}{a^{(i)}}$. Let ord$(2,i) : [K] \to [k]$ be the index within $R$ of the $i$th slope of $f'$, $\frac{(b^{(i)})'}{(a^{(i)})'}$. In the following exposition, we will refer to "line $j$" as the homogeneous, 2D line y $= s_j$x, with slope $s_j, j \in [k]$. We will denote "line $j$" by the symbol $l_j$. We will call the set of $k \le 2K$ unique slopes $\{s_i\}_{i \in [k]}$ a configuration.

**Definition 6.4.** A $(k,T)$-configuration is a set of $k$ distinct $T$-precision rational numbers $\{s_i\}_{i \in [k]} \subset (0,1)$.

We will refer to strings in $\{0,1\}^*$ synonymously as *discrete test-functions*, because there is a one-to-one correspondence between $x$ and the sequence of 2D points $\{(j, \text{ps}(x)_j)\}_{j \in [|x|]} \subset \mathbb{R}^2$. The latter set of points acts as a "test" of whether two programs $f, f' \in $ C-RASP$^{2,K,T}$ are different or not.

**Definition 6.5** (Discrete Test-Function). Given a $(k,T)$-configuration $\{s_i\}_{i \in [k]}$, a discrete test-function $\mathcal{X}$, with respect to $\{s_i\}_{i \in [k]}$ and of length $n < \infty$, is a function $\{0,1,\ldots,n\} \to \{0,1,\ldots,n\}$ where $\mathcal{X}(0) = 0$ and $\forall j \in [n]$, $\mathcal{X}(j) = \mathcal{X}(j-1)$ or $\mathcal{X}(j) = \mathcal{X}(j-1) + 1$. The induced activations $(B_1(\mathcal{X}), \ldots, B_k(\mathcal{X}))$ of $\mathcal{X}$ with respect to the $(k,T)$-configuration are defined as: $\forall i \in [k]$, $B_i(\mathcal{X}) := \frac{1}{n} \sum_{j=1}^{n} \mathbb{1}[\mathcal{X}(j) > s_i \cdot j]$

In our proof sketch, we will need to analyze properties of a continuous analog to discrete test-functions, which can be thought of as corresponding to infinite-length strings.

**Definition 6.6** (Continuous Test-Function). Given a $(k,T)$-configuration $\{s_i\}_{i \in [k]}$, a continuous test-function $\mathcal{Y}$, with respect to $\{s_i\}_{i \in [k]}$, is a 1-Lipschitz, monotone non-decreasing continuous function $[0,1] \to [0,1]$, with $\mathcal{Y}(0) = 0$. Continuous test-functions can only intersect the $k$ lines $\{l_i\}_{i \in [k]}$ of slopes given by $\{s_i\}_{i \in [k]}$ at finitely many points. The induced activations $(B_1(\mathcal{Y}), \ldots, B_k(\mathcal{Y}))$ of $\mathcal{Y}$ w.r.t. $\{s_i\}_{i \in [k]}$ are: $\forall i \in [k]$, $B_i(\mathcal{Y}) := \int_0^1 \mathbb{1}[\mathcal{Y}(j) > s_i \cdot j]dj$.

We study continuous test-functions as a proxy for discrete test-functions. As each C-RASP$^2$ function can be thought of as a linear threshold function over the induced activations of the input string (test-function) with respect to the $(k,T)$-configuration given by the parameters of its first layer, it will be important to study the activations which can be induced by continuous test-functions: $\mathbb{A}(\{s_i\}_{i \in [k]}) := \{(B_1(\mathcal{Y}), \ldots, B_k(\mathcal{Y})) : \mathcal{Y} \text{ continuous test-function w.r.t } \{s_i\}_{i \in [k]}\}$.

The importance of the activations induced by continuous test-functions motivates the segmentation of continuous test-functions at the points where the continuous test-function

intersects the $k$ lines, $\{y = s_i x : i \in [k]\}$.

**Definition 6.7** (Segment). Given any $(k, T)$-configuration $\{s_i\}_{i \in [k]} \subset (0, 1)$, a *segment* is a restricted test-function $S : [a, b] \to [0, 1]$ where $[a, b] \subset [0, 1]$ which maps a continuous subset $[a, b]$ to $[0, 1]$. $S$ is 1-Lipschitz and monotone non-decreasing. The segment's start-point $(a, S(a))$ and the end-point $(b, S(b))$ each lie on one of the $k$ lines, in the sense that there exists some $i, j \in [k]$ where $S(a) = s_i \cdot a$ and $S(b) = s_j \cdot b$, where $i = j$ or $|i - j| = 1$. No other points $(x, S(x))$, $x \in (a, b)$ can lie on a line $l_1, \ldots, l_k$.

Figure 2 depicts a continuous test-function. Figure 7 shows four generic types of segments. Two different test-functions can share the same schema: roughly, the order which the test-functions cross the $k$ lines.

**Definition 6.8** (Schema). Given any $(k, T)$-configuration $\{s_i\}_{i \in [k]} \subset (0, 1)$, a *schema* $Y$ is a blueprint for a continuous test-function, specifying a sequence of lines $\{l_i\}_{i \in [k]}$ that any test-function of the schema must cross. It consists of an integer $0 < M < \infty$ and two tuples $\{idx(i)\}_{i \in [M]} \subset [k]^M$, $\{sec_i\}_{i \in [M]} \subset [k+1]^M$, where $|idx(i) - idx(i+1)| \leq 1$ for all $i \in [M - 1]$. If $|idx(i) - idx(i + 1)| = 1$, then $sec_{i+1}$ is unique and must be $\max(idx(i), idx(i + 1))$. If $idx(i) = idx(i + 1)$, then $sec_{i+1}$ can be either $idx(i + 1)$ or $idx(i + 1) + 1$. $sec_1$ can be either $idx(1)$ or $idx(1) + 1$.

Any continuous test-function of schema $Y = (\{idx(i)\}_{i \in [M]}, \{sec_i\}_{i \in [M]})$ consists of exactly $M$ segments $S_1, S_2, \ldots, S_M$ whose domains are a partition of $[0, 1]$. For each $i \in [M]$, the $i$th segment $S_i$'s end-point lies on $l_{idx(i)}$. For $i > 1$, $S_i$'s start-point lies on line $l_{idx(i-1)}$, and $S_1$'s start-point is the origin, $(0, 0)$. In addition, the $i$th segment must be contained in $\text{Sector}_{sec_i}$, where $\text{Sector}_1$ is the subset of the positive quadrant of the 2D plane which lies above $y = s_1 x$, $\text{Sector}_{k+1}$ is the subset of the positive quadrant which lies below $y = s_k x$, and for $i \in \{2, \ldots, k\}$, $\text{Sector}_i$ is the subset of the positive quadrant which lies below $y = s_{i-1} x$ and above $y = s_i x$.

There is a useful approximation property between discrete and continuous test-functions which we use later.

**Lemma 6.9** (Discrete Approximation to Continuous Test–Function, weaker version of Lemma E.24). *Suppose we are given a $(k, T)$-configuration $\{s_i\}_{i \in [k]}$. For any schema $Y$ of $M$ segments, suppose $\mathcal{Y}$ is any continuous test-function of schema $Y$, with segment lengths $(\overline{n}_1(\mathcal{Y}), \ldots, \overline{n}_M(\mathcal{Y})) \in [0, 1]^M$ (where we assumed WLOG that $\sum_j \overline{n}_j(\mathcal{Y}) = 1$). Suppose every $\overline{n}_j(\mathcal{Y})$ is a rational number and that the common denominator of all $(\overline{n}_j(\mathcal{Y}))_{j \in [M]}$ is $p$. Then there exists an $n_0 \leq O(p \cdot T^k)$ so that for any positive integer multiple $n$ of $n_0$, there exists a discrete test-function $\mathcal{X}$ of length $n$ so that $\forall i \in [k], |B_i(\mathcal{Y}) - B_i(\mathcal{X})| \leq \frac{T^2 + M}{n}$.*

**Definition 6.10** (Margin). Given a linear inequality $L$ over $M$ variables and a point $x \in \mathbb{R}^M$, define $L(x)$ as the dif-

ference between the left-hand-side and right-hand-side of the inequality when the coordinates of $x$ are plugged into $L$. Let $L(x) = 0 \iff x$ satisfies the inequalities with equality. We say $L(x)$ is the margin of $x$ for $L$.

## 6.2. Proof Sketch of Lemma 6.3.

**Proof Plan.** We want to show that any two C-RASP programs $f, f' \in \text{C-RASP}^{2,K,T}$ that are not equal must be distinguished by some string $x'$ of length at most $O(T^{O(K^2)})$. Consider any two C-RASP programs $f, f' \in \text{C-RASP}^{2,K,T}$, with parameters $(a^{(i)})_{i \in [K]}, (b^{(i)})_{i \in [K]}, (\lambda_i)_{i \in [K]}, z$ and $((a^{(i)})')_{i \in [K]}, ((b^{(i)})')_{i \in [K]}, (\lambda'_i)_{i \in [K]}, z'$, respectively.

For any $n > 0$, consider the induced activations by an arbitrary string $x \in \{0, 1\}^n$, defined as $\{\frac{\text{ps}(h^{(i)}(x)_n)}{n}\}_{i \in [K]} \cup \{\frac{\text{ps}((h^{(i)})'(x)_n)}{n}\}_{i \in [K]}$, per Definition 6.5. With $k \leq 2K$ being the number of unique slopes (i.e. unique values in $\{\frac{b^{(i)}}{a^{(i)}}\}_{i \in [K]} \cup \{\frac{(b^{(i)})'}{(a^{(i)})'}\}_{i \in [K]}$) among the heads in the first layers of $f$ and $f'$, denote $(B_1(x), B_2(x), \ldots, B_k(x)) := \{\frac{\text{ps}(h^{(i)}(x)_n)}{n}\}_{i \in [K]} \cup \{\frac{\text{ps}((h^{(i)})'(x)_n)}{n}\}_{i \in [K]}$.

To argue that the existence of distinguisher $x_0$ for $f$ and $f'$ implies existence of a short distinguisher for for $f$ and $f'$, it suffices to find an $x \in \{0, 1\}^n, n \leq O((KT)^{O(K^2)})$ that induces activations $(B_i(x))_{i \in [k]}$ where either $\sum_{i \in [K]} \lambda_i B_{\text{ord}(1,i)}(x) > z$ and $\sum_{i \in [K]} \lambda'_i B_{\text{ord}(2,i)}(x) \leq z'$ or $\sum_{i \in [K]} \lambda_i B_{\text{ord}(1,i)}(x) \leq z$ and $\sum_{i \in [K]} \lambda'_i B_{\text{ord}(2,i)}(x) > z'$. In particular, define the following halfspaces, $H_1^+, H_1^-, H_2^+, H_2^-$. $\forall j \in \{1, 2\}, H_j^+ := \{(B_1, \ldots, B_k) : \sum_{i \in [K]} \lambda_i B_{\text{ord}(j,i)} > z\}$ and $H_j^- := \{(B_1, \ldots, B_k) : \sum_{i \in [K]} \lambda_i B_{\text{ord}(j,i)} < z\}$. With this goal in mind, we use the following proof plan.

1. First, we argue that if $f \neq f'$, then there exists a continuous test-function $\mathcal{Y}_0$ which induces activations $(B_1(\mathcal{Y}_0), \ldots, B_k(\mathcal{Y}_0))$ that is either contained in $H_1^+ \cap H_2^+$ or $H_1^- \cap H_2^-$. This is a non-trivial step which uses a technical Lemma E.30, whose proof is deferred to Appendix E.6. WLOG, suppose that $(B_1(\mathcal{Y}_0), \ldots, B_k(\mathcal{Y}_0)) \in H_1^+ \cap H_2^+$.

2. Next, we will show that because the set $\mathbb{A}(\{s_i\}_{i \in [k]}) \cap H_1^+ \cap H_2^+$ is non-empty, then $\mathbb{A}(\{s_i\}_{i \in [k]}) \cap H_1^+ \cap H_2^+$ must be at least a minimum "size." We accomplish this via the following steps.

   - With Corollary E.14, we first characterize the set $\mathbb{A}(\{s_i\}_{i \in [k]})$ as the union of a finite number of polytopes, where each polytope is equal to the set of activations which can be induced by test-functions of a particular schema. In this way, each polytope corresponds to a unique schema, and the finite set of schemas corresponding to the finite set of polytopes

serves as a "basis" of all test-functions.

- We convert $\mathcal{Y}_0$ into a new test-function which induces the same activations as $\mathcal{Y}_0$, but which follows one of the "basis" schema, $Y$. This new test-function, which we call $\mathcal{Y}_1$, is specified by a tuple of $M := M(Y) \leq k^2$ numbers $(n_1, n_2, \ldots, n_M) \in [0,1]^M$, the lengths of the segments of $\mathcal{Y}_1$ in schema $Y$.

- Let $A^{(M)}(Y) \subset [0,1]^M$ denote the polytope of valid settings of the lengths of segments of schema $Y$, as explained in Lemma E.6. Then, there are two halfspaces $H_1^{(M)}$ and $H_2^{(M)}$ over $M$ variables such that the polytope $P := \mathrm{cl}(A^{(M)}(Y) \cap H_1^{(M)} \cap H_2^{(M)})$ is the set of settings of the lengths of segments of schema $Y$, which correspond to test-functions whose induced activations are contained in $\mathrm{cl}(H_1^+) \cap \mathrm{cl}(H_2^+)$. By the existence of $\mathcal{Y}_1$, $P \neq \emptyset$. Moreover, $P$ is a polytope whose faces are low-precision, in the sense of Definition 5.2.

- Denote the set of vertices of $P$ as $V$. Let $c := \frac{1}{|V|} \sum_{v \in V} v \in P$ be the average of the vertices of $P$. We apply Lemma E.18 to $P$ to derive a lower-bound of the margin of $c$ to the faces of $P$, of $\gamma \geq \frac{1}{|V|} \cdot \frac{1}{(\mathrm{poly}(K,T)\sqrt{M})^M}$. Since $c \in P \subset [0,1]^M$ is a valid setting of the lengths of the segments of schema $Y$, there is a continuous test-function $\mathcal{Y}_2$ of schema $Y$ with segment lengths given by $c$.

3. It follows from steps 1 and 2 of the proof plan that if $f, f'$ are not equal, then there exists a point $c \in P$ whose margin to the faces of $P$ is at least $\gamma \geq \frac{1}{3M^2} \cdot \frac{1}{(\mathrm{poly}(K,T)\sqrt{M})^M}$, where we used the fact that $|V| \leq 3M^2$, from Lemma E.22.

   Using Lemma E.29, we perturb the coordinates of $c \in [0,1]^M$ slightly to attain a point that still has $\frac{\gamma}{2}$ margin to the faces of $P$, but whose coordinates have precision at most $\frac{\mathrm{poly}(K,T)}{\gamma}$, in the sense of Definition 5.2. We will call this perturbed point $c_* := (n_1^{(c_*)}, n_2^{(c_*)}, \ldots, n_M^{(c_*)}) \in [0,1]^M$. Let $(B_1^*, \ldots, B_k^*)$ be the activations induced by a test-function of schema $Y$ with segment lengths set according to $(n_1^{(c_*)}, n_2^{(c_*)}, \ldots, n_M^{(c_*)})$.

4. Finally, we apply Lemma 6.9 to the test-function of schema $Y$, with segment lengths set according to $(n_1^{(c_*)}, n_2^{(c_*)}, \ldots, n_M^{(c_*)})$, to attain a discrete test-function $\mathcal{X}'$ of length $n$ whose induced activations are such that $\|(B_1(\mathcal{X}'), \ldots, B_k(\mathcal{X}')) - (B_1^*, \ldots, B_k^*)\|_\infty \leq O(\frac{\mathrm{poly}(K) \cdot \mathrm{poly}(T)}{n})$. Note that there is a minimal value $n_0 \leq \mathrm{poly}(K,T) \cdot T^{O(K)} \cdot \frac{1}{\gamma}$, which $n$ must be a multiple of, as a requirement of Lemma 6.9.

   Now, we discuss how large $n$ needs to be so that discrete test-function $\mathcal{X}'$ distinguishes $f$ and $f'$. First,

$n$ must be larger than $n_0$, which was a prerequisite of using Lemma 6.9. There is a second, important requirement. WLOG, suppose that the continuous test-function $\mathcal{Y}_0$ outputted in step 1 of the proof plan is such that: $(B_1(\mathcal{Y}_0), \ldots, B_k(\mathcal{Y}_0)) \in H_1^+ \cap H_2^+$. Then, the procedure in steps 2 and 3 are such that $(B_1(\mathcal{Y}_1), \ldots, B_k(\mathcal{Y}_1)) \in H_1^+ \cap H_2^+$, $(B_1(\mathcal{Y}_2), \ldots, B_k(\mathcal{Y}_2)) \in H_1^+ \cap H_2^+$, and $(B_1^*, \ldots, B_k^*) \in H_1^+ \cap H_2^+$. Now, we would like $n$ to be large enough so that $(B_1(\mathcal{X}'), \ldots, B_k(\mathcal{X}'))$ is also in $H_1^+ \cap H_2^+$, so that $\mathcal{X}'$ would distinguish $f$ and $f'$. To do this, we need to ensure the coordinate-wise distance between $(B_1(\mathcal{X}'), \ldots, B_k(\mathcal{X}'))$ and $(B_1^*, \ldots, B_k^*)$ is smaller than the margin $\gamma$ of $c_*$ on the faces of $H_1^{(M)} \cap H_2^{(M)}$ divided by the maximum $L_1$ norm of the linear inequalities defining the faces of $\mathbb{A}(\{s_i\}_{i \in [k]}) \cap H_1^+ \cap H_2^+$. It is important that the margin of $c_*$ is large, as the smallest value of $n$ which ensures that $(B_1(\mathcal{X}'), \ldots, B_k(\mathcal{X}')) \in H_1^+ \cap H_2^+$ is proportional to $\frac{1}{\gamma}$. In short, we need $n \gtrsim \Theta(\max(n_0, \frac{\mathrm{poly}(K,T)}{\gamma}))$ for the resulting discrete test-function $\mathcal{X}'$ to distinguish $f$ and $f'$. Since $M \leq k^2$, $k \leq 2K$, and $K \leq T^2$, it suffices for $n = (\mathrm{poly}(T))^M = O(T^{O(K^2)})$. $\mathcal{X}'$ corresponds to a string $x' \in \{0,1\}^{O(T^{O(K^2)})}$ that distinguishes $f$ and $f'$.

Extra materials for the proof sketch are in Appendix A.2.

## 7. Conclusion

We prove guarantees of length generalization for various function classes in an idealized setting. We formalize the framework of non-asymptotic length generalization, which requires a computable upper bound for length complexity. We show the Minimum-Complexity Interpolator learning algorithm achieves optimal length complexity. We show that whether a function class admits non-asymptotic length generalization is equivalent to the decidability of its language equivalence problem, which implies that there is no computable upper bound for the length complexity of CFGs. On the positive side, we show that the length complexity of DFAs is $2n-2$ where $n$ is the number of states of the ground-truth automaton. We show that the length complexity of 1-layer C-RASP functions is $O(T^2)$ when the ground-truth function has precision $T$, and that the length complexity of 2-layer C-RASP functions is $O(T^{O(K)})$ when the ground-truth function has precision $T$ and $K$ heads.

It is open whether the proof techniques can be extended to 3-Layer C-RASP programs, or to C-RASP programs whose layers contain bias terms. It is open how to formalize a weaker notion of partial length generalization which does not entail length generalization to arbitrary length inputs.

## Acknowledgements

This work was supported by NSF IIS 2211780 and the Stanford HAI–Google Cloud Credits Program. TC acknowledges funding from an NSF Graduate Research Fellowship. ZL acknowledges funding from an OpenAI Superalignment Grant.

## Impact Statement

This paper presents work whose goal is to advance the field of Machine Learning. There are many potential societal consequences of our work, none which we feel must be specifically highlighted here.

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

## Organization of Appendices.

Section A contains supplementary material to the main paper, particularly the proof sketch. Section B contains a discussion of Gold (1967)'s asymptotic notions of learnability and the proof of Proposition B.2. Section C contains the proofs of Theorem 3.2, Lemma 3.4, and Lemma C.6. Section D contains the proof of Propositions 2.7 and 3.5. Section E.1 contains the proof of Theorem 5.5. Section E.2 contains the full proof of Theorem 6.1, which uses Definitions and Lemmas defined in the later sections E.3 to E.6. Section E.3 contains proofs of Lemmas relating to completeness of basis schema. Section E.4 contains proofs of Lemmas relating to lower bounding the margin of the average-of-vertices of a polytope with low-precision faces. Section E.5 contains proofs of Lemmas for approximating a continuous test-function with a discrete test-function. Section E.6 contains auxiliary Lemmas.

## A. Missing Parts from Main Paper

Here, we put some supporting content which were referred to in the main paper, but omitted due to space constraints.

### A.1. List of Allowed C-RASP Operations in (Yang & Chiang, 2024), in the Context of Definition 5.1.

*Table 2.* Boolean- and count-valued operations allowed in a C-RASP program.

| **Boolean-Valued Operations** | | **Count-Valued Operations** | |
|---|---|---|---|
| **Initial** | $h_j^{(i)} := \mathbb{1}[x_j = a]$ for $a \in \{0,1\}$ | **Partial Sum** | $h_j^{(i)} := \mathrm{ps}(h^{(i')})_j$ |
| **Boolean** | $h_j^{(i)} := \neg h_j^{(i')}$ 
 $h_j^{(i)} := h_j^{(i')} \wedge h_j^{(i'')}$ | **Conditional** | $h_j^{(i)} := h_j^{(i')} \,?\, h_j^{(i'')} : h_j^{(i''')}$ |
| **Sign** | $h_j^{(i)} := \mathbb{1}[h_j^{(i')} > 0]$ | **Addition** | $h_j^{(i)} := h_j^{(i')} + h_j^{(i'')}$ |
| **Constant** | $h_j^{(i)} := 1$ | **Subtraction** | $h_j^{(i)} := h_j^{(i')} - h_j^{(i'')}$ |
| | | **Min/Max** | $h_j^{(i)} := \min(h_j^{(i')}, h_j^{(i'')})$ 
 $h_j^{(i)} := \max(h_j^{(i')}, h_j^{(i'')})$ |
| | | **Constant** | $h_j^{(i)} := 1$ |

### A.2. Supporting Material for the Proof Sketch in Section 6.

**Proof that Theorem 5.6 follows from Theorem 6.1.** *Proof of Theorem 5.6.* Every function $f$ in C-RASP$^{2,K,T}$ has at most $K$ heads and $T$ precision, and so $\mathcal{C}(f) \leq T^K =: \alpha_*$. Thus, Theorem 6.1 implies that inputs of length at most $N_{\mathcal{A}_{\mathrm{mci}}}(\alpha_*) \leq O(\alpha_*^{O(1)}) = O(T^{O(K)})$ suffices to learn C-RASP$^{2,K,T}$. ∎

**Proof that Lemma 6.2 follows from Lemma 6.3.** *Proof of Lemma 6.2.* By Theorem 3.2, it suffices to show that for any two un-equal functions $f, f' \in$ C-RASP$^2$ such that $T(f)^{K(f)} \leq \alpha$ and $T(f')^{K(f')} \leq \alpha$, there is a string of length at most $O(\alpha^{O(\log^2 \alpha)})$ which distinguishes them. If $T(f)^{K(f)} \leq \alpha$ and $T(f')^{K(f')} \leq \alpha$, then $T(f), T(f')$ are upper bounded as $T(f), T(f') \leq \alpha$, maximized when $K(f), K(f') = 1$. Meanwhile, $K(f), K(f')$ are upper bounded as $K(f), K(f') \leq \log \alpha$, maximized when $K(f) = T(f)^2, K(f') = T(f')^2$. Thus, $f, f' \in$ C-RASP$^{2,\log \alpha, \alpha}$. Applying Lemma 6.3 implies that there is a string of length $O(\alpha^{O(\log^2 \alpha)})$ distinguishing $f$ and $f'$. ∎

**Formal Summary of Each Step of Proof Plan in Section 6.2.** Below, we formalize the conclusions drawn from each step of the proof plan in the proof sketch of Lemma 6.3. Putting these Lemmas together yields Lemma 6.3.

**Lemma A.1** (Analysis of Step 1). *If $f \neq f' \in$ C-RASP$^{2,K,T}$, then there exists a continuous test-function $\mathcal{Y}_0$ such that $(B_i(\mathcal{Y}_0))_{i \in [k]} \in (H_1^+ \cap H_2^+) \cup (H_1^- \cap H_2^-)$. WLOG, suppose $(B_i(\mathcal{Y}_0))_{i \in [k]} \in H_1^+ \cap H_2^+$.*

**Lemma A.2** (Analysis of Step 2). *Given continuous test-function $\mathcal{Y}_0$ where $(B_i(\mathcal{Y}_0))_{i \in [k]} \in H_1^+ \cap H_2^+$, then there exists a continuous test function $\mathcal{Y}_2$ of a schema $Y$ of $M \leq k^2$ segments, where lengths of the segments of $\mathcal{Y}_2$ is given by*

$(n_i^{(c)})_{i \in [M]} \in [0,1]^M$. *In addition, the $M$-dimensional point $(n_i^{(c)})_{i \in [M]}$ has margin at least $\gamma \geq \frac{1}{3M^2} \frac{1}{(\mathrm{poly}(K,T)\sqrt{M})^M}$ to the faces of polytope $P := \mathrm{cl}(A^{(M)}(Y) \cap H_1^{(M)} \cap H_2^{(M)})$.*

**Lemma A.3** (Analysis of Step 3). *Given any point $c := (n_i^{(c)})_{i \in [M]} \in [0,1]^M$ which are segment lengths of a particular test-function of schema $Y$, suppose $c$ has margin at least $\gamma > 0$ to faces of polytope $P = \mathrm{cl}(A^{(M)}(Y) \cap H_1^{(M)} \cap H_2^{(M)})$. Then, one can perturb the coordinates of $c$ to get a point $c_* \in [0,1]^M$ such that (1) the margin of $c_*$ to the faces of $P$ is at least $\frac{\gamma}{2}$ and (2) the coordinates of $c_*$ have precision at most $\frac{\mathrm{poly}(K,T)}{\gamma}$.*

**Lemma A.4** (Analysis of Step 4). *If there exists $\gamma > 0, c_* \in P$ such that the margin of $c_*$ to the faces of $P$ is at least $\frac{\gamma}{2}$ and the coordinates of $c_*$ have precision at most $\frac{\mathrm{poly}(K,T)}{\gamma}$, then there exists discrete test-function $\mathcal{X}'$ such that $(B_i(\mathcal{X}'))_{i \in [k]} \in H_1^+ \cap H_2^+$ and the length of $\mathcal{X}'$ is at most $O(\frac{\mathrm{poly}(K,T)}{\gamma} \cdot T^k)$.*

With $M \leq k^2$, $k \leq 2K$, and $K \leq T^2$, we conclude that if $f \neq f'$, then there exists a string $x', |x'| \leq O(T^{O(K^2)})$ such that $f(x') \neq f'(x')$.

**Details of Key Steps of Proof Plan of Section 6.2.** We now elaborate on key steps of the proof plan. We will focus on Step 2, because Step 1 just applies Lemma E.30, Step 3 just applies Lemma E.29, and Step 4 applies Lemma 6.9 and analyzes how large $n$ needs to be so that the final discrete test-function distinguishes $f, f'$.

**Details of Step 2.** First, we characterize $\mathbb{A}(\{s_i\}_{i \in [k]})$ as the union of a finite number of polytopes.

**Lemma A.5** (Characterization of $\mathbb{A}(\{s_i\}_{i \in [k]})$, combination of Corollary E.14 and Lemma E.16). *For any $(k,T)$-configuration $\{s_i\}_{i \in [k]}$, there are a finite number of $k$-dimensional convex polytopes $\{A_j\}_{j \in [N_k]}$, $N_k < \infty$, such that*

$$\mathbb{A}(\{s_i\}_{i \in [k]}) = \bigcup_{j \in [N_k]} A_j$$

Figure 9 pictorally depicts the completeness Lemma for $k = 2$, which shows that $\mathbb{A}(\{s_i\}_{i \in [2]})$ is the union of two triangles.

The way Lemma A.5 is proved is by showing that for any continuous test-function $\mathcal{Y}$, there exists another continuous test-function of one of a finite number of possible schemas which induces the same activations $\{B_i(\mathcal{Y})\}_{i \in [k]}$. We call these $N_k$ schemas "basis" schema; they are a "basis" in the sense that any continuous test-function is equivalent to a continuous test-function which follows some basis schema. Figure 8 depicts a basis schema. We defer the details to Appendix E.3. The main take-away is that each polytope $A_j \subset [0,1]^k$ corresponds to a schema $Y_j$, and $A_j$ is the set of activations which can be induced by a test-function of schema $Y_j$.

$$\forall j \in [N_k], A_j := \{(B_1(\mathcal{Y}), \ldots, B_k(\mathcal{Y})) : \mathcal{Y} \text{ valid test-function of schema } Y_j\} \subset [0,1]^k$$

Thus, given $\mathcal{Y}_0$ from Step 1 such that, WLOG, $(B_i(\mathcal{Y}_0))_{i \in [k]} \in H_1^+ \cap H_2^+$, then there is an equivalent test-function $\mathcal{Y}_1$ of one of the basis schema, $Y \in \{Y_j\}_{j \in [N_k]}$, such that $(B_i(\mathcal{Y}_1))_{i \in [k]} = (B_i(\mathcal{Y}_0))_{i \in [k]} \in H_1^+ \cap H_2^+$. Denote the set $A(Y)$ as the particular $A_j$ which corresponds to schema $Y$.

$$A(Y) := \{(B_1(\mathcal{Y}), \ldots, B_k(\mathcal{Y})) : \mathcal{Y} \text{ valid test-function of schema } Y\} \subset [0,1]^k$$

Now, suppose that schema $Y$ consists of $M$ segments, whose lengths we denote $(n_1, n_2, \ldots, n_M) \in [0,1]^M$ with $\sum_{i \in [M]} n_i = 1$. Note that not all settings of $(n_1, n_2, \ldots, n_M)$ are valid, because there are linear constraints which must be met by $(n_1, n_2, \ldots, n_M)$, which are described in Lemma E.6. Define $A^{(M)}(Y) \subset [0,1]^M$ as the set of all valid settings of $(n_1, n_2, \ldots, n_M)$, per Lemma E.6. $A^{(M)}(Y)$ is an $(M-1)$-dimensional polytope, whose faces are parameterized by linear inequalities whose coefficients are at most $\mathrm{poly}(K,T)$ in magnitude (i.e. $\mathrm{poly}(K,T)$-precision).

$$A^{(M)}(Y) := \{(n_1, \ldots, n_M) : \text{ valid segment lengths of schema } Y \text{ and } \sum_{i \in [M]} n_i = 1\} \subset [0,1]^M$$

Further, define $H_1^{(M)}$ and $H_2^{(M)}$ as the analogous halfspaces to $H_1^+$ and $H_2^+$ but in the space of segment lengths as follows. There exists a linear map $L : \mathbb{R}^M \to \mathbb{R}^k$ which maps points in $A^{(M)}(Y)$ to points in $A(Y)$. $L \in \{0,1\}^{k \times M}$ is such that $L_{ij} = 1 \iff$ segment $j$ in schema $Y$ lies above line $i$ (that is, for every x in the domain of segment $j$, the y-value of the segment at x is at least $s_i \cdot$ x) and hence contributes to the $i$th activation $B_i(\mathcal{Y})$ of any test-function $\mathcal{Y}$ of schema $Y$. Using $L$, we can rewrite the inequalities which characterize $H_1, H_2$ in terms of $(n_1, \ldots, n_M)$.

$$H_1^{(M)} := \{(n_1, \ldots, n_M) : \sum_{i=1}^{K} \lambda_i B_{\text{ord}(1,i)} > z\} = \{(n_1, \ldots, n_M) : \sum_{i=1}^{K} \lambda_i \sum_{j \text{ s.t. } L_{\text{ord}(1,i),j}=1} n_j > z\}$$

$$H_2^{(M)} := \{(n_1, \ldots, n_M) : \sum_{i=1}^{K} \lambda_i' B_{\text{ord}(2,i)} < z'\} = \{(n_1, \ldots, n_M) : \sum_{i=1}^{K} \lambda_i' \sum_{j \text{ s.t. } L_{\text{ord}(2,i),j}=1} n_j < z'\}$$

With $P := \text{cl}(A^{(M)}(Y) \cap H_1^{(M)} \cap H_2^{(M)})$, we use the fact that $(B_i(\mathcal{Y}_1)_{i \in [k]} \in H_1^+ \cap H_2^+$ to deduce that $A^{(M)}(Y) \cap H_1^{(M)} \cap H_2^{(M)} \neq \emptyset$. Let $V$ be the set of vertices of $P$. This next Lemma is for lower-bounding the margin of point $c = \frac{1}{|V|} \sum_{v \in V} v \in [0,1]^M$ on the faces of $P$, when the latter is non-empty.

**Lemma A.6** (Margin lower bound, slightly informal version of Lemma E.18). *Consider a nonempty d-dimensional polytope $P \subset \mathbb{R}^d$ with vertices $V$ and $N$ faces. Suppose the faces of $P$ are each defined by a linear inequality over variables $\{x_i\}_{i \in [d]}$, with integer coefficients of magnitude at most $p_{\text{face}}$, where points on the face satisfy the linear inequality with equality. For $j \in [N]$, define $L_j$ as the linear inequality for the jth face of $P$. Then, for any $j \in [N]$, for any vertex $x \in V$ which does not lie on the jth face of $P$, we have the following lower bound on the margin of $x$ on the jth face of $P$.*

$$L_j(x) \gtrapprox \frac{1}{(p_{\text{face}} \sqrt{d})^d}$$

With $d = M$ and $p_{\text{face}} = \text{poly}(K, T)$ and using the linearity of the margin, it follows that the margin of $c = \frac{1}{|V|} \sum_{v \in V} v$ on any face of $P$ is at least $\frac{1}{|V|} \frac{1}{(\text{poly}(K,T)\sqrt{M})^M}$. $c$ is a point in $[0,1]^M$, and it is a valid setting of segment lengths for a test-function of schema $Y$, since $c$ has positive margin to the faces of $P$, so it is contained in $P \subset A^{(M)}(Y)$. The activations of the test-function with segment lengths given by $c$ will be contained in $H_1^+ \cap H_2^+$ since $c \in P \subset H_1^{(M)} \cap H_2^{(M)}$. This concludes Step 2.

## B. Asymptotic Length Generalization, Adapted from (Gold, 1967)

Define the following asymptotic notion of learnability.

**Definition B.1** (Length Generalization in the Limit, adapted from Gold (1967)). *A function class $\mathcal{F} \subseteq \mathcal{F}^{\mathcal{R}}$ admits length generalization in the limit w.r.t. encoding system $\mathcal{R}$ if there exists a learning algorithm $\mathcal{A}$ such that for all $f^* \in \mathcal{F}$, there exists a natural number $N$ such that for all $N' \geq N$, $\mathcal{A}$ length-generalizably learns $f^*$ at input length $N'$.*

The above definition of length generalization in the limit is a special case of the so-called *identification in the limit* in the informant model in (Gold, 1967). The major difference is that (Gold, 1967) requires the function class $\mathcal{F}$ to be learnable in arbitrary order of data presentation, while in our case, we are only interested in a particular order of data presentation, namely the order of increasing input length.

Proposition B.2 is a result about which function classes are learnable in the sense of Definition B.1. We provide a proof here, which is similar to that of Theorem I.4 of (Gold, 1967)

**Proposition B.2** (Adapted from Theorem I.4 of Gold (1967)). *For all encoding systems $\mathcal{R}$, the function class $\mathcal{F}^{\mathcal{R}}$ admits length generalization in the limit, and thus so does any function class $\mathcal{F} \subseteq \mathcal{F}^{\mathcal{R}}$.*

*Proof.* Consider a learning algorithm which enumerates functions in $\mathcal{F}^{\mathcal{R}}$. It maintains a current hypothesis, which is initialized to the first element of the enumeration. Given an input training set, the learning algorithm checks whether the current hypothesis interpolates the training data. If it does not, then the current hypothesis is updated to the next function in the enumeration until the current hypothesis interpolates the training data, at which point the current hypothesis is outputted.

First, because $\mathcal{R}$ maps $\{0, 1\}^*$ to $\mathcal{F}^{\mathcal{R}}$, then we can enumerate $\mathcal{F}^{\mathcal{R}}$ simply by enumerating all strings $p \in \{0, 1\}^*$, say in the order of increasing binary-representations of natural numbers, and then computing $\mathcal{R}(p)$ for each string $p$. The computation of $\mathcal{R}(p)$ will always halt by Definition 2.1. Every function in $\mathcal{F}^{\mathcal{R}}$ appears at some point in this enumeration by the definition of $\mathcal{F}^{\mathcal{R}}$. Suppose that we denote this enumeration by $\mathcal{E} = \{f_0, f_1, f_2, \ldots\}$, with $\mathcal{E}_c := \{f_0, f_1, f_2, \ldots, f_c\}$ being the first $c + 1$ elements of the enumeration, for any $c \in \mathbb{N}$.

Second, because the output of $\mathcal{R}$ is always a computable function which halts on every input, the learning algorithm described in the first paragraph is computable.

By Theorem 3.2 with $\mathcal{C}(\cdot)$ being the mapping from a binary string to the integer it represents in binary (so that each finite representation is mapped to its order in the aforementioned enumeration), it suffices to show that for any $c \in \mathbb{N}$, $N(\mathcal{E}_c) := \min\{n : \forall f \neq f' \in \mathcal{E}_c, \exists x \in \{0, 1\}^{\leq n} \text{ s.t. } f(x) \neq f'(x)\} < \infty$. This follows directly from the fact that $|\mathcal{E}_c| = c < \infty$ and that for any $f \neq f' \in \mathcal{E}_c$, there exists some string $x, |x| < \infty$ where $f(x) \neq f'(x)$. For each pair of unequal functions in $\mathcal{E}_c$, pick the shortest such string $x$ which distinguishes them, and take $N(\mathcal{E}_c)$ to be the maximum of the lengths of these distinguishers. This must be finite. ∎

## C. Proofs on Equivalence of Different Non-Asymptotic Length Generalization Definitions

### C.1. Properties of $\mathcal{A}_{\mathsf{mci}}$

We will prove that $\mathcal{A}_{\mathsf{mci}}$ is optimal with respect to the quantity $N_{\mathcal{A}_{\mathsf{mci}}}(c)$. Note we will drop the superscripts for $\mathcal{F}, \mathcal{C}, \mathcal{R}$ from the notation for convenience.

**Theorem 3.2** (Optimality of Minimum-Complexity Interpolator). *Given any encoding system $\mathcal{R}$ and complexity measure $\mathcal{C}$, for all $c \in \mathbb{N}$, it holds that $N_{\mathcal{A}_{\mathsf{mci}}^{\mathcal{R},\mathcal{C}}}^{\mathcal{R},\mathcal{C}}(c) = \min_{\mathcal{A}} N_{\mathcal{A}}^{\mathcal{R}}(c) = N(\mathcal{F}_c^{\mathcal{R}})$. As a consequence, the following three statements are equivalent:*

- *Function class $\mathcal{F}^{\mathcal{R}}$ admits non-asymptotic length generalization;*
- *Function class $\mathcal{F}^{\mathcal{R}}$ admits non-asymptotic length generalization, via learning algorithm $\mathcal{A}_{\mathsf{mci}}^{\mathcal{R},\mathcal{C}}$;*
- *For all $c \in \mathbb{N}$, length complexity of $\mathcal{F}_c^{\mathcal{R}}$, $N(\mathcal{F}_c^{\mathcal{R}})$, is computably bounded in $c$.*

*Proof of Theorem 3.2.* For any $c \in \mathbb{N}$, we have

$$N(\mathcal{F}_c^{\mathcal{R}}) := \min\{n \in \mathbb{N} : \forall f \neq f' \in \mathcal{F} \text{ with } \mathcal{C}^{\mathcal{R}}(f'), \mathcal{C}^{\mathcal{R}}(f) \leq c,$$
$$\text{there exists } x \in \{0, 1\}^{\leq n} \text{ s.t. } f(x) \neq f'(x)\}$$

We will show separately that $N_{\mathcal{A}_{\mathsf{mci}}}(c) \geq \min_{\mathcal{A}} N_{\mathcal{A}}(c) \geq N(\mathcal{F}_c^{\mathcal{R}})$ and $N_{\mathcal{A}_{\mathsf{mci}}}(c) \leq N(\mathcal{F}_c^{\mathcal{R}})$.

(1). $\min_{\mathcal{A}} N_{\mathcal{A}}(c) \geq N(\mathcal{F}_c^{\mathcal{R}})$ follows from the fact that for any $N < N(\mathcal{F}_c^{\mathcal{R}})$, there exists $f \neq f'$ with $\mathcal{C}^{\mathcal{R}}(f'), \mathcal{C}^{\mathcal{R}}(f) \leq c$ that agree on all inputs $x \in \{0, 1\}^{\leq N}$. Given training set $\{(x, f(x)) : \forall x \in \{0, 1\}^{\leq N}\} = \{(x, f'(x)) : \forall x \in \{0, 1\}^{\leq N}\}$, any algorithm $\mathcal{A}$ which outputs an encoding of a function which is not equal to $f$ will be wrong in the case the ground truth is $f$. Meanwhile, any algorithm $\mathcal{A}$ which outputs an encoding of a function which is not equal to $f'$ will be wrong in the case the ground truth is $f'$. Thus, $N_{\mathcal{A}}(f) > N$ or $N_{\mathcal{A}}(f') > N$, so $N_{\mathcal{A}}(c) > N$.

(2). $N_{\mathcal{A}_{\mathsf{mci}}}(c) \leq N(\mathcal{F}_c^{\mathcal{R}})$. For $f$ with $\mathcal{C}^{\mathcal{R}}(f) \leq c$, suppose that on input of $\{(x, f(x)) : \forall x \in \{0, 1\}^{\leq N(\mathcal{F}_c^{\mathcal{R}})}\}$, $\mathcal{A}_{\mathsf{mci}}$ outputs $p'$, where $\mathcal{R}(p') \neq f$. This implies that $\mathcal{C}(\mathcal{R}(p')) \leq \mathcal{C}(p') \leq \mathcal{C}^{\mathcal{R}}(f) \leq c$ and that $\mathcal{R}(p')$ is consistent with $\{(x, f(x)) : \forall x \in \{0, 1\}^{\leq N(\mathcal{F}_c^{\mathcal{R}})}\}$. This is a contradiction of the definition of $N(\mathcal{F}_c^{\mathcal{R}})$, which says that all pairs of unequal functions $f, f'$ with $\mathcal{C}^{\mathcal{R}}(f), \mathcal{C}^{\mathcal{R}}(f') \leq c$ can be distinguished from each other by some input of length at most $N(\mathcal{F}_c^{\mathcal{R}})$. ∎

Denote $p \equiv p'$ if $\mathcal{R}(p) = \mathcal{R}(p')$. The following Lemma is also useful for showing that the same bounds on the length complexity hold whether we think of $\mathcal{A}_{\mathsf{mci}}$ as operating over finite descriptions $p \in \{0, 1\}^*$ (which is the actual implementation) or as operating over functions $f \in \mathcal{F}$ (which is a more abstract way to think of $\mathcal{A}_{\mathsf{mci}}$).

**Lemma C.1.** *Given $(\mathcal{F}, \mathcal{R}, \mathcal{C})$, with $\mathcal{C}^{\mathcal{R}}(f) := \min_{p \in \{0,1\}^*, \mathcal{R}(p)=f} \mathcal{C}(p)$, then for all $c \in \mathbb{N}$:*

$$\min\{n : \forall f \neq f' \in \mathcal{F}, \mathcal{C}^{\mathcal{R}}(f), \mathcal{C}^{\mathcal{R}}(f') \leq c, \exists x, |x| \leq n, f(x) \neq f'(x)\}$$
$$= \min\{n : \forall p \not\equiv p' \in \{0,1\}^*, \mathcal{C}(p), \mathcal{C}(p') \leq c, \exists x, |x| \leq n, \mathcal{R}(p)(x) \neq \mathcal{R}(p')(x)\}$$

*Proof.* Every function $f \in \mathcal{F}$ with $\mathcal{C}^{\mathcal{R}}(f) \leq c$ has some finite description $p$ with $\mathcal{R}(p) = f$ and $\mathcal{C}(p) \leq c$. No function $f \in \mathcal{F}$ with $\mathcal{C}^{\mathcal{R}}(f) > c$ has any finite description $p$ with $\mathcal{R}(p) = f$ and $\mathcal{C}(p) \leq c$. ∎

We also have the following result which says that $\mathcal{A}_{\mathsf{mci}}$ is computable if $\mathcal{C}$ satisfies Assumption 2.4.

**Lemma 3.3.** *Under Assumption 2.4, Algorithm 1 is computable and thus a valid learning algorithm.*

*Proof of Lemma 3.3.* By Assumption 2.4, we use TM $E$ to enumerate all programs $p \in \{0,1\}^*$ in non-decreasing order of $\mathcal{C}(p)$ and at each iteration check if $\mathcal{R}(p)$ is consistent with the training data. If it is, we output $p$ and stop. Otherwise, we continue to the next program. Both the enumeration and checking-consistency procedures are computable, as assumed by Assumption 2.4 and Definition 2.1. ∎

## C.2. Equivalence of Non-Asymptotic Length Generalization with Decidability of Language Equivalence Problem

Now, we will prove that Definition 2.6 is equivalent to a few other definitions. First, recall the definition of non-asymptotic length generalization and the language equivalence problem.

**Definition 2.6** (Non-Asymptotic Length Generalization). A function class $\mathcal{F} \subseteq \mathcal{F}^{\mathcal{R}}$ admits *non-asymptotic length generalization* w.r.t. encoding system $\mathcal{R}$ and complexity measure $\mathcal{C}$ if there exists a learning algorithm $\mathcal{A}$ and a computable function $\hat{N}_{\mathcal{A}}^{\mathcal{R}, \mathcal{F}} : \mathbb{N} \to \mathbb{N}$ such that for all $f^* \in \mathcal{F}$ and for all $N' \geq \hat{N}_{\mathcal{A}}^{\mathcal{R}, \mathcal{F}}(\mathcal{C}^{\mathcal{R}}(f^*))$, $\mathcal{A}$ length-generalizably learns $f^*$ at input length $N'$.

**Definition C.2** (Language Equivalence Problem). The *Language Equivalence Problem* for encoding system $\mathcal{R}$ is the computational problem where given any $p, q \in \{0,1\}^*$, determine whether $\mathcal{R}(p) = \mathcal{R}(q)$.

We now prove the following equivalences.

**Lemma 3.4.** *For any encoding system $\mathcal{R}$ and complexity measure $\mathcal{C}$ satisfying Assumption 2.4, the Language Equivalence problem for $\mathcal{R}$ is decidable if and only if length complexity of $\mathcal{F}_c^{\mathcal{R}}$, $N(\mathcal{F}_c^{\mathcal{R}})$, is computably bounded in $c$. Thus it is also equivalent to the property that $\mathcal{F}^{\mathcal{R}}$ admits non-asymptotic length generalization.*

*Proof.* Suppose TM $E$ enumerates elements in $\{0,1\}^*$ in non-decreasing order of their complexity according to $\mathcal{C}$.

$N(\mathcal{F}_c^{\mathcal{R}})$ **computably bounded in** $c \implies$ **Language equivalence problem for** $\mathcal{R}$ **decidable.**

Suppose there is a computable procedure $F$ that, for any $c$, receives as input $c$ and outputs an upper bound on $N(\mathcal{F}_c^{\mathcal{R}})$. We will describe an algorithm that is given any two finite representations, $p, q \in \{0,1\}^*$, and uses $F$ to determine if $\mathcal{R}(p) = \mathcal{R}(q)$.

Given $p, q \in \{0,1\}^*$, compute $c = \max(\mathcal{C}(p), \mathcal{C}(q))$ and generate the training dataset $D_{F(c)}(\mathcal{R}(q))$. Now, check whether $\forall (x, \mathcal{R}(q)(x)) \in D_{F(c)}(\mathcal{R}(q))$, that $\mathcal{R}(p)(x) = \mathcal{R}(q)(x)$. If this is the case, return "equivalent." Otherwise, return "non-equivalent".

To argue correctness, if there is some $(x, \mathcal{R}(q)(x)) \in D_{F(c)}(\mathcal{R}(q))$ where $\mathcal{R}(p)(x) \neq \mathcal{R}(q)(x)$, then clearly these two functions are not equal. If $\forall (x, \mathcal{R}(q)(x)) \in D_{F(c)}(\mathcal{R}(q)), \mathcal{R}(p)(x) = \mathcal{R}(q)(x)$, then since $\mathcal{C}(\mathcal{R}(p)) \leq \mathcal{C}(p) \leq c$ and $\mathcal{C}(\mathcal{R}(q)) \leq \mathcal{C}(q) \leq c$, then it must be that $\mathcal{R}(p) = \mathcal{R}(q)$, or else $F(c)$ is not an upper bound of $N(\mathcal{F}_c^{\mathcal{R}})$.

**Language equivalence problem for** $\mathcal{R}$ **decidable** $\implies N(\mathcal{F}_c^{\mathcal{R}})$ **computably bounded in** $c$. Suppose TM $M$ solves the Language Equivalence Problem for $\mathcal{R}$. By Lemma C.1, it suffices to show that the following quantity is computably bounded in $c \in \mathbb{N}$.

$$\min\{n : \forall p \not\equiv p' \in \{0,1\}^*, \mathcal{C}(p), \mathcal{C}(p') \leq c, \exists x, |x| \leq n, \mathcal{R}(p)(x) \neq \mathcal{R}(p')(x)\}$$

Algorithm 2 computes an upper bound of this quantity.

---

**Algorithm 2** Computation of $N(\mathcal{F}_c^{\mathcal{R}})$

---

**Require:** Integer $c \in \mathbb{N}$
**Ensure:** $N(\mathcal{F}_c^{\mathcal{R}})$
1: $N \leftarrow 0$
2: **for all** pairs of programs $(p, q)$ (enumerated by $E$) with $\mathcal{C}(p), \mathcal{C}(q) \leq c$ **do**
3:     **if** $M(p, q) =$ "non-equivalent" **then**
4:         **for all** strings $x \in \{0, 1\}^*$ with non-decreasing length **do**
5:             **if** $\mathcal{R}(p)(x) \neq \mathcal{R}(q)(x)$ **then**
6:                 $N \leftarrow \max(N, |x|)$
7:                 **break**
8:             **end if**
9:         **end for**
10:     **end if**
11: **end for**
12: **return** $N$

---

Algorithm 2 always terminates, since the for-loop in line 4 only is performed on $p, q \in \{0, 1\}^*$ which are not equivalent, which are distinguished by a finite string. It returns the smallest length required to distinguish every two non-equivalent $p, q \in \{0, 1\}^*$.

Finally, it follows from Theorem 3.2 that the language equivalence problem for $\mathcal{R}$ being decidable is equivalent to non-asymptotic length generalization of $\mathcal{F}^{\mathcal{R}}$ w.r.t. $\mathcal{R}, \mathcal{C}$ when $\mathcal{C}$ satisfies Assumption 2.4. ∎

### C.3. Connection to Finite Identification, (Gold, 1967)

Our notion of non-asymptotic length generalization is related to Gold's notion of Finite Identification (Gold, 1967), which is restated as follows in the context of length generalization.

**Definition C.3** (Finite Length Generalization ("Identification", (Gold, 1967))). A function class $\mathcal{F} \subseteq \mathcal{F}^{\mathcal{R}}$ admits *Finite Length Generalization* w.r.t. encoding system $\mathcal{R}$ if there exists a Turing Machine (TM) $\mathcal{A}$, where for any $f_* \in \mathcal{F}$, on input of a training set $D_N(f_*)$ for some $N \in \mathbb{N}$, $\mathcal{A}$ satisfies:

1. $\mathcal{A}$ can only output either "pass" or some program $\hat{p} \in \{0, 1\}^*$. $\mathcal{A}$ outputs $\hat{p}$ at least for one $N \in \mathbb{N}$.

2. Whenever $\mathcal{A}$ outputs some program $\hat{p} \in \{0, 1\}^*$, it must be correct in the sense that $\mathcal{R}(\hat{p}) = f_*$.

It is clear that Finite Length Generalization implies Length Generalization in the limit — one can replace "pass" by arbitrary program $\hat{p}$ and sticks to the same (correct) output $\hat{p}$ once $\mathcal{A}$ outputs any program $\hat{p}$. Intuitively, a function class admits Finite Length Generalization if there exists a learning algorithm which can perfectly length generalize at some finite input length (and where the learning algorithm knows the length at which it length generalizes), rather than only generalizing in the limit. Thus, Finite Length Generalization is a very desirable property. However, it is in general too good to be true. We give a characterization of Finite Length Generalization in Lemma C.4. As a consequence of Lemma C.4, functions classes as simple as the set of all languages that are finite do not admit Finite Length Generalization.

**Lemma C.4** (Characterization of Finite Length Generalization). *A function class $\mathcal{F}^{\mathcal{R}}$ admits Finite Length Generalization w.r.t. encoding system $\mathcal{R}$ if and only if for any $f_* \in \mathcal{F}^{\mathcal{R}}$, there exists a natural number $N$ such that $f^*$ is the only function that is consistent with the training set $D_N(f_*)$.*

The proof of Lemma C.4 is straightforward and omitted.

Since Finite Length Generalization is in general too restrictive, we relax the definition to allow the learning algorithm to output a program $\hat{p}$ with some information on the complexity measure of ground truth $f_*$, namely some $c \geq \mathcal{C}^{\mathcal{R}}(f_*)$. This leads to the definition of Finite Length Generalization with Complexity Information in Definition C.5. Interestingly, Definition C.5 is equivalent to non-asymptotic length generalization (Definition C.3).

**Definition C.5** (Finite Length Generalization with Complexity Information). A function class $\mathcal{F} \subseteq \mathcal{F}^{\mathcal{R}}$ admits *Finite Length Generalization* w.r.t. encoding system $\mathcal{R}$ and complexity measure $\mathcal{C}$ if there exists a Turing Machine (TM) $\mathcal{A}$, which for any $f_* \in \mathcal{F}$, on input of a training set $D_N(f_*)$ for some $N \in \mathbb{N}$ and **a natural number** $c \geq \mathcal{C}^{\mathcal{R}}(f_*)$, it satisfies that:

1. $\mathcal{A}$ can only output either "pass" or some program $\hat{p} \in \{0,1\}^*$. $\mathcal{A}$ outputs $\hat{p}$ at least for one $N \in \mathbb{N}$.

2. Whenever $\mathcal{A}$ outputs some program $\hat{p} \in \{0,1\}^*$, it must be correct in the sense that $\mathcal{R}(\hat{p}) = f_*$.

**Lemma C.6** (Equivalence of Finite Length Generalization Definitions). *For any encoding system $\mathcal{R}$ and complexity measure $\mathcal{C}$ satisfying Assumption 2.4, function class $\mathcal{F}^{\mathcal{R}}$ admits Finite Length Generalization with Complexity Information w.r.t. $\mathcal{R}$ and $\mathcal{C}$ if and only if length complexity of $\mathcal{F}_c^{\mathcal{R}}$, $N(\mathcal{F}_c^{\mathcal{R}})$, is computably bounded in c. Thus it is also equivalent to $\mathcal{F}^{\mathcal{R}}$ admits non-asymptotic length generalization.*

*Proof of Lemma C.6.* Suppose TM $E$ enumerates elements in $\{0,1\}^*$ in non-decreasing order of their complexity according to $\mathcal{C}$.

$N(\mathcal{F}_c^{\mathcal{R}})$ **computably bounded in** $c$ $\implies$ $\mathcal{F}^{\mathcal{R}}$ **admits Finite Length Generalization with Complexity Information w.r.t.** $\mathcal{R}$ **and** $\mathcal{C}$**.** Let $F : \mathbb{N} \to \mathbb{N}$ be a computable upper bound on $N(\mathcal{F}_c^{\mathcal{R}})$. Let $\mathcal{A}$ be given by Algorithm 3, on input $D_N(f_*)$ and $c \geq \mathcal{C}^{\mathcal{R}}(f_*) \in \mathbb{N}$.

---

**Algorithm 3** Learning Algorithm $\mathcal{A}$

---

**Require:** Dataset $D_N(f_*)$; $c \in \mathbb{N}$ with $c \geq \mathcal{C}^{\mathcal{R}}(f_*)$
**Ensure:** Either "**pass**" or an program $\hat{p}$
 1: **if** $N < F(c)$ **then**
 2:     **return** "**pass**"
 3: **else**
 4:     $\hat{p} \leftarrow \mathcal{A}_{\mathrm{mci}}\big(D_N(f_*)\big)$
 5:     **return** $\hat{p}$
 6: **end if**

---

First, $\mathcal{A}$ only ever returns "pass" or some $\hat{p} \in \{0,1\}^*$ since $\mathcal{A}_{\mathrm{mci}}$ only ever returns elements of $\{0,1\}^*$. Second, by Theorem 3.2, for $c \geq \mathcal{C}^{\mathcal{R}}(f_*)$ and $\forall N \geq F(c)$, then $\mathcal{R}(\mathcal{A}_{\mathrm{mci}}(D_N(f_*))) = f_*$. Thus, whenever $\mathcal{A}$ returns an element $\hat{p} \in \{0,1\}^*$, it is such that $\mathcal{R}(\hat{p}) = f_*$, and there is at least one $N$ where $\mathcal{R}(\mathcal{A}_{\mathrm{mci}}(D_N(f_*))) = f_*$.

$\mathcal{F}^{\mathcal{R}}$ **admits Finite Length Generalization with Complexity Information w.r.t.** $\mathcal{R}$ **and** $\mathcal{C}$ $\implies$ $N(\mathcal{F}_c^{\mathcal{R}})$ **computably bounded in** $c$**.** We need to prove that there exists some computable $F$ which upper bounds $N(\mathcal{F}_c^{\mathcal{R}})$. Suppose $\mathcal{A}$ satisfies Definition C.5. Let $F$ be given by Algorithm 4.

---

**Algorithm 4** Algorithm for $F$

---

**Require:** Integer $c \in \mathbb{N}$
**Ensure:** $N_{\max}$
 1: $N_{\max} \leftarrow 0$
 2: **for all** $p \in \{0,1\}^*$ with $\mathcal{C}(p) \leq c$ (Enumerate with $E$) **do**
 3:     $N \leftarrow 0$
 4:     **while** $\mathcal{A}(D_N(\mathcal{R}(p)), \mathcal{C}(p)) = $ "pass" **do**
 5:         $N \leftarrow N + 1$
 6:     **end while**
 7:     $N_{\max} \leftarrow \max(N_{\max}, N)$
 8: **end for**
 9: **return** $N_{\max}$

---

Algorithm 4 is computable, due to the guarantee of $\mathcal{A}$ that for any $f_*$, given $c \geq \mathcal{C}^{\mathcal{R}}(f_*)$, there exists some $N$ where $\mathcal{R}(\mathcal{A}(D_N(f_*))) = f_*$, which ensures termination of Algorithm 4.

Regarding correctness, if there exists some $c_* \in \mathbb{N}$ where $F(c_*) < \min\{n : \forall p \not\equiv p' \in \{0,1\}^*, \mathcal{C}(p), \mathcal{C}(p') \leq c_*, \exists x, |x| \leq n, \mathcal{R}(p)(x) \neq \mathcal{R}(p')(x)\}$, then there must exist two $p, q \in \{0,1\}^*$ with $\mathcal{C}(p), \mathcal{C}(q) \leq c_*$, which are not equivalent, but which agree on all inputs of length at most $F(c_*)$. WLOG, suppose that $\mathcal{C}(p) \leq \mathcal{C}(q)$.

There exists $N_q \leq F(c_*)$ where $\mathcal{A}(D_{N_q}(\mathcal{R}(q)), \mathcal{C}(q)) \neq$ "pass". In particular, $\mathcal{A}(D_{N_q}(\mathcal{R}(q)), \mathcal{C}(q))$ must return a $\hat{p}$ which is equivalent to $q$, where $q \not\equiv p$. On the other hand, since $N_q \leq F(c_*)$, we have $D_{N_q}(\mathcal{R}(q)) = D_{N_q}(\mathcal{R}(p))$. Thus, we

have shown that $\mathcal{A}(D_{N_q}(\mathcal{R}(p)), \mathcal{C}(q)) \neq$ "pass" and is a finite representation which is not equivalent to the ground-truth $p$. Since $\mathcal{C}(p) \leq \mathcal{C}(q)$, this yields a contradiction with the fact that whenever $\mathcal{A}$ is given an upper bound on the ground-truth complexity and $\mathcal{A}$ does not return "pass", it must return a finite description $\hat{p}$ which is equivalent to the ground-truth. $\blacksquare$

Figure C.3 sums up all these equivalences.

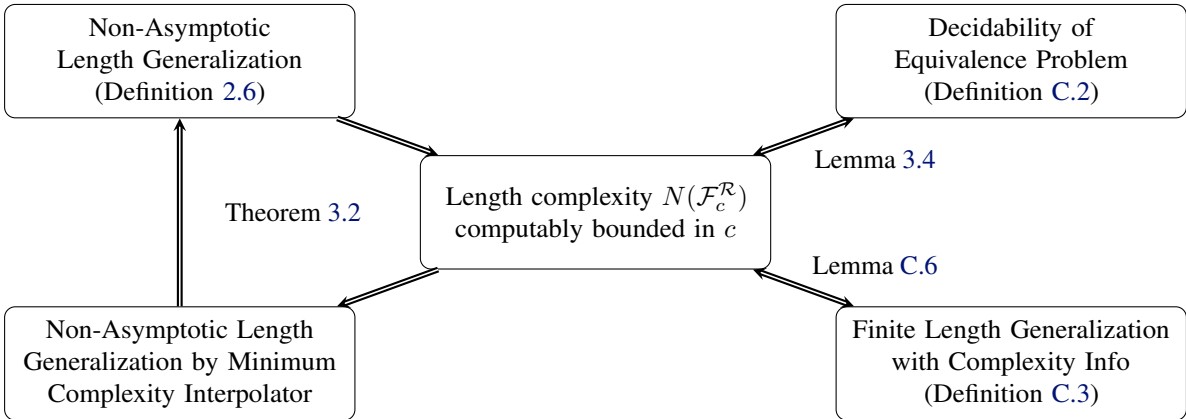

*Figure 1.* Summary of equivalence results between different characterizations of length generalization in Section 3 under some mild simplicity assumptions on the complexity measure $\mathcal{C}$ (Assumption 2.4). Each arrow represents an implication proven by the corresponding theorem.

# D. Proofs on Length Generalization for DFAs and CFGs.

**Proposition 2.7** (Non-Asymptotic Length Generalization for DFAs). *Let $\mathcal{R}_{DFA}$ be the DFA encoding system defined in Definition 2.2, and let $\mathcal{C}_{DFA}$ be the number of states in DFA. Regular languages $\mathcal{F}^{\mathcal{R}}$ admits non-asymptotic length generalization w.r.t. encoding system $\mathcal{R}_{DFA}$ and complexity measure $\mathcal{C}_{DFA}$. More specifically, there exists a learning algorithm $\mathcal{A}$ such that $N_{\mathcal{A}}^{\mathcal{R}_{DFA}}(c) \leq 2c - 2$ for all $c \in \mathbb{N}$.*

In the following proof, we essentially describe and analyze the State Minimization Algorithm (Hopcroft & Ullman, 1979). Before proving the proposition, we will need a few concepts and Lemma D.1.

Suppose there is some ground-truth minimal DFA, $D$, of $n$ states, with language $L := L(D)$. Define $\epsilon$ as the empty string, of length 0. For two strings $u, v$, denote $uv$ as their concatenation. For any $x \in \{0, 1\}^*$, denote $f_*(x) = 1 \iff x \in L$. Denote $Q$ as the set of $n$ states for $D$ and $\delta(\cdot, \cdot) : Q \times \{0, 1\} \to Q$ as the state transition function for $D$, with start state $q_0$. Let $F \subset Q$ be the set of accepting states. We say that states $q, q' \in Q$ are distinguished by string $v \in \{0, 1\}^*$ if $\delta(q, v) \in F$ and $\delta(q', v) \notin F$, or if $\delta(q, v) \notin F$ and $\delta(q', v) \in F$.

In general, applying a string from the start state $q_0$ in the DFA will cause the DFA to end up in some state $q \in Q$, and an identification can be made between the input string and the state it causes the DFA to end up in. Thus, to learn the DFA, we would like to find a mapping from $\{0, 1\}^*$ to $[n]$, which tells us which strings correspond to which states in the ground-truth DFA. To do this, we claim it is sufficient to consider the equivalence class given by the sets $\{E(u)\}_{u \in \{0,1\}^{\leq n}}$, where:

$$\forall u \in \{0, 1\}^{\leq n}, E(u) := \{v \in \{0, 1\}^{\leq n-2} : uv \in L\}$$

If for $u, u' \in \{0, 1\}^{\leq n}, E(u) = E(u')$, then we will claim that these two strings correspond to the same state. Otherwise, $u, u'$ correspond to different states. This is what is meant by an equivalence class over $\{E(u)\}_{u \in \{0,1\}^{\leq n}}$.

Once a learning algorithm can make this identification between strings and the states they correspond to, then the learning algorithm can infer the transition function of the DFA by considering, for each $u \in \{0, 1\}^{\leq n-1}$, the states $E(u), E(u0)$, and $E(u1)$, via the sets $\{E(u)\}_{u \in \{0,1\}^{\leq n-1}}$ and $\{E(u)\}_{u \in \{0,1\}^{\leq n}}$. This gives the transition function. $E(\epsilon)$ corresponds to the start state. Strings $u$ corresponding to accept states will be such that $\epsilon \in E(u)$. Note that we just need to consider

$\{E(u)\}_{u \in \{0,1\}^{\leq n}}$ instead of $\{E(u)\}_{u \in \{\epsilon\} \cup \{0,1\}^*}$ to characterize the states of $D$, since any $u$ of length larger than $n$ will cause the DFA to reach some state that is also reached by a string of length at most $n$.

We claim that proving the following property about our sets $E(u)$ suffices to prove that there is a learning algorithm which identifies the ground-truth DFA.

**Lemma D.1.** *For any minimal DFA $D$ with $n \geq 2$ states and transition function $\delta$, then with $E(u)$ defined as above for each $u \in \{0,1\}^{\leq n}$, we have $\forall u, u' \in \{0,1\}^{\leq n}, \delta(q_0, u) = \delta(q_0, u') \iff E(u) = E(u')$.*

First, we will show how to prove Proposition 2.7 with this Lemma.

*Proof of Proposition 2.7.* Suppose a DFA learning algorithm is given inputs of length $2n - 2$. For $n \geq 2$, Lemma D.1 implies that the learning algorithm can identify the states of $D$ by constructing the sets $\{E(u)\}_{u \in \{0,1\}^{\leq n}}$, using its training data $\{(x, f_*(x)) : |x| \leq 2n - 2\}$. Each state is identified with an equivalence class of $\{E(u)\}_{u \in \{0,1\}^{\leq n-1}}$, since each state in an $n$ state DFA can be reached by a string of length at most $n - 1$. The state transitions can be identified from $\{E(u)\}_{u \in \{0,1\}^{\leq n}}$ by considering, for each $u \in \{0,1\}^{\leq n-1}$, the states $E(u)$, $E(u0)$, and $E(u1)$. $E(\epsilon)$ corresponds to the start state. Strings $u$ corresponding to accept states will be exactly those such that $\epsilon \in E(u)$. In the case that the learning algorithm only receives inputs of length at most $0$ (i.e. it only receives $(\epsilon, f_*(\epsilon))$) or $1$, it will simply output the DFA with language $\emptyset$ if the labels of all inputs received are $0$ and it will output the DFA with language $\{0,1\}^*$ if the labels of all inputs received are $1$. This handles the case where $n = 1$.

The learning algorithm above identifies the ground-truth DFA, requiring inputs of length at most $2n - 2$ when the ground-truth DFA has $n$ states. By Theorem 3.2, the Minimum Complexity Interpolator $\mathcal{A}_{\mathsf{mci}}$ will also require inputs of length at most $2n - 2$ to identify the ground-truth DFA. This proves the Proposition. ∎

Now we prove Lemma D.1.

*Proof of Lemma D.1.* Showing $\delta(q_0, u) = \delta(q_0, u') \implies E(u) = E(u')$ is easy, since if $\delta(q_0, u) = \delta(q_0, u')$, then the states reached by $u, u'$ from $q_0$ are the same, so no string of any length can distinguish them $\delta(q_0, u), \delta(q_0, u')$, so $E(u) = E(u')$.

We now show $\delta(q_0, u) \neq \delta(q_0, u') \implies E(u) \neq E(u')$. We are given that the ground truth DFA has $n$ states and is the minimal DFA for its language, so that there are no DFAs of fewer states with the same language. Given this, we claim that there are exactly $n$ equivalence classes among $\{E(u)\}_{u \in \{0,1\}^{\leq n}}$. Each equivalence class must correspond to a unique state, so each of $n$ unique states can be identified with a unique $E(u)$, finishing the proof of Lemma D.1.

We will now show that there are exactly $n$ equivalence classes among $\{E(u)\}_{u \in \{0,1\}^{\leq n}}$. For $0 \leq i \leq n - 2$, $u \in \{0,1\}^{\leq n}$, define $E_i(u) := \{v \in \{0,1\}^{\leq i} : uv \in L\}$. To do this, we claim that for any $i < n - 2$, if the number of equivalence classes in $\{E_i(u)\}_{u \in \{0,1\}^{\leq n}}$ is less than $n$, then there must be some $u, u' \in \{0,1\}^{\leq n}$ and $v_{i+1} \in \{0,1\}^{i+1}$ such that

$$E_i(u) = E_i(u')$$
$$\text{and } f_*(uv_{i+1}) \neq f_*(u'v_{i+1})$$

Suppose this were not the case. That is, suppose that for some $i < n - 2$, (1) the number of equivalence classes in $\{E_i(u)\}_{u \in \{0,1\}^{\leq n}}$ is less than $n$. (2) Yet, $\forall u, u' \in \{0,1\}^{\leq n}, \forall v_{i+1} \in \{0,1\}^{i+1}, E_i(u) = E_i(u') \implies f_*(uv_{i+1}) = f_*(u'v_{i+1})$. We want to show that together, (1) and (2) contradict the minimality of the ground-truth DFA.

We claim that, by induction, (2) implies that for every $k \geq i$, for every $u, u' \in \{0,1\}^{\leq n}$ and $v_{k+1} \in \{0,1\}^{k+1}, E_k(u) = E_k(u') \implies f_*(uv_{k+1}) = f_*(u'v_{k+1})$. This is true for base case $k = i$ by (2).

For the inductive step, suppose the claim is true for $k > i$. Suppose $u, u' \in \{0,1\}^{\leq n}$ are such that $E_k(u) = E_k(u')$. Then $\forall v_k \in \{0,1\}^{\leq k}, f_*(uv_k) = f_*(u'v_k)$. For any such string $v_k$ with length $k$, we can break $v_k$ into a prefix of $k - i$ bits and a suffix of $i$ bits. Thus,

$$\forall v_{k-i} \in \{0,1\}^{k-i}, \forall v_i \in \{0,1\}^i, f_*(uv_{k-i}v_i) = f_*(u'v_{k-i}v_i)$$

Each string $uv_{k-i}$ and $u'v_{k-i}$ will correspond to some state reachable by some strings $u'', u''' \in \{0,1\}^{\leq n}$, in the sense

that $\delta(q_0, uv_{k-i}) = \delta(q_0, u'')$ and $\delta(q_0, u'v_{k-i}) = \delta(q_0, u''')$, as each state in the DFA can be reached by a string of length at most $n-1$. Thus, $E_i(u'') = E_i(u''')$, so that by (2), $\forall v_{i+1} \in \{0,1\}^{i+1}, f_*(u''v_{i+1}) = f_*(u'''v_{i+1})$. Since $\delta(q_0, uv_{k-i}) = \delta(q_0, u'')$ and $\delta(q_0, u'v_{k-i}) = \delta(q_0, u''')$ imply that for any such $v_{i+1}$, $f_*(uv_{k-i}v_{i+1}) = f_*(u''v_{i+1})$ and $f_*(u'v_{k-i}v_{i+1}) = f_*(u'''v_{i+1})$, then $\forall v_{i+1} \in \{0,1\}^{i+1}, f_*(uv_{k-i}v_{i+1}) = f_*(u'v_{k-i}v_{i+1})$. In short,

$$\forall v_{k-i} \in \{0,1\}^{k-i}, \forall v_{i+1} \in \{0,1\}^{i+1}, f_*(uv_{k-i}v_{i+1}) = f_*(u'v_{k-i}v_{i+1})$$

Thus, $\forall v_{k+1} \in \{0,1\}^{k+1}, f_*(uv_{k+1}) = f_*(u'v_{k+1})$, completing the induction. We have proved the claim for all $k \geq i$.

However, this will lead to a contradiction if we take $k \to \infty$, since applying the guarantee for all $k \geq i$, we have that

$$\forall u, u' \in \{0,1\}^{\leq n}, E_i(u) = E_i(u') \implies E_\infty(u) = E_\infty(u')$$

Where $E_\infty(u) := \{v \in \{\epsilon\} \cup \{0,1\}^* : uv \in L\}$. We know that any two distinct states in a minimal DFA will be distinguished by some finite string (See Theorem 4.24 of (Hopcroft & Ullman, 1979)). Thus, for any $u, u' \in \{0,1\}^{\leq n}$,

$$E_i(u) = E_i(u')$$
$$\implies E_\infty(u) = E_\infty(u')$$
$$\implies \delta(q_0, u) = \delta(q_0, u')$$

So $n = \#(\text{Equivalence classes in } \{E_\infty(u)\}_{u \in \{0,1\}^{\leq n}}) = \#(\text{Equivalence classes in } \{E_i(u)\}_{u \in \{0,1\}^{\leq n}}) < n$, where the last inequality is due to (1). This is a contradiction.

Thus, while the number of equivalence classes in $\{E_i(u)\}_{u \in \{0,1\}^{\leq n}}$ is less than $n$, there must be some $u, u' \in \{0,1\}^{\leq n}$ and $v_{i+1} \in \{0,1\}^{i+1}$ such that

$$E_i(u) = E_i(u')$$
$$\text{and } f_*(uv_{i+1}) \neq f_*(u'v_{i+1})$$

This implies that the number of equivalence classes in $\{E_{i+1}(u)\}_{u \in \{0,1\}^{\leq n}}$ is at least 1 greater than the number of equivalence classes in $\{E_i(u)\}_{u \in \{0,1\}^{\leq n}}$ while the latter is less than $n$.

Finally, $\{E_0(u)\}_{u \in \{0,1\}^{\leq n}}$ has 2 equivalence classes for any non-trivial DFA that whose language isn't $\emptyset$ or $\{0,1\}^*$, and any $n \geq 2$ state minimal DFA must be non-trivial, as the minimal trivial DFAs have at most one state. The two equivalence classes of $\{E_0(u)\}_{u \in \{0,1\}^{\leq n}}$ are given by the states that are accepting and those that are not. For all $i < n-2$, the number of equivalence classes in $\{E_{i+1}(u)\}_{u \in \{0,1\}^{\leq n}}$ grows by at least 1 from that of $\{E_i(u)\}_{u \in \{0,1\}^{\leq n}}$. We can have no more equivalence classes than $n$, and the $i$ where $\{E_i(u)\}_{u \in \{0,1\}^{\leq n}}$ has $n$ equivalence classes will be at most $n-2$. This implies Lemma D.1 is true if we let $E(u) = E_{n-2}(u), \forall u \in \{0,1\}^{\leq n}$. ∎

Now, we will prove an impossibility result for linear CFGs.

**Proposition 3.5** ((Linear) CFGs only admit length generalization in the limit). *Recall $\mathcal{R}_{\text{L-CFG}}$ is the encoding system for Linear CFGs defined in Definition 2.3 and $\mathcal{C}_{\text{CFG}}(\langle G \rangle)$ is the complexity measure that maps CFG $G = (N, T, P, S = \{0,1\})$ to $|N| + |T| + |P|$. Then for any learning algorithm $\mathcal{A}$, the length complexity, $N_{\mathcal{A}}^{\mathcal{R}_{\text{L-CFG}}}$, is not computably bounded. That is, Linear CFGs do not admit non-asymptotic length generalization (w.r.t. standard CFG encoding system $\mathcal{R}_{\text{CFG}}$), and neither does the set of all CFGs.*

*Proof of Proposition 3.5.* This follows directly from Lemma 3.4 and Theorem 3.2, with $\mathcal{R}$ being an encoding system which maps string encodings of linear CFGs to the corresponding language, $\mathcal{F} = \mathcal{F}^{\mathcal{R}}$ being the languages recognized by linear CFGs, and $\mathcal{C}$ being the complexity measure described in Definition 2.3. Because $\mathcal{R}_{\text{L-CFG}}$ satisfies Assumption 2.4, Lemma 3.4 and Theorem 3.2 imply that encoding system $\mathcal{F}^{\mathcal{R}}$ admits non-asymptotic length generalization w.r.t. $\mathcal{R}, \mathcal{C}$ iff the language equivalence problem for $\mathcal{R}$ is decidable. However, by (Baker & Book, 1974), the latter is undecidable. ∎

# E. Proofs for Length Generalization of C-RASP

## E.1. Proof of Theorem 5.5

**Theorem 5.5** (C-RASP[1] Length Generalization). *Let $\mathcal{F} = $ C-RASP[1] and $\mathcal{C}(f) = \max(|a|, |b|, |d|)$, defined in Definition 5.3. Then $\forall T \in \mathbb{N}$, we have $N_{\mathcal{A}_{mci}}(T) \leq O(T^2)$. That is, the Minimum-Complexity Interpolator, with complexity $\mathcal{C}$ and function class $\mathcal{F}$, can length generalize given inputs of length $O(T^2)$ when the ground-truth has complexity $T$.*

*Proof.* With integer $T$ and integers $a, b, d \in [-T, T]$, $a > 0$, recall C-RASP$^{1,T}$ is the set of functions of the form:

$$f_{a,b,d}(x) = \mathbb{1}[a \cdot \mathtt{ps}(x) - b \cdot n - d > 0]$$

By Theorem 3.2, it is sufficient to show that for any tuples of integers $(a, b, d), (a', b', d') \in [-T, T]^3$ with $a, a' > 0$, if there exists an $x \in \{0, 1\}^*$ with $f_{a,b,d}(x) \neq f_{a',b',d'}(x)$, then there exists an $x' \in \{0, 1\}^*$ with $|x'| \leq O(T^2)$ such that $f_{a,b,d}(x') \neq f_{a',b',d'}(x')$.

Suppose there is a string $x$ that distinguishes $f_{a,b,d}$ and $f_{a',b',d'}$. We can rewrite $f$ and $f'$ as follows.

$$f(x) := f_{a,b,d}(x) = \mathbb{1}[\mathtt{ps}(x) - \frac{b}{a} \cdot n - \frac{d}{a} > 0]$$
$$f'(x) := f_{a',b',d'}(x) = \mathbb{1}[\mathtt{ps}(x) - \frac{b'}{a'} \cdot n - \frac{d'}{a'} > 0]$$

Since $f, f'$ differ on $x$, either $\frac{b}{a} \neq \frac{b'}{a'}$; or $\frac{b}{a} = \frac{b'}{a'}$ but $\frac{d}{a} \neq \frac{d'}{a'}$.

**Case 1:** $\frac{b}{a} \neq \frac{b'}{a'}$

There are 3 categories for the value of a slope $\frac{b}{a}$ (resp. $\frac{b'}{a'}$).

- Type (i): $\frac{b}{a} \geq 1$ (resp. $\frac{b'}{a'} \geq 1$)

- Type (ii): $\frac{b}{a} \leq 0$ (resp. $\frac{b'}{a'} \leq 0$)

- Type (iii): $\frac{b}{a} \in (0, 1)$ (resp. $\frac{b'}{a'} \in (0, 1)$)

Below we will consider how to distinguish two functions $f, f'$ whose slopes $\frac{b}{a}, \frac{b'}{a'}$ are in each of the categories.

- $\frac{b}{a}$ is type (iii); $\frac{b'}{a'}$ is either type (iii), (ii), or (i)

  For any two 2D lines $y = \frac{b}{a}x + \frac{d}{a}$ and $y = \frac{b'}{a'}x + \frac{d'}{a'}$, there is a lattice point of $x-$coordinate $x_* := |aa'|(2\max(|\frac{d}{a}|, |\frac{d'}{a'}|) + 1)$ that lies strictly above the line with smaller slope and un-strictly below the line with greater slope. This follows from the fact that when $\frac{b'}{a'} \neq \frac{b}{a}$, then the smallest that $|\frac{b'}{a'} - \frac{b}{a}|$ can be is $\frac{1}{|aa'|}$. Thus, $|\frac{b'}{a'} - \frac{b}{a}| \cdot |aa'|(2\max(|\frac{d}{a}|, |\frac{d'}{a'}|) + 1) \geq 2\max(|\frac{d}{a}|, |\frac{d'}{a'}|) + 1$ so that $|\frac{b}{a}x_* + \frac{d}{a} - (\frac{b'}{a'}x_* + \frac{d'}{a'})| \geq 1$. Because the vertical gap between the two lines at horizontal coordinate $x_*$ is at least 1, there is a lattice point $(x_*, y_*) \in \mathbb{Z}^2$ with $y_* \in (\min(\frac{b}{a}x_* + \frac{d}{a}, \frac{b'}{a'}x_* + \frac{d'}{a'}), \max(\frac{b}{a}x_* + \frac{d}{a}, \frac{b'}{a'}x_* + \frac{d'}{a'})]$. In fact, for any $x \geq x_*$, one can find such a lattice point with horizontal coordinate $x$, since the gap between the two lines will continue to grow as $x$ increases for $x \geq x_*$, and so the gap will always be at least 1.

  We want to pick such a lattice point $(\tilde{x}, \tilde{y})$ subject to four constraints:

  1. $\tilde{y} \leq \max(\frac{b}{a}\tilde{x} + \frac{d}{a}, \frac{b'}{a'}\tilde{x} + \frac{d'}{a'})$
  2. $\tilde{y} > \min(\frac{b}{a}\tilde{x} + \frac{d}{a}, \frac{b'}{a'}\tilde{x} + \frac{d'}{a'})$
  3. $\tilde{y} \leq \tilde{x}$
  4. $\tilde{y} \geq 0$

Since $\frac{b}{a} \in (0,1)$, then it suffices to find such a lattice point either between $\mathsf{y} = \frac{b}{a}\mathsf{x} + \frac{d}{a}$ and $\mathsf{y} = \mathsf{x}$, between $\mathsf{y} = \frac{b}{a}\mathsf{x} + \frac{d}{a}$ and $\mathsf{y} = 0$, or between $\mathsf{y} = \frac{b}{a}\mathsf{x} + \frac{d}{a}$ and $\mathsf{y} = \frac{b'}{a'}\mathsf{x} + \frac{d'}{a'}$. For each of the 3 cases, any lattice point between the two lines specified in that case will satisfy all four constraints.

By the argument in the previous paragraph, for each of the three cases, we can construct such a lattice point with horizontal coordinate $\mathsf{x}_*$, which will be at most the following, across the three cases.

$$\tilde{\mathsf{x}} \leq \max\left[|aa'|(2\max(|\frac{d}{a}|, |\frac{d'}{a'}|) + 1), |a \cdot 1|(2\max(|\frac{d}{a}|, 0) + 1), |a \cdot 1|(2\max(|\frac{d}{a}|, 0) + 1)\right]$$
$$\leq 2\max(|da'|, |ad'|) + aa'$$
$$\leq 3T^2$$

Denote the vertical coordinate of this lattice point as $0 \leq \tilde{\mathsf{y}} \leq \tilde{\mathsf{x}}$. Finally, any string $x' \in \{0,1\}^*$ with $|x'| = \tilde{\mathsf{x}}$, consisting of $\tilde{\mathsf{y}} \geq 0$ ones and $\tilde{\mathsf{x}} - \tilde{\mathsf{y}} \geq 0$ zeros will distinguish $f$ and $f'$. This string will have length at most $3T^2$.

- $\frac{b}{a}$ and $\frac{b'}{a'}$ are type (ii)

  Since $\frac{b}{a} \neq \frac{b'}{a'}$, say WLOG that $\frac{b}{a} < 0$.

  If $\frac{b'}{a'} < 0$, then after $\mathsf{x}$ coordinate at most $T + 1$, then lines $\{\mathsf{y} = \frac{b}{a}\mathsf{x} + \frac{d}{a}, \mathsf{y} = \frac{b'}{a'}\mathsf{x} + \frac{d'}{a'}\}$ go below $\mathsf{y} = 0$ and no lattice point of nonnegative y coordinate can be below one but above the other (i.e. both $f, f'$ become the all-ones function on strings of length at least $T + 1$). Thus, any distinguisher of $f, f'$ (in particular, the $x$ presumed in the beginning of the proof of this Theorem) must have length at most $T$, and we take $x' = x$.

  If $\frac{b'}{a'} = 0$, then in the case $\frac{d}{a} < 0$, the same argument implies that $|x| \leq T$.

  In the case that $\frac{d}{a} \geq 0$, then a lattice point which lies between the two lines is $(T + 1, 0)$. Thus, the string $x' = 0_{T+1}$ distinguishes $f, f'$ and has length $T + 1$.

- $\frac{b}{a}$ is type (ii) and $\frac{b'}{a'}$ is type (i)

  The analysis of this case is similar to the Types (iii) versus (iii), (ii), (i) case. One can take the line of slope $\geq 1$ and modify just its slope to be between $(0, 1)$, apply the analysis in the Types (iii) versus (iii), (ii), (i) case to attain a lattice point which lies below the original line of slope $\geq 1$ and above the line of slope $\leq 0$. Such a lattice point corresponds to a string of length at most $2T$ which distinguishes $f, f'$.

- $\frac{b}{a}$ and $\frac{b'}{a'}$ are type (i)

  Analogous to the argument about Type (ii) versus (ii). There will be a string of length at most $T + 1$ which distinguishes $f, f'$.

**Case 2:** $\frac{b}{a} = \frac{b'}{a'}$ **but** $\frac{d}{a} \neq \frac{d'}{a'}$ If the slopes are at least 1 or at most 0, an analogous argument as Case 1, Type (ii) versus (ii) or (i) versus (i) will suffice, where it must be the case that the string $x$ which was presumed to distinguish $f, f'$ has length at most $T + 1$.

Otherwise, if both slopes are in $(0, 1)$, then we can take a lattice point $(\mathsf{x}_*, \mathsf{y}_*)$ which lies on the one of larger intercept. Such lattice points occur periodically with spacing $a \leq T$, and we need only take one such lattice point where $\mathsf{x}_* \geq \mathsf{y}_* \geq 0$. We can find one such lattice point where $\mathsf{x}_* \leq 3T$. ∎

**Notation and Conventions.** For the remaining sections, we will use lower-case $x$ to denote bit-strings in $\{0,1\}^*$.

We define $[k] := \{1, 2, \ldots, k - 1, k\}$ as the first $k$ positive integers.

We will use symbol $f$ to denote functions in the relevant hypothesis class, like C-RASP or DFAs. These will be mappings from $\{0,1\}^*$ to $\{0,1\}$. We will sometimes refer to functions in C-RASP[1] and C-RASP[2] as programs, which is synonymous with "functions." We will say that two functions are equal if they agree on all strings $\{0,1\}^*$, and that they are unequal if they are distinguished by at least one string $x \in \{0,1\}^*$. We will call such an $x$ a distinguisher of the two functions. To be clear, the word function here has a distinct meaning and type from test-functions.

We will use $\mathcal{X}$ to denote discrete test-functions and $\mathcal{Y}$ to denote continuous test-functions. When we say "test-function" without specifying whether it is discrete or continuous, assume we mean a continuous test-function. We will use $(B_i(\mathcal{Y}))_{i\in[k]}$ (or $(B_i)_{i\in[k]}$ when clear from context) to represent the activations induced by a continuous test-function (see Definition 6.6). We will use symbol $Y$ to denote a schema of continuous test-functions (see Definition E.5). When we talk about a 2D coordinate system, in the context of test-functions, we will use symbol x to denote the horizontal axis of this 2D coordinate system and y to denote the vertical axis. To be clear, x is distinct from the symbol $x$, which we use to denote bit-strings.

Regarding geometric objects, denote $\mathrm{cl}(S)$ as the closure of a set $S$. Denote $S^c$ as the complement of set $S$. For a convex set $S$, the affine hull of $S$ is the set of linear combinations of points in $S$. Denote $\dim(S)$ as the dimension of the affine hull of $S$. Denote the interior of a convex set $S$ as $\mathrm{int}(S)$ and the relative interior of convex set $S$ as $\mathrm{ri}(S)$, which is the set of points in $S$ such that there is some non-zero radius such that the Euclidean ball centered at that point with that radius, intersected with the affine hull of $S$, is contained in $S$. Finally, sometimes we will talk about a geometrical set in $\mathbb{R}^k$ and its "analog" in $\mathbb{R}^M$, where $M > k$ (what "analog" means, we won't go into here). Notationally, if we use symbol $A \in \mathbb{R}^k$ for the set in $\mathbb{R}^k$, we will use symbol $A^{(M)}$ to denote a set that's analogous to $A$, but in $\mathbb{R}^M$.

A halfspace of $\mathbb{R}^d$ is a subset of $\mathbb{R}^d$ which satisfy a linear inequality over the $d$ coordinates of $\mathbb{R}^d$. A polytope is the intersection of a finite number of halfspaces. The polytopes we will be working with will be restricted to $[0,1]^d$ for some dimension $d$. The faces of a polytope $P \subset [0,1]^d$ refer to the $d-1$ dimensional polytopes which form the boundary $P$. Each face is associated with a linear inequality (halfspace) which defines the face in the sense that points on the face satisfy the linear inequality with equality. A $d$-dimensional simplex is a $d$-dimensional polytope with $d+1$ vertices and $d+1$ faces.

### E.2. Proof of Theorem 6.1.

We reiterate the definition of C-RASP$^2$ here.

**Definition 5.4** (C-RASP$^2$). With integers $T$ and $1 \le K \le T^2$, let C-RASP$^{2,K,T}$ be the set of programs of the following form. Each program $f$ has parameters $0 < z \le T$, $\forall i \in [K]$, $a^{(i)}, b^{(i)}, \lambda_i \in \{-T, \ldots, T\}$, with $a^{(i)} > 0$. We require that for all $i \in [K]$, $\frac{b^{(i)}}{a^{(i)}} \in (0,1)$ and is distinct from $\frac{b^{(i')}}{a^{(i')}}$ for $i' \ne i$. We also require $\sum_{i\in[K]} \lambda_i > z$.

For any $n > 0$, on input $x \in \{0,1\}^n$, the first layer computes the values of $K$ heads, $\{h^{(i)}\}_{i\in[K]}$, on the $n$ prefixes of $x$: $\{\{x_1\}, \{x_1, x_2\}, \ldots, \{x_1, \ldots, x_n\}\}$ as follows: $\forall j \in [n], \forall i \in [K], h_j^{(i)} = \mathbb{1}[\mathtt{ps}(x)_j > \frac{b^{(i)}}{a^{(i)}} j]$. Subscript $j$ indicates the value of a quantity on the $j$th prefix of $x$.

The second layer computes the output, which is the $n$th bit of the final sequence: $f(x) = \mathbb{1}[\sum_{i\in[K]} \lambda_i \mathtt{ps}(h^{(i)})_n > z \cdot n]$.

Then, C-RASP$^2 = \bigcup_{1\le T, 1\le K\le T^2}$ C-RASP$^{2,K,T}$. Given a function $f \in$ C-RASP$^2$, let $K(f)$ be the number of heads, $h^{(i)}$, in the first layer of $f$ and let $T(f) := \max(\max_{i\in[K(f)]} |a^{(i)}|, \max_{i\in[K(f)]} |b^{(i)}|, \max_{i\in[K(f)]} |\lambda_i|, |z|)$. The complexity measure is $\mathcal{C}(f) = T(f)^{K(f)}$, the precision of the function's parameters to the power of the number of heads.

We will prove the following length generalization guarantee.

**Theorem 6.1.** Let $\mathcal{F} =$ C-RASP$^2$ and $\mathcal{C}(f) = T(f)^{K(f)}$, as in Definition 5.4. Then $\forall \alpha \in \mathbb{N}$, $N_{\mathcal{A}_{mci}}(\alpha) \le O(\alpha^{O(1)})$.

Note that Theorem 6.1 is actually stronger than a result which says that we can learn C-RASP$^{2,K,T}$ with length $O(T^{O(K)})$. This is because for a fixed $T$ and $K$, C-RASP$^{2,K,T}$ only contains functions of precision at most $T$ and at most $K$ heads, whereas Theorem 6.1 also provides length generalization guarantees for C-RASP$^2$ functions $f$ with $T(f) > T, K(f) < K$ or $T(f) < T, K(f) > K$ such that $T(f)^{K(f)} \le T^K$.

*Proof of Theorem 6.1.* By Theorem 3.2, to upper bound $N_{\mathcal{A}_{mci}}(\alpha)$, it suffices to bound:

$$\min\{n \in \mathbb{N} : \forall f \ne f' \in \text{C-RASP}^2 \text{ with } \mathcal{C}(f), \mathcal{C}(f') \le \alpha, \text{ exists } x \in \{0,1\}^{\le n} \text{ s.t. } f(x) \ne f'(x)\}$$

Suppose $f, f' \in$ C-RASP$^2$ are not equal and differ on $x_0 \in \{0,1\}^*$. For any $n > 0$, suppose $f$ and $f'$ have the following 2-layer form on an arbitrary input $x \in \{0,1\}^n$.

$$\forall j \in [n], \forall i \in [K], \quad h_j^{(i)}(x) = \mathbb{1}[\mathbf{ps}(x)_j > \frac{b^{(i)}}{a^{(i)}}j]$$

$$f(x) = \mathbb{1}[\sum_{i \in [K]} \lambda_i \mathbf{ps}(h^{(i)}(x))_n > z \cdot n]$$

$$\forall j \in [n], \forall i \in [K'], \quad (h^{(i)})_j'(x) = \mathbb{1}[\mathbf{ps}(x)_j > \frac{(b^{(i)})'}{(a^{(i)})'}j]$$

$$f'(x) = \mathbb{1}[\sum_{i \in [K']} \lambda_i' \mathbf{ps}((h^{(i)})'(x))_n > z' \cdot n]$$

Where $f$ has $K$ first-layer heads and integer parameters of precision $T$ and $f'$ has $K'$ first-layer heads and integer parameters of precision $T'$, where $C(f) = T^K \leq \alpha$ and $C(f') = (T')^{K'} \leq \alpha$, and $K \leq T^2$ and $K' \leq (T')^2$. WLOG, suppose that $T \geq T'$. We will find a short string $x_* \in \{0,1\}^{O(\alpha^{O(1)})}$ such that $f(x_*) \neq f'(x_*)$.

Suppose the set of unique slopes $R := \{\frac{b^{(i)}}{a^{(i)}}\}_{i \in [K]} \cup \{\frac{(b^{(i)})'}{(a^{(i)})'}\}_{i \in [K']} \subset (0,1)$ between the first layer of $f$ and $f'$ has size $k$, where $\max(K, K') \leq k \leq K + K'$. We will denote $R = \{s_j\}_{j \in [k]} \subset (0,1)$, where $s_1$ is the largest slope and $s_k$ is the smallest slope, and the slopes are sorted in descending order so that $s_1 > \ldots > s_k$. Let $\mathrm{ord}(1, i) : [K] \to [k]$ be the index within $R$ of the $i$th slope of $f$, $\frac{b^{(i)}}{a^{(i)}}$. Let $\mathrm{ord}(2, i) : [K'] \to [k]$ be the index within $R$ of the $i$th slope of $f'$, $\frac{(b^{(i)})'}{(a^{(i)})'}$. In the following exposition, we will refer to "line $i$" as the homogeneous, 2D line $\mathsf{y} = s_i \mathsf{x}$, with slope $s_i, i \in [k]$.

Definition 5.4 requires that $z, z' > 0$, and that $\sum_{i \in [K]} \lambda_i > z$ and $\sum_{i \in [K']} \lambda_i' > z'$. Since $f$ differs from $f'$ on discrete test-function given by $x_0$, then the non-trivial Lemma E.37 implies that there exists a continuous test-function $\mathcal{Y}_1$ that induces activations $(B_1(\mathcal{Y}_1), \ldots, B_k(\mathcal{Y}_1))$ which satisfies either Case I or Case II.

$$\text{Case I: } \sum_{i \in [K]} \lambda_i B_{\mathrm{ord}(1,i)}(\mathcal{Y}_1) > z \text{ and } \sum_{i \in [K']} \lambda_i' B_{\mathrm{ord}(2,i)}(\mathcal{Y}_1) < z'$$

$$\text{Case II: } \sum_{i \in [K]} \lambda_i B_{\mathrm{ord}(1,i)}(\mathcal{Y}_1) < z \text{ and } \sum_{i \in [K']} \lambda_i' B_{\mathrm{ord}(2,i)}(\mathcal{Y}_1) > z'$$

For the remainder of the proof, we will suppose Case I is true. The proof for Case II is entirely analogous to what we will present below, since we will not use the direction of the signs of the two halfspaces in the following proof. Denote the two halfspaces induced by the second-layer of $f$ and $f'$ by $H_1$ and $H_2$ respectively.

$$H_1 := \{B \in \mathbb{R}^k : \sum_{i \in [K]} \lambda_i B_{\mathrm{ord}(1,i)} > z\}$$

$$H_2 := \{B \in \mathbb{R}^k : \sum_{i \in [K']} \lambda_i' B_{\mathrm{ord}(2,i)} < z'\}$$

By Corollary E.14 (Completeness of Basis Schema), there exists a continuous test-function $\mathcal{Y}_2$ of a basis test-function schema, $Y$, specified in Corollary E.14, such that $(B_1(\mathcal{Y}_2), \ldots, B_k(\mathcal{Y}_2)) = (B_1(\mathcal{Y}_1), \ldots, B_k(\mathcal{Y}_1))$. Thus,

$$(B_1(\mathcal{Y}_2), \ldots, B_k(\mathcal{Y}_2)) \in H_1 \cap H_2$$

Let $M$ be the number of segments in the basis schema $Y$ of $\mathcal{Y}_2$. Note that our previous application of Lemma E.37 guarantees that $Y$ is a basis schema of either one or two monotone curves (see Definition E.9), ensuring that

$$M \leq 2k \tag{1}$$

as opposed to the naive bound of $M \leq k^2$ via Corollary E.15. This fact will be useful at the end for achieving a better final bound.

Denote the lengths of the $M$ segments of schema $Y$ as $(n_1, \ldots, n_M) \in [0, 1]^M$. Let $A^{(M)}(Y)$ be the set of valid segment lengths $(n_1, \ldots, n_M)$ for a continuous test-functions of schema $Y$, where a particular setting of $(n_1, \ldots, n_M)$ is valid if it obeys the constraints described by Lemma E.6 for schema $Y$ and $\sum_{i \in [M]} n_i = 1$.

$$A^{(M)}(Y) := \{(n_1, \ldots, n_M) : \text{ valid segment lengths of schema } Y \text{ and } \sum_{i \in [M]} n_i = 1\} \subset [0, 1]^M$$

Additionally, define $A(Y)$ as the analogous set to $A^{(M)}(Y)$ in the space of activations rather than the space of segment lengths.

$$A(Y) := \{(B_1(\mathcal{Y}), \ldots, B_k(\mathcal{Y})) : \mathcal{Y} \text{ valid test-function of schema } Y\} \subset [0, 1]^k$$

There exists a linear map $L : \mathbb{R}^M \to \mathbb{R}^k$ which maps points in $A^{(M)}(Y)$ to points in $A(Y)$. $L \in \{0, 1\}^{k \times M}$ is such that $L_{ij} = 1 \iff$ segment $j$ in schema $Y$ lies above line $i$ (that is, for every $\times$ in the domain of segment $j$, the y-value of the segment at $\times$ is at least $s_i \cdot \times$) and hence contributes to the $i$th activation $B_i(\mathcal{Y})$ of any test-function $\mathcal{Y}$ of schema $Y$. Using $L$, we can rewrite the inequalities which characterize $H_1, H_2$ in terms of $(n_1, \ldots, n_M)$.

$$H_1^{(M)} := \{(n_1, \ldots, n_M) : \sum_{i=1}^{K} \lambda_i B_{\mathrm{ord}(1,i)} > z\} = \{(n_1, \ldots, n_M) : \sum_{i=1}^{K} \lambda_i \sum_{j \text{ s.t. } L_{\mathrm{ord}(1,i),j}=1} n_j > z\}$$

$$H_2^{(M)} := \{(n_1, \ldots, n_M) : \sum_{i=1}^{K'} \lambda_i' B_{\mathrm{ord}(2,i)} < z'\} = \{(n_1, \ldots, n_M) : \sum_{i=1}^{K'} \lambda_i' \sum_{j \text{ s.t. } L_{\mathrm{ord}(2,i),j}=1} n_j < z'\}$$

Since $(\lambda_i)_{i \in [K]}, z$ are integers at most $T$ in magnitude and $(\lambda_i')_{i \in [K']}, z'$ are integers at most $T'$ in magnitude, the coefficients of the linear inequality for $H_1^{(M)}$ (resp. $H_2^{(M)}$), $\sum_{i=1}^{K} \lambda_i \sum_{j \text{ s.t. } L_{\mathrm{ord}(1,i),j}=1} n_j > z$ (resp. $\sum_{i=1}^{K'} \lambda_i' \sum_{j \text{ s.t. } L_{\mathrm{ord}(2,i),j}=1} n_j < z'$) are integers of at most $KT \leq T^3$ (resp. $K'T' \leq (T')^3$) in magnitude, since for each $j \in [M]$, at most every $i \in [K]$ (resp. $i \in [K']$) can contribute to the $j$th coefficient.

Now, we describe a few more properties of $A^{(M)}(Y)$. Suppose the segment lengths of $\mathcal{Y}_2$ in schema $Y$ is $(n_1(\mathcal{Y}_2), \ldots, n_M(\mathcal{Y}_2)) \in [0, 1]^M$. Then, we have:

$$(n_1(\mathcal{Y}_2), \ldots, n_M(\mathcal{Y}_2)) \in A^{(M)}(Y) \cap H_1^{(M)} \cap H_2^{(M)} \neq \emptyset$$

By Lemma E.16 $A^{(M)}(Y)$ is a polytope: the intersection of a finite number of halfspaces. It follows that $A^{(M)}(Y)$ is convex. Moreover, by Lemma E.16, we have $\dim(A^{(M)}(Y)) = M - 1$. Because $A^{(M)}(Y) \cap H_1^{(M)} \cap H_2^{(M)} \neq \emptyset$ and $H_1^{(M)}, H_2^{(M)}$ are open sets of dimension $M$, then by Lemma E.27, $\dim(A^{(M)}(Y) \cap H_1^{(M)} \cap H_2^{(M)}) = \dim(A^{(M)}(Y)) = M - 1$.

We have that $\mathrm{cl}(A^{(M)}(Y) \cap H_1^{(M)} \cap H_2^{(M)})$ is the intersection of a finite number of halfspaces. Each face of the polytope $\mathrm{cl}(A^{(M)}(Y) \cap H_1^{(M)} \cap H_2^{(M)})$ is defined by one of these halfspaces. We now discuss the precision of the linear inequalities

which define the faces of $\mathrm{cl}(A^{(M)}(Y) \cap H_1^{(M)} \cap H_2^{(M)})$, where precision of an linear inequality with integer coefficients is the maximum magnitude of the integer coefficients, per Definition 5.2. The linear inequalities of the halfspaces which form the faces of $A^{(M)}(Y)$ are such that there is a subset of at most $6K$ of them with precision at most $T^2$, while the remaining faces of $A^{(M)}(Y)$ have precision at most $(T')^2$. This is due to the following argument. First, because the $(k, T)$-configuration $\{s_i\}_{i \in [k]}$ is such that there is a subset of at most $K$ of the $k$ elements of $\{s_i\}_{i \in [k]}$ which are precision at most $T$, while the rest of $\{s_i\}_{i \in [k]}$ are precision at most $T'$. Next, referring to Lemma E.6, the only faces of $A^{(M)}(Y)$ with $T^2$ precision are ones that correspond to a segment of $Y$ whose start-point or end-point is on one of the $K$ lines of slope whose precision is $T^2$. Third, by Corollary E.14, any basis schema of one or two monotone curves will cross each of the $k$ lines at most three times. Since each of the (at most) $K$ slopes of precision $T$ correspond to at most 3 crossing-points of $Y$ with that the line of that slope, and each crossing point is adjacent to at most two segments of $Y$, then there are at most $2 \cdot 3K$ segments of $Y$ such that the start-point or end-point is on a line of slope that is precision $T$. Thus, at most $6K$ of the faces of $A^{(M)}(Y)$ are defined by linear inequalities of precision at most $T^2$, while the remaining faces of $A^{(M)}(Y)$ are defined by linear inequalities of precision at most $(T')^2$. Note that the square (i.e. in $T^2$ and $(T')^2$) comes from the form of the inequalities defining the faces of $A^{(M)}(Y)$, stated in Lemma E.6. Finally, as argued in an earlier paragraph, the face given by $H_1^{(M)}$ has precision at most $T^3$ while that given by $H_2^{(M)}$ has precision at most $(T')^3$. In summary, there is a subset of at most $7K$ faces of $\mathrm{cl}(A^{(M)}(Y) \cap H_1^{(M)} \cap H_2^{(M)})$ such that each face of the subset has precision at most $T^3$ while the faces not in the subset all have precision at most $(T')^3$. In short, we've shown that polytope $\mathrm{cl}(A^{(M)}(Y) \cap H_1^{(M)} \cap H_2^{(M)})$ satisfies the pre-conditions of Corollary E.21, which we will apply in the next step.

Now, we return to the process of converting $\mathcal{Y}_2$ into a short distinguisher of $f, f'$. Let $V$ denote the set of vertices of the polytope $\mathrm{cl}(A^{(M)}(Y) \cap H_1^{(M)} \cap H_2^{(M)})$. Let $c \in [0, 1]^M$ be the average of the vertices in $V$.

$$c = \frac{1}{|V|} \sum_{x \in V} x$$

Label the coordinates of $c$ as $c := (n_1^{(c)}, \ldots, n_M^{(c)})$. $c$ is in the relative interior of $\mathrm{cl}(A^{(M)}(Y) \cap H_1^{(M)} \cap H_2^{(M)})$ (which is non-empty by Lemma E.25) so that $c \in A^{(M)}(Y) \cap H_1^{(M)} \cap H_2^{(M)}$. In particular,

$$\sum_{i=1}^{K} \lambda_i \sum_{j \text{ s.t. } L_{\mathrm{ord}(1,i),j}=1} n_j^{(c)} > z$$

$$\sum_{i=1}^{K'} \lambda'_i \sum_{j \text{ s.t. } L_{\mathrm{ord}(2,i),j}=1} n_j^{(c)} < z'$$

We will now lower bound the margin (see Definition 6.10) of $c$ to the faces of $\mathrm{cl}(A^{(M)}(Y) \cap H_1^{(M)} \cap H_2^{(M)})$, which includes the faces given by $H_1^{(M)}, H_2^{(M)}$. Suppose $\mathrm{cl}(A^{(M)}(Y) \cap H_1^{(M)} \cap H_2^{(M)})$ has $N$ faces, and let $\{L_i\}_{i \in [N]}$ denote the linear inequalities which define each face of $\mathrm{cl}(A^{(M)}(Y) \cap H_1^{(M)} \cap H_2^{(M)})$, so that $L_i(c) \in \mathbb{R}$ is the non-negative margin of $c$ on the $i$th face. Noting that $M \leq 2k \leq 2(K + K')$, we apply Corollary E.21 on polytope $\mathrm{cl}(A^{(M)}(Y) \cap H_1^{(M)} \cap H_2^{(M)})$ to get the following lower bound on the margin of $c$ on the faces of $\mathrm{cl}(A^{(M)}(Y) \cap H_1^{(M)} \cap H_2^{(M)})$.

$$\forall i \in [N], L_i(c) \geq \Omega\left(\frac{1}{|V|} \frac{1}{\alpha^{O(1)}}\right)$$

Then by Lemma E.22, we have $|V| \leq 3M^2$, so we deduce that:

$$\forall i \in [N], L_i(c) \geq \Omega\left(\frac{1}{M^2} \frac{1}{\alpha^{O(1)}}\right)$$

In particular, the margins, $\gamma_1, \gamma_2$ of $c$ on the two inequalities defining $H_1^{(M)}$ and $H_2^{(M)}$ will be at least:

$$\gamma_1 := \sum_{i=1}^{K} \lambda_i \sum_{j \text{ s.t. } L_{\text{ord}(1,i),j}=1} n_j^{(c)} - z$$

$$\geq \Omega(\frac{1}{M^2} \frac{1}{\alpha^{O(1)}})$$

$$\gamma_2 := z' - \sum_{i=1}^{K'} \lambda_i' \sum_{j \text{ s.t. } L_{\text{ord}(2,i),j}=1} n_j^{(c)}$$

$$\geq \Omega(\frac{1}{M^2} \frac{1}{\alpha^{O(1)}})$$

Now that we have shown that $c$ has large margin, we need to augment it one final time before converting it into a discrete test-function. This process will find a nearby point $c_* \in \text{Ball}_r^\infty(c) \cap \{\sum_{i \in [M]} n_i = 1\} := \{(n_1, \ldots, n_M) \in \mathbb{R}^{\mathbb{M}} : \|(n_1, \ldots, n_M) - c\|_\infty \leq r\} \cap \{\sum_{i \in [M]} n_i = 1\}$ such that $c_*$ has both large margin to the faces of $\text{cl}(A^{(M)}(Y) \cap H_1^{(M)} \cap H_2^{(M)})$ and low-precision coordinates. With $\|L_i\|_1$ denoting the sum of the magnitudes of the coefficients of the linear inequality which defines the $i$th face of $\text{cl}(A^{(M)}(Y) \cap H_1^{(M)} \cap H_2^{(M)})$, let:

$$\gamma_{LB} := \frac{1}{\lceil O(M^2 \alpha^{O(1)}) \rceil} \leq \gamma_1, \gamma_2$$

$$r := \frac{\gamma_{LB}}{2 \cdot \lceil \max_{i \in [N]} \|L_i\|_1 \rceil}$$

By linearity of the margin of a point in its coordinates $(n_1, \ldots, n_M)$, each point $c' \in \text{Ball}_r^\infty(c) \cap \{\sum_{i \in [M]} n_i = 1\}$ will have margin at least $\gamma_{LB} - r \cdot \max_{i \in [N]} \|L_i\|_1 \geq \frac{\gamma_{LB}}{2}$ to each face of $\text{cl}(A^{(M)}(Y) \cap H_1^{(M)} \cap H_2^{(M)})$. This also means that every such $c'$ is contained in $A^{(M)}(Y) \cap H_1^{(M)} \cap H_2^{(M)}$ as $\gamma_{LB} > 0$.

By Lemma E.29, there exists a low-precision point $c_* \in \text{Ball}_r^\infty(c) \cap \{\sum_{i \in [M]} n_i = 1\}$, denoted $c_* := (n_1^{(c_*)}, \ldots, n_M^{(c_*)})$, such that for all $i \in [M]$ $n_i^{(c_*)}$ is a rational number, and the least common denominator of all elements in the tuple $(n_1^{(c_*)}, \ldots, n_M^{(c_*)})$ is $p_{c_*}$, where:

$$p_{c_*} \leq \lceil M \cdot \frac{1}{r} \rceil$$

$$\leq O(M \lceil \max_{i \in [N]} \|L_i\|_1 \rceil \cdot \lceil 3M^2 \alpha^{O(1)} \rceil)$$

$$\leq O(M^3 \cdot (M \cdot T^3) \cdot \alpha^{O(1)})$$

Let the tuple $c_* = (n_1^{(c_*)}, \ldots, n_M^{(c_*)})$ be the segment lengths of the continuous test-function $\mathcal{Y}_*$ of schema $Y$. Note that $c_*$ has positive margin $\frac{\gamma_{LB}}{2}$ to all the faces of $\text{cl}(A^{(M)}(Y) \cap H_1^{(M)} \cap H_2^{(M)})$ and is contained in $A^{(M)}(Y) \cap H_1^{(M)} \cap H_2^{(M)}$, so $c_*$ is a valid setting of segment lengths of schema $Y$, respecting the constraints of Lemma E.6.

Finally, applying Lemma E.24 on the continuous test-function of schema $Y$ with segment lengths $c_* = (n_j^{(c_*)})_{j \in [M]}$, we deduce that there exists a $n_0 \leq O(p_{c_*} \cdot \alpha^{O(1)})$ so that for any integer multiple $n$ of $n_0$, there exists a discrete test-function $\mathcal{X}_* : \{0, \ldots, n\} \to \{0, \ldots, n\}$, of length $n$, corresponding to string $x_* \in \{0, 1\}^n$ of length $n$, such that

$$\forall i \in [k], |B_i(\mathcal{Y}_*) - B_i(\mathcal{X}_*)| \leq \frac{T^2 + M}{n}$$

Where $(B_i(\mathcal{Y}_*))_{i\in[k]}$ are the activations induced by $\mathcal{Y}_*$, a test-function of schema $Y$ with segment lengths $(n_i^{(c_*)})_{i\in[M]}$, and $(B_i(\mathcal{X}_*))_{i\in[k]}$ the activations induced by $\mathcal{X}_*$. Because $\forall i, |\lambda_i| \leq T, |\lambda_i'| \leq T' \leq T$, then the difference between the margin of $(B_i(\mathcal{Y}_*))_{i\in[k]}$ and $(B_i(\mathcal{X}_*))_{i\in[k]}$ on $H_1$ and $H_2$ can be bounded by a term proportional to $\frac{1}{n}$.

$$|\sum_{i=1}^{K} \lambda_i B_{\text{ord}(1,i)}(\mathcal{Y}_*) - \sum_{i=1}^{K} \lambda_i B_{\text{ord}(1,i)}(\mathcal{X}_*)| \tag{2}$$

$$\leq (\max_{i\in[k]} |B_i(\mathcal{Y}_*) - B_i(\mathcal{X}_*)|) \cdot (\max_{\forall i\in[k], |\lambda_i|\leq T} \sum_{i=1}^{K} \lambda_i) \tag{3}$$

$$\leq \frac{(T^2+M)KT}{n} \tag{4}$$

$$\leq O(\frac{KMT^3}{n}) \tag{5}$$

Using an analogous argument, the difference between $\sum_{i=1}^{K'} \lambda_i' B_{\text{ord}(2,i)}(\mathcal{Y}_*)$ and $\sum_{i=1}^{K'} \lambda_i' B_{\text{ord}(2,i)}(\mathcal{X}_*)$ is also bounded by an analogous expression.

$$|\sum_{i=1}^{K'} \lambda_i' B_{\text{ord}(2,i)}(\mathcal{Y}_*) - \sum_{i=1}^{K'} \lambda_i' B_{\text{ord}(2,i)}(\mathcal{X}_*)| \leq O(\frac{K'MT^3}{n})$$

Together, these imply that for sufficiently large $n$, the difference in the margin of $(B_i(\mathcal{X}_*))_{i\in[k]}$ and $(B_i(\mathcal{Y}_*))_{i\in[k]}$ on $H_1$ and $H_2$, caused by the discrete test-function approximation, will be smaller than the margin of $(B_i(\mathcal{Y}_*))_{i\in[k]}$ on $H_1$ and $H_2$. The latter equals the margin of $c_*$ on $H_1^{(M)}$ and $H_2^{(M)}$, by the definition of $H_1^{(M)}$ and $H_2^{(M)}$ as the analogous halfspaces to $H_1$ and $H_2$, which is lower bounded by $\frac{\gamma_{LB}}{2}$. More precisely,

$$\sum_{i=1}^{K} \lambda_i B_i(\mathcal{X}_*) \geq z + \frac{\gamma_{LB}}{2} - O(\frac{KMT^3}{n})$$

$$\sum_{i=1}^{K'} \lambda_i' B_i(\mathcal{X}_*) \leq z' - \frac{\gamma_{LB}}{2} + O(\frac{K'MT^3}{n})$$

To this end, since $\gamma_{LB} = \Omega(\frac{1}{M^2\alpha^{O(1)}})$, it suffices for $n$ to be the following value in order for activations $(B_i(\mathcal{X}_*))_{i\in[k]}$ induced by $\mathcal{X}_*$ to be in $H_1 \cap H_2$ (and therefore to cause $f$ and $f'$ to differ, as $\mathcal{Y}_1$ and $\mathcal{Y}_*$ do).

$$\Omega(\frac{1}{M^2\alpha^{O(1)}}) - O(\frac{\max(K,K')MT^3}{n}) > 0 \iff n > O(M^3 \max(K,K')T^3\alpha^{O(1)}) \tag{6}$$

Let $x_*$ be the string of length $n$ corresponding to $\mathcal{X}_*$, which is uniquely determined by $\mathcal{X}_*$. For $n = \max(1 + O(M^3 \max(K,K')T^3\alpha^{O(1)}), O(p_{c_*} \cdot \alpha^{O(1)}))$, $n$ will be sufficiently large to make the approximation error smaller than the margin of $c_*$ per Equation 6, and also satisfy the condition required to apply Lemma E.24 in the previous part of this proof. Thus, with this value of $n$, $x_*$ will cause $f(x_*) = 1, f'(x_*) = 0$.

We noted previously in Equation (1) that $M \leq 2k \leq 2(K+K')$ for the basis schema $Y$ as a result of Lemma E.37. Plugging this in for $M$, and noting that $\alpha \geq \max(T^K, (T')^{K'})$, we conclude that the length of such an $x'$ distinguishing $f, f'$ need only be at most

$$n = \max(1 + O(M^3 \max(K, K')T^3 \alpha^{O(1)}), O(p_{c_*} \cdot \alpha^{O(1)}))$$
$$\leq \max(1 + O(M^3 \max(K, K')T^3 \alpha^{O(1)}), O(M^3 \cdot (MT^3) \cdot \alpha^{O(1)} \cdot \alpha^{O(1)})))$$
$$\leq \boxed{O(\alpha^{O(1)})}$$

∎

### E.3. Lemmas for Completeness of Basis Schema

**Goal.** In the main proof, we will fix two arbitrary unequal $f, f' \in$ C-RASP$^2$ and prove they have a short distinguisher. The goal of this section is to prove that the set of realizable activations $\mathbb{A}(\{s_i\}_{i \in [k]})$ equals the union of the set of activations of a small number of *basis* test-function schema. This culminates in Corollary E.14.

**Basic Definitions.** Recall the definition of precision.

**Definition 5.2** ($p$-precision)**.** An integer of absolute value at most $p$ is of $p$-precision. A rational number between $[0, 1]$ is of $p$-precision if in simplest form, where the numerator and denominator are relatively prime, its denominator is at most $p$ in magnitude. A tuple of rational numbers in $[0, 1]$ is precision $p$ if the least common denominator of its entries is at most $p$ in magnitude.

Note we will say a halfspace defined via a linear inequality with integer coefficients has $p$-precision if each coefficient is at most $p$ in magnitude.

Suppose $f$ and $f'$ have $K$ and $K'$ heads, respectively, and consider the set of $\max(K, K') \leq k \leq K + K'$ distinct slopes from the parameters of the first layer of $f$ and $f'$: $\{\frac{b^{(i)}}{a^{(i)}}\}_{i \in [K]} \cup \{\frac{(b^{(i)})'}{(a^{(i)})'}\}_{i \in [K']}$. Disregard for now which slope belongs to $f$ or to $f'$, and denote these $k$ slopes as $\{s_i\}_{i \in [k]} \subset (0, 1)$, sorted descending so that $s_1$ is largest and $s_k$ is smallest. We will refer to $\{s_i\}_{i \in [k]} \subset (0, 1)$ as a $(k, T)$-configuration.

**Definition 6.4.** A $(k, T)$-configuration is a set of $k$ distinct $T$-precision rational numbers $\{s_i\}_{i \in [k]} \subset (0, 1)$.

These $k$ slopes $\{s_i\}_{i \in [k]}$ specify $k$ homogeneous, 2D lines, of the form $\mathsf{y} = s_i \cdot \mathsf{x}$. Denote these $k$ lines as $l_1, \dots, l_k$ with line $l_i$ having slope $s_i$ for all $i \in [k]$.

Recall the definition of Discrete Test-Functions.

**Definition 6.5** (Discrete Test-Function)**.** Given a $(k, T)$-configuration $\{s_i\}_{i \in [k]}$, a discrete test-function $\mathcal{X}$, with respect to $\{s_i\}_{i \in [k]}$ and of length $n < \infty$, is a function $\{0, 1, \dots, n\} \to \{0, 1, \dots, n\}$ where $\mathcal{X}(0) = 0$ and $\forall j \in [n], \mathcal{X}(j) = \mathcal{X}(j-1)$ or $\mathcal{X}(j) = \mathcal{X}(j-1) + 1$. The induced activations $(B_1(\mathcal{X}), \dots, B_k(\mathcal{X}))$ of $\mathcal{X}$ with respect to the $(k, T)$-configuration are defined as: $\forall i \in [k], \quad B_i(\mathcal{X}) := \frac{1}{n} \sum_{j=1}^{n} \mathbb{1}[\mathcal{X}(j) > s_i \cdot j]$

For a string $x \in \{0, 1\}^*$, the discrete test-function induced by $x$ is the set of 2D points $\{(j, \mathsf{ps}(x)_j)\}_{j \in [|x|]}$ where we will associate the y-axis for $\mathsf{ps}(x)_j$ and the x-axis for $j$.

Recall two central objects: continuous test-functions and $\mathbb{A}(\{s_i\}_{i \in [k]})$.

**Definition 6.6** (Continuous Test-Function)**.** Given a $(k, T)$-configuration $\{s_i\}_{i \in [k]}$, a continuous test-function $\mathcal{Y}$, with respect to $\{s_i\}_{i \in [k]}$, is a 1-Lipschitz, monotone non-decreasing continuous function $[0, 1] \to [0, 1]$, with $\mathcal{Y}(0) = 0$. Continuous test-functions can only intersect the $k$ lines $\{l_i\}_{i \in [k]}$ of slopes given by $\{s_i\}_{i \in [k]}$ at finitely many points. The induced activations $(B_1(\mathcal{Y}), \dots, B_k(\mathcal{Y}))$ of $\mathcal{Y}$ w.r.t. $\{s_i\}_{i \in [k]}$ are: $\forall i \in [k], \quad B_i(\mathcal{Y}) := \int_0^1 \mathbb{1}[\mathcal{Y}(j) > s_i \cdot j]dj$.

**Definition E.1.** ($\mathbb{A}(\{s_i\}_{i \in [k]})$) Given $(k, T)$-configuration $\{s_i\}_{i \in [k]}$, define $\mathbb{A}(\{s_i\}_{i \in [k]})$ as the set of activations induced by continuous test-functions with respect to $\{s_i\}_{i \in [k]}$.

$$\mathbb{A}(\{s_i\}_{i \in [k]}) := \{(B_1(\mathcal{Y}), \dots, B_k(\mathcal{Y})) : \mathcal{Y} \text{ continuous test-function w.r.t. } \{s_i\}_{i \in [k]}\}$$

Regarding properties of continuous test-functions, first note that the scaling of $\mathcal{Y}$ can be set WLOG because of the homogeneity of the $k$ lines. Thus, we let their domain be $[0, 1]$.

Second, for any continuous test-function $\mathcal{Y}$, we can let the end point of $\mathcal{Y}$ be on one of the lines $\{l_i\}_{i \in [k]}$. Suppose the last line crossed by $\mathcal{Y}$ is $l_i$. Then we can adjust the segment of $\mathcal{Y}$ between its last crossing point at $l_i$ and its endpoint so that the endpoint is also on line $l_i$. We can make this tweak so that no other lines $\{l_i\}_{i \in [k]}$ are crossed and so that $(B_1(\mathcal{Y}), \ldots, B_k(\mathcal{Y}))$ remains unchanged by this tweak. In short, the endpoint of any continuous test-function $\mathcal{Y}$ is, WLOG, $(1, s_i)$ where $l_i$ is the last line crossed by $\mathcal{Y}$.

**Lemma E.2.** *For any configuration $\{s_i\}_{i \in [k]}$, for any continuous test-function $\mathcal{Y}$ w.r.t. $\{s_i\}_{i \in [k]}$, suppose the last line $\{l_i\}_{i \in [k]}$ crossed by $\mathcal{Y}$ is $l_i$, for $i \in [k]$. Then, WLOG, we can let $\mathcal{Y}$'s end point be $(1, s_i)$ without changing the activations induced by $\mathcal{Y}$.*

*Proof.* Suppose that the last line $\mathcal{Y}$ crosses is $l_i$ at the point $(x, s_i \cdot x)$. Then, the portion of $\mathcal{Y}$ on the interval $[x, 1]$ is wedged between either the two lines $l_i$ and $l_{i+1}$ or the two lines $l_i$ and $l_{i-1}$, since $\mathcal{Y}$ will not cross any other line on the interval. The quantity of interest are the activations with respect to the $k$ lines induced by $\mathcal{Y}$:

$$\forall i \in [k], B_i(\mathcal{Y}) := \int_0^1 \mathbb{1}[\mathcal{Y}(j) > s_i \cdot j] dj$$

Suppose that $\forall j \in (x, 1], s_i j > \mathcal{Y}(j) > s_{i+1} j$. The $k$ quantities $\{B_i(\mathcal{Y})\}_{i \in [k]}$ will be unchanged if we adjust the values of $\mathcal{Y}(j)$ for $j \in (x, 1]$ so long as we retain that $\forall j \in (x, 1], s_i j > \mathcal{Y}(j) > s_{i+1} j$ except on a set of measure 0. With $\mathcal{Y}$ allowed to be any continuous function with slopes in $[0, 1]$ and with $s_i \in (0, 1)$, we can adjust $\mathcal{Y}(j)$ to stay between lines $l_i$ and $l_{i+1}$ but closely follow the line $l_i$ in the sense that $|\mathcal{Y}(j) - s_i j| > 0$ can be made arbitrarily small at all points $j \in (x, 1)$, and $\mathcal{Y}(1) = s_i$ (note, this end-point $(1, s_i)$ violates the condition that $s_i j > \mathcal{Y}(j) > s_{i+1} j$ but only at a single point). This ensures that the modified test-function, call it $\mathcal{Y}'$, is such that $\forall i \in [k], B_i(\mathcal{Y}) = B_i(\mathcal{Y}')$.

Suppose that $\forall j \in (x, 1], s_{i-1} j > \mathcal{Y}(j) > s_i j$. Then an analogous adjustment to $\mathcal{Y}(j)$ on the interval $j \in (x, 1]$ can be made so that the endpoint is $(1, s_i)$. ∎

From now on, assume that each test-function will have starting-point at the origin $(0, 0)$ and have end point on some line $l_i \in \{l_1, \ldots, l_k\}$, at point $(1, s_i)$. This will make the following definitions about segments and schema cleaner.

Define the span of a continuous test-function as the set of lines in $\{l_i\}_{i \in [k]}$ which the continuous test-function intersects at some point.

**Definition E.3.** The span of a test-function is the set of lines $\{l_1, \ldots, l_k\}$ that the test-function crosses at least once. That is, $\mathcal{Y}$ crosses $l_i$ if there exists $x$ where $\mathcal{Y}(x) = s_i \cdot x$. Note that the span must be a contiguous subset of $[k]$.

We'll also give names to the regions of the positive quadrant of the 2D plane between consecutive lines in $\{l_i\}_{i \in [k]}$.

**Definition E.4.** (Sectors) Given a $(k, T)$-configuration $\{s_i\}_{i \in [k]}$, a sector is a region of the 2D space between two consecutive lines. Define $\text{Sector}_1$ as the sector strictly above line $l_1$, and $\text{Sector}_{k+1}$ as the sector below line $l_k$. For $i \in \{2, \ldots, k\}$ define $\text{Sector}_i$ as the sector below line $l_{i-1}$ and strictly above line $l_i$.

Sectors are depicted in Figure 7. We'll now define segments and schema.

**Definition E.5.** (Segments and Schema) Given any $(k, T)$-configuration $\{s_i\}_{i \in [k]} \subset (0, 1)$, define the following.

1. A *segment* is a restricted test-function $S : [a, b] \to [0, 1]$ where $[a, b] \subset [0, 1]$ which maps a continuous subset $[a, b]$ to $[0, 1]$. $S$ is 1-Lipschitz and monotone non-decreasing. The segment's start-point $(a, S(a))$ and the end-point $(b, S(b))$ each lie on one of the $k$ lines, in the sense that there exists some $i, j \in [k]$ where $S(a) = s_i \cdot a$ and $S(b) = s_j \cdot b$, where $i = j$ or $|i - j| = 1$. No other points $(x, S(x))$, $x \in (a, b)$ can lie on a line $l_1, \ldots, l_k$.

2. A *schema* $Y$ is a blueprint for a continuous test-function, specifying a sequence of lines $\{l_i\}_{i \in [k]}$ that any test-function of the schema must cross. It consists of an integer $0 < M < \infty$ and two tuples $\{\text{idx}(i)\}_{i \in [M]} \subset [k]^M$, $\{\text{sec}_i\}_{i \in [M]} \subset [k+1]^M$, where $|\text{idx}(i) - \text{idx}(i+1)| \le 1$ for all $i \in [M-1]$. If $|\text{idx}(i) - \text{idx}(i+1)| = 1$, then $\text{sec}_{i+1}$ is unique and must be $\max(\text{idx}(i), \text{idx}(i+1))$. If $\text{idx}(i) = \text{idx}(i+1)$, then $\text{sec}_{i+1}$ can be either $\text{idx}(i+1)$ or $\text{idx}(i+1) + 1$. $\text{sec}_1$ can be either $\text{idx}(1)$ or $\text{idx}(1) + 1$.

   Any continuous test-function of schema $Y = (\{\text{idx}(i)\}_{i \in [M]}, \{\text{sec}_i\}_{i \in [M]})$ consists of exactly $M$ segments $S_1, S_2, \ldots, S_M$ whose domains are a partition of $[0, 1]$. For each $i \in [M]$, the $i$th segment $S_i$'s end-point lies

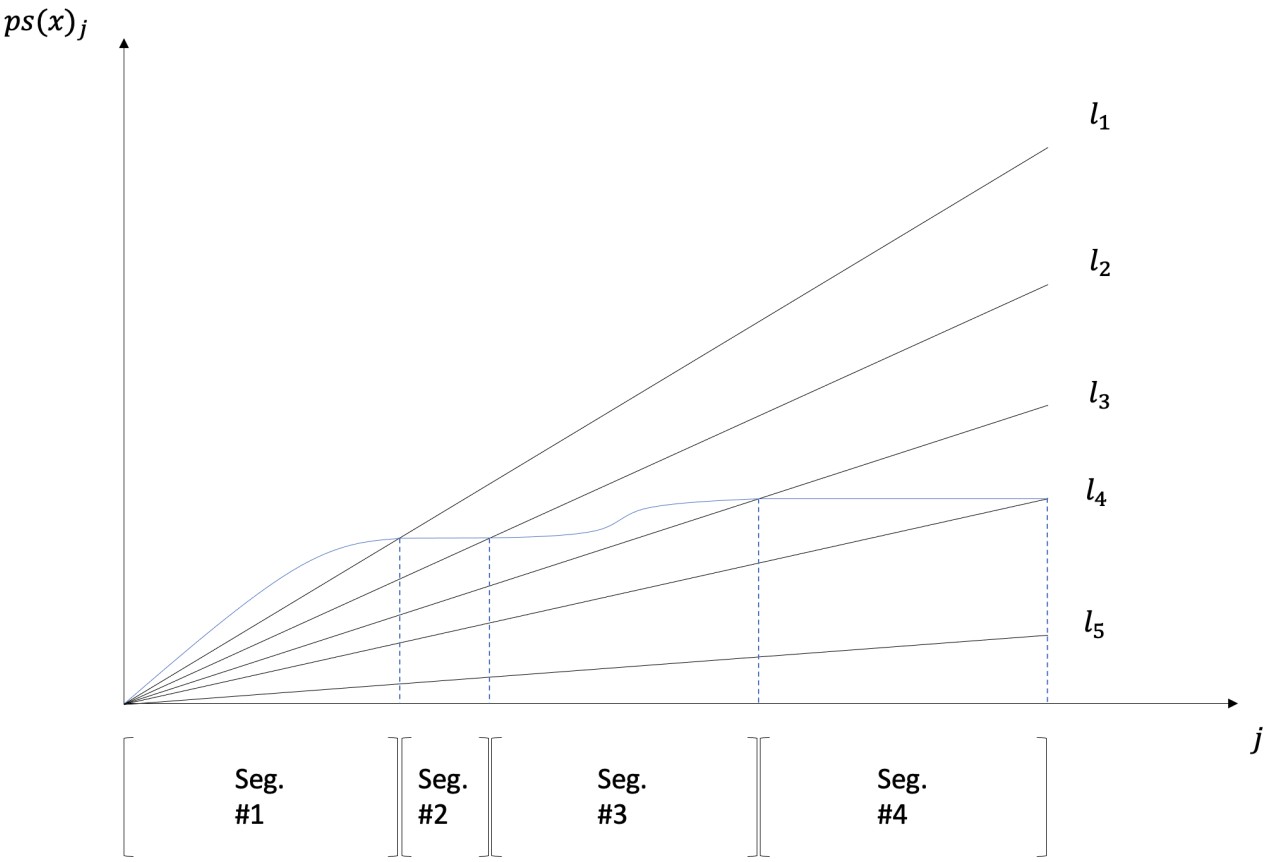

*Figure 2.* Depiction of a Test-Function consisting of 4 segments. y-axis shows the prefix sum of input string $x$. x-axis shows the length of the prefix of input string $x$. The five lines have slopes $\{s_1, \ldots, s_5\}$. Sector$_1$ is the portion of the quadrant which is above $l_1$. Sector$_6$ is the portion of the quadrant below $l_5$. For $2 \le i \le 5$, Sector$_i$ is the portion of the quadrant below $l_{i-1}$ and above $l_i$.

on $l_{\text{idx}(i)}$. For $i > 1$, $S_i$'s start-point lies on line $l_{\text{idx}(i-1)}$, and $S_1$'s start-point is the origin, $(0, 0)$. In addition, the $i$th segment must be contained in Sector$_{\text{sec}_i}$.

Notationally, we denote $n_i \in [0, 1]$ as the length of the $i$th segment $S_i$, but note that different test-functions of the same schema may have different segment lengths $\{n_i\}_{i \in [M]}$, subject to some constraints we detail below. For a schema $Y$, denote

$$A(Y) := \{(B_1(\mathcal{Y}), \ldots, B_k(\mathcal{Y})) : \mathcal{Y} \text{ continuous test-function of schema } Y \text{ w.r.t. } \{s_i\}_{i \in [k]}\}$$

Figure 2 shows a test-function of a schema with 4 segments. Figure 7 shows four generic types of segments.

We will think of a schema as a list of $M$ segments, for some $M > 0$. We will denote the length of the $i$th segment as $n_i$. Let $\text{idx}(\cdot)$ be the mapping from $[M]$ to $[k]$ that gives the index of the line that segment $i$'s endpoint is on.

For continuous test-functions, the first segment of length $n_1$ has start-point at the origin, $(0, 0)$. The last segment of length $n_M$ has end-point which lies on the line $l_{\text{idx}(M)}$. The $i$th segment has start-point $(\sum_{j=1}^{i-1} n_j, s_{\text{idx}(i-1)} \cdot \sum_{j=1}^{i-1} n_j)$ on line $l_{\text{idx}(i-1)}$ (as long as $i \ge 2$) and end-point $(\sum_{j=1}^{i} n_j, s_{\text{idx}(i)} \cdot \sum_{j=1}^{i} n_j)$ on line $l_{\text{idx}(i)}$ and has length $n_i$. There is freedom in choosing $\{n_i\}_{i \in [M]}$, so long as they satisfy the following constraints.

**Lemma E.6.** *For any segment in any schema, the constraints given in Definition E.5 exactly characterize the range of values allowed for that segment. For all $i \in [M]$, these constraints are:*

- *(Segment Starts and Ends on Same Line) If* $\text{idx}(i - 1) = \text{idx}(i)$ *or* $i = 1$, *the only constraint on the segment's length is*

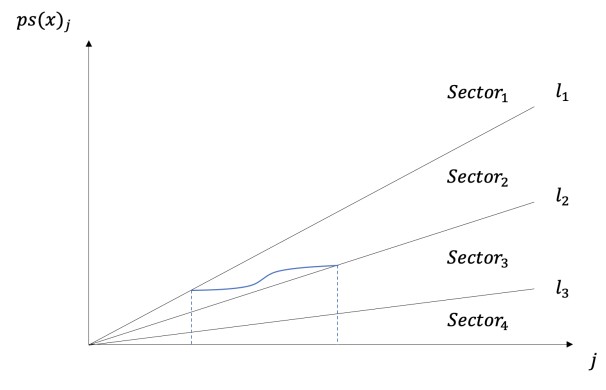

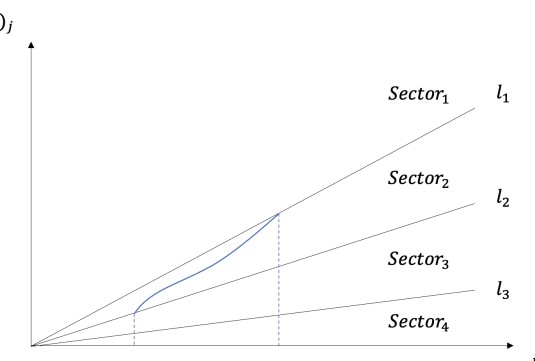

*Figure 3.* *
Start-point on $l_1$, End-point on $l_2$.

*Figure 4.* *
Start-point on $l_2$, End-point on $l_1$.

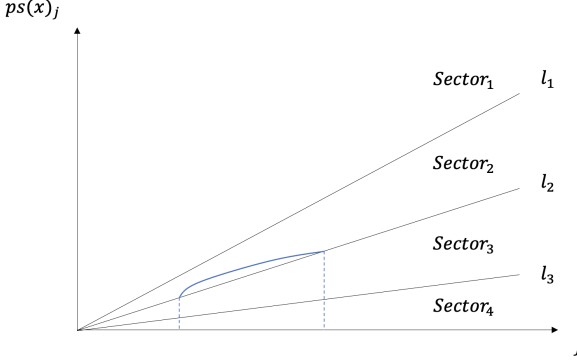

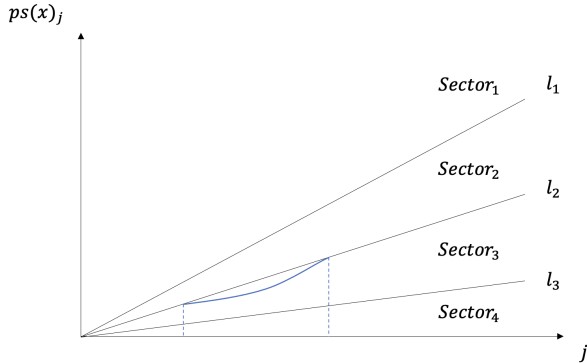

*Figure 5.* *
Start-point and End-point on $l_2$, and in Sector$_2$.

*Figure 6.* *
Start-point and End-point on $l_2$, and in Sector$_3$.

*Figure 7.* Four types of Segments, based on which lines their start-point and end-point lie on, and the sector they are in.

$n_i \geq 0$.

- *(Segment Crosses Down) If* $\mathrm{idx}(i-1) = \mathrm{idx}(i) - 1$ *and* $i \geq 2$, *then* $n_i \geq \left(\frac{s_{\mathrm{idx}(i-1)}}{s_{\mathrm{idx}(i)}} - 1\right) \sum_{j=1}^{i-1} n_j$

- *(Segment Crosses Up) If* $\mathrm{idx}(i-1) = \mathrm{idx}(i) + 1$ *and* $i \geq 2$, *then* $n_i \geq \left(\frac{1 - s_{\mathrm{idx}(i-1)}}{1 - s_{\mathrm{idx}(i)}} - 1\right) \sum_{j=1}^{i-1} n_j$

*Proof.* **Case: Segment Starts and Ends on Same Line.** For segment $i$ with length $n_i$, if $\mathrm{idx}(i-1) = \mathrm{idx}(i)$, then $n_i$ can be any nonnegative number. This is because for each $n_1, \ldots, n_{i-1}$ and for all $n_i \geq 0$, there exists a segment of length $n_i$ that crosses line $\mathrm{idx}(i)$ at the start-point $(\sum_{j \leq i-1} n_j, s_{\mathrm{idx}(i)} \sum_{j \leq i-1} n_j)$ and end-point $(\sum_{j \leq i} n_j, s_{\mathrm{idx}(i)} \sum_{j \leq i} n_j)$ and nowhere in between. An example of such a segment is one which stays arbitrarily close to the line $l_{\mathrm{idx}(i)}$ but doesn't cross it until the end-point $(\sum_{j \leq i} n_j, s_{\mathrm{idx}(i)} \sum_{j \leq i} n_j)$, which is possible since each line has slope in the range $(0, 1)$ while the test-function can have slopes at each point be in the range $[0, 1]$.

**Case: Segment Crosses Down.** If $\mathrm{idx}(i-1) = \mathrm{idx}(i) - 1$, then for any $n_1, \ldots, n_{i-1}$, the minimum value of $n_i$ in terms of $n_1, \ldots, n_{i-1}$ is achieved if the segment has slope of 0 at all points, the "minimal segment", which will let the segment cross lines $l_{\mathrm{idx}(i-1)}$ and $l_{\mathrm{idx}(i)}$ most efficiently. Such a minimal segment will cross lines $l_{\mathrm{idx}(i-1)}$ and $l_{\mathrm{idx}(i)}$ at start-point $(\sum_{j \leq i-1} n_j, s_{\mathrm{idx}(i-1)} \sum_{j \leq i-1} n_j)$ and end-point $(\sum_{j \leq i} n_j, s_{\mathrm{idx}(i)} \sum_{j \leq i} n_j)$, respectively. The value of $n_i$ required for this minimum traversal can be calculated by:

$$s_{\text{idx}(i)} \sum_{j=1}^{i} n_j = s_{\text{idx}(i-1)} \sum_{j=1}^{i-1} n_j$$

$$\implies n_i = \left(\frac{s_{\text{idx}(i-1)}}{s_{\text{idx}(i)}} - 1\right) \sum_{j=1}^{i-1} n_j$$

The first equation follows from the fact that the zero-slope trajectory will enforce that the y-coordinate of the start-point of the segment $s_{\text{idx}(i-1)} \sum_{j=1}^{i-1} n_j$ equals that of the end-point $s_{\text{idx}(i)} \sum_{j=1}^{i} n_j$. Having $n_i$ be any smaller will not suffice to make the crossing, under the trajectory of the minimal segment (where the slope is 0 everywhere) and certainly under any other trajectories.

Now, having $n_i > \left(\frac{s_{\text{idx}(i-1)}}{s_{\text{idx}(i)}} - 1\right) \sum_{j=1}^{i-1} n_j$ is always possible, since the segment can always follow the trajectory of the minimal segment to cross to line $\text{idx}(i)$, and then use the extra slack, $n_i - \left(\frac{s_{\text{idx}(i-1)}}{s_{\text{idx}(i)}} - 1\right) \sum_{j=1}^{i-1} n_j > 0$ to closely follow line $\text{idx}(i)$ until it crosses it at the designated end-point, $(\sum_{j \leq i} n_j, s_{\text{idx}(i)} \sum_{j \leq i} n_j)$. The latter phase reduces to the case where $\text{idx}(i-1) = \text{idx}(i)$. Thus, the allowed values for $n_i$ are $n_i \geq \left(\frac{s_{\text{idx}(i-1)}}{s_{\text{idx}(i)}} - 1\right) \sum_{j=1}^{i-1} n_j$.

**Case: Segment Crosses Up.** The setup of lines and $[0,1]$-slope test-functions has a "reflection" symmetry about the line $y = \frac{1}{2}x$. Thus, the argument for the constraint on values of $n_i$ for the $\text{idx}(i-1) = \text{idx}(i) + 1$ case reduces to that of the case $\text{idx}(i-1) = \text{idx}(i) - 1$, except with "reflected" slopes $1 - s_{\text{idx}(i-1)}$ and $1 - s_{\text{idx}(i)}$. Plugging in these reflected slopes into the constraint for the Cross Down case yields $n_i \geq \left(\frac{1-s_{\text{idx}(i-1)}}{1-s_{\text{idx}(i)}} - 1\right) \sum_{j=1}^{i-1} n_j$.

Another way to see how to derive the constraint by considering the minimal trajectory as one where the segment has slope 1 everywhere; and arguing that $n_i$ can be anything larger than the length required by the minimal trajectory.

$$(1 - s_{\text{idx}(i)}) n_i = (s_{\text{idx}(i)} - s_{\text{idx}(i-1)}) \sum_{j=1}^{i-1} n_j$$

$$\implies n_i = \left(\frac{1 - s_{\text{idx}(i-1)}}{1 - s_{\text{idx}(i)}} - 1\right) \sum_{j=1}^{i-1} n_j$$

The first equation above is derived as "Object 1 of relative speed of $(1 - s_{\text{idx}(i)})$ to Object 2 takes $n_i$ time to close the initial gap of $(s_{\text{idx}(i)} - s_{\text{idx}(i-1)}) \sum_{j=1}^{i-1} n_j$." ∎

**Partial Test-functions and Monotone Curves.** We'll now define partial test-functions, which is a slight generalization of continuous test-functions where the start-point does not need to be $(0,0)$, which will be later in the later proofs.

**Definition E.7.** (Partial Continuous Test-Function) Given a $(k, T)$-configuration $\{s_i\}_{i \in [k]}$, a continuous test-function $\mathcal{Y}$ is partial if it is allowed to have a start-point at $(n_1, s_i, n_1), n_1 \in [0, 1), i \in [k]$, instead of $(0, 0)$. A partial test-function is undefined on $[0, n_1)$. The induced activations $(B_1(\mathcal{Y}), \ldots, B_k(\mathcal{Y}))$ of partial continuous test-function $\mathcal{Y}$ given $k$ slopes $\{s_i\}_{i \in [k]} \subset (0, 1)$ are defined as:

$$\forall i \in [k], \quad B_i(\mathcal{Y}) := \int_{n_1}^{1} \mathbb{1}[\mathcal{Y}(j) > s_i \cdot j] dj$$

Schemas of partial test-functions can be thought of as a schema of a continuous test-function. We will still denote the lengths of the segments of the schema as $\{n_i\}_{i \in [M]}$ for some $M > 1$, except that the test-function is undefined on the first segment's domain $[0, n_1)$.

Note that by a re-scaling argument, due to the homogeneity of the $k$ lines $\{l_i\}_{i \in [k]}$, this definition also captures continuous test-functions where the start-point does not need to be at $(0, 0)$ and the end-point need not have x-coordinate of 1.

We'll define type $(\text{I}, k)$ and $(\text{II}, k)$ partial test-functions, which are particular partial test-functions whose start-point is on a line whose index is either the smallest or largest element in the span of the test-function.

**Definition E.8.** (Partial test-functions of type (I, $k$), (II, $k$)) Given a $(k, T)$-configuration $\{s_i\}_{i \in [k]}$,

- Define a type (I, $k$) partial test-function as a test-function $[0, 1] \to [0, 1]$ that is undefined on $[0, n_1)$ for some $0 \le n_1 < 1$. It may span any consecutive subset of lines $\{a, \dots, b\} \subset \{1, 2, \dots, k-1, k\}, a \le b$. With this span, we require that its start-point is at $(n_1, s_a n_1)$, that it is 1-Lipschitz and monotone non-decreasing, and that it can only intersect the $k$ lines at finitely many points.

- Define a type (II, $k$) partial test-function similarly as a test-function $[0, 1] \to [0, 1]$ undefined on $[0, n_1)$. It may span any consecutive subset of lines $\{a, \dots, b\} \subset \{1, 2, \dots, k-1, k\}, a \le b$. With this span, we require that its start-point is $(n_1, s_b n_1)$, that it is 1-Lipschitz and monotone non-decreasing, and that it can only intersect the $k$ lines at finitely many points.

Note that for both type (I, $k$) partial test-functions, if the start-point is on $l_a$ with $a > 1$, then its span cannot contain $l_1$ and it cannot intersect $l_1$ at any point. An analogous observation holds for type (II, $k$) partial test-functions.

We'll describe one primitive (partial test-function) schema that will be important for our basis test-function schema later: monotone curves.

**Definition E.9.** (Monotone Curve) Given a $(k, T)$-configuration $\{s_i\}_{i \in [k]}$, we define two schemas: Monnotone Up Curve and Monotone Down Curve. Both schema have $k + 2$ segments.

- (Monotone Down) Begins at line 1 at $(n_1, s_1 \cdot n_1)$, goes above and recrosses line 1 at $(n_1 + n_2, s_1 \cdot [n_1 + n_2])$, crosses each intermediate line $\{2, \dots, k-1\}$ once, then crosses line $k$ twice at $(\sum_{i=1}^{k+1} n_i, s_k \cdot \sum_{i=1}^{k+1} n_i)$ and $(\sum_{i=1}^{k+2} n_i, s_k \cdot \sum_{i=1}^{k+2} n_i)$.

- (Monotone Up) Begins at line $k$ at $(n_1, s_k \cdot n_1)$, goes below and recrosses line $k$ at $(n_1 + n_2, s_k \cdot [n_1 + n_2])$, crosses each intermediate line once, then crosses line 1 twice at $(\sum_{i=1}^{k+1} n_i, s_1 \cdot \sum_{i=1}^{k+1} n_i)$ and $(\sum_{i=1}^{k+2} n_i, s_1 \cdot \sum_{i=1}^{k+2} n_i)$.

Note $n_1 \in [0, 1)$ denotes the length of the first, "empty" segment on which the test-function is undefined; there are $k + 1$ nonempty segments. WLOG by homogeneity of the $k$ lines that $\sum_{i=2}^{k+2} n_i = 1$.

*Remark* E.10. Test-functions of the monotone curve schema are monotone in the sense that they don't re-cross lines which they previously crossed, except for line 1 and line $k$.

*Remark* E.11. Type (I, $k$), (II, $k$) test-functions are strictly more general than test-function since we can just set $n_1 = 0$ to recover the usual test-function definition. Type (I, $k$) test-functions start on the top-most line while (II, $k$) test-functions start on the bottom-most line. Monotone Down Curves are Type (I, $k$) while Monotone Up Curves are Type (II, $k$)

**Rearrangement Lemma for Monotone Curve.**

**Definition E.12.** (Equivalence) Define two partial test-functions as *equivalent* if both test-functions have the same start-point, end-point, length, and induced activations over the $k$ lines.

We will now introduce a central Lemma in proving that a certain set of schema is complete. This Lemma is about rearranging a test-function into an equivalent test-function with a simpler schema.

**Lemma E.13.** *Given a $(k, T)$-configuration $\{s_i\}_{i \in [k]}$, suppose a type (I, $k$) test-function (resp. a type (II, $k$)) has its start point on line 1 and end point on line $k$ (resp. start point on line $k$ and end point on line 1) and spans lines $\{1, \dots, k\}$. Then there is an equivalent, monotone curve of the same length, start point, end point, and that induces the same $(B_1, \dots, B_k)$.*

*Proof.* We will only prove the conversion of a type (I, $k$) test-function to a monotone (down) curve. The conversion of a type (II, $k$) test-function to a monotone (up) curve is an analogous argument with effective slopes $s_i' = 1 - s_i, \forall i \in [k]$.

Suppose the type (I, $k$) test-function, $\mathcal{Y}$, has $M$ segments of length $n_1, \dots, n_M$, where $n_1$ is the x-coordinate of the starting point of $\mathcal{Y}$. Consider a monotone line $\mathcal{Y}^{(M)}$ that, like $\mathcal{Y}$, starts at $(n_1, s_1 \cdot n_1)$ and ends at $(\sum_{i=1}^{M} n_i, s_k \cdot \sum_{i=1}^{M} n_i)$. $\mathcal{Y}^{(M)}$ is comprised of $k + 2$ segments of length $n_1', n_2', \dots, n_k', n_{k+1}', n_{k+2}'$ (where $n_1'$ is an empty segment, denoting the coordinate of the start point), calculated from $n_1, \dots, n_M$ as follows.

$$n_1' = n_1$$

$$\forall i \in [k+1], \quad n_{i+1}' = \sum_{j \in \{2,3,\dots,M\}:n_j \in \text{Sector}_i} n_j$$

First, we must show that this "rearrangement" forms a valid test-function that meets the constraints that would be enforced on $n_1', n_2', \dots, n_k', n_{k+1}', n_{k+2}'$ described in Lemma E.6.

First, the only constraint on $n_2'$ and $n_{k+2}'$ is that they must be nonnegative.

Second, $\forall i \in [3, k+1]$, the $i$th segment traverses sector $i-1$ and must cross from $l_{i-2}$ to $l_{i-1}$. By Lemma E.6, the following constraint is satisfied iff this crossing is possible:

$$n_i' \geq (\frac{s_{i-2}}{s_{i-1}} - 1) \sum_{j=1}^{i-1} n_j'$$

We would like to show that $(n_i')_{i \in [k+2]}$ meets these constraints. First, $\forall i \in [3, k+1]$, define $j_i := \max\{j \in \{2, 3, \dots, M\} : n_j \text{ crosses Sector}_{i-1}\}$ as the index of the last segment where $\mathcal{Y}$ crosses $\text{Sector}_{i-1}$, starting on line $l_{i-2}$ and ending on line $l_{i-1}$. Then because the curve $\mathcal{Y}$ is continuous and its endpoint is on line $k$, then $j_i$ is monotone in $i$: $j_3 < j_4 < \dots < j_{k+1}$. Then,

$$
\begin{aligned}
n_i' &= \sum_{j \in \{2,3,\dots,M\}:n_j \in \text{Sector}_{i-1}} n_j \\
&\geq n_{j_i} \quad \text{just the last segment crossing Sector}_{i-1} \text{ suffices} \\
&\geq (\frac{s_{i-2}}{s_{i-1}} - 1) \sum_{j=1}^{(j_i)-1} n_j \\
&\geq (\frac{s_{i-2}}{s_{i-1}} - 1) \sum_{j=1}^{j_{i-1}} n_j \quad \text{monotonicity of } j_i \\
&\geq (\frac{s_{i-2}}{s_{i-1}} - 1) \sum_{j \leq j_{i-1} \text{ s.t. } n_j \in \bigcup_{m=1}^{i-2} \text{Sector}_m} n_j \\
&= (\frac{s_{i-2}}{s_{i-1}} - 1) \sum_{j=1}^{i-1} n_j'
\end{aligned}
$$

Thus, the monotone curve $\mathcal{Y}^{(M)}$ is a valid monotone test-function. $\mathcal{Y}^{(M)}$ has the same length as $\mathcal{Y}$ since it just rearranged the segments while preserving their length. $\mathcal{Y}^{(M)}$ starts at line 1 and ends at line $k$, and it starts at $(n_1, s_1 n_1)$ just like $\mathcal{Y}$, so it must also end at the same point as $\mathcal{Y}^{(M)}$ on line $k$. $\mathcal{Y}^{(M)}$ induces the same activations $(B_1, \dots, B_k)$ as $\mathcal{Y}$ since the rearranged segment lengths stay in their original sectors in $\mathcal{Y}^{(M)}$.

The proof for type (II, $k$) test-function is analogous. ∎

**Basis Schema.** The following is the main result of this section. It says that the following finite set of *basis* schema is complete, in the sense that any continuous test-function is equivalent to some test-function whose schema is one of the basis schema.

Each basis schema described below is indexed by integer $m \in [k]$ and a set of $m$ tuples $\{(y_1^i, y_2^i)\}_{i \in [m]} \subset \{1, \dots, k\}^m$. These $m$ tuples parameterize $m-1$ monotone-curves that, when concatenated, yield the schema. For $i \in [m-1]$, the $i$th tuple $(y_1^i, y_2^i) \in [k]^2$ indicates that the $i$th monotone curve in the schema will have start-point on line $y_1^i$ and end-point on line $y_2^i$. The concatenation of all $m-1$ monotone curves yield the basis schema.

**Corollary E.14.** *(Completeness of Basis Schema) Given a $(k, T)$-configuration $\{s_i\}_{i \in [k]}$, for any $1 \leq m \leq k$, say that the list of tuples $\{(y_1^i, y_2^i)\}_{i \in [m]} \subset \{1, \ldots, k\}^m$ is valid if they satisfy the following.*

$$y_1^m = y_2^m$$
$$\forall i \in [m-1], y_1^i \neq y_2^i$$
$$\forall i \in [m-2], y_1^i < y_2^i \implies y_2^i = y_1^{i+1} > y_2^{i+1} > y_1^i$$
$$\forall i \in [m-2], y_1^i > y_2^i \implies y_2^i = y_1^{i+1} < y_2^{i+1} < y_1^i$$
$$(y_1^1, y_2^1) = (1, k) \text{ or } (y_1^1, y_2^1) = (k, 1)$$

*For any $m \in [k]$ and valid $\{(y_1^i, y_2^i)\}_{i \in [m]}$, define the basis schema, $Y_{\{(y_1^i, y_2^i)\}_{i \in [m]}}$ as the concatenation of $m-1$ monotone curves, where for $i \in [m-1]$, the ith monotone curve has start-point on line $y_1^i \in [k]$ and end-point on line $y_2^i \in [k]$. The 1st monotone curve has start-point at the origin, $(0, 0)$.*

*Then, the set of basis schemas over all $m$ and valid $\{(y_1^i, y_2^i)\}_{i \in [m]}$ satisfying the above is complete in the following sense.*

$$\mathbb{A}(\{s_i\}_{i \in [k]}) = \bigcup_{m \in [k], valid \; \{(y_1^i, y_2^i)\}_{i \in [m]}} A(Y_{\{(y_1^i, y_2^i)\}_{i \in [m]}})$$

*Where $A(Y_{\{(y_1^i, y_2^i)\}_{i \in [m]}}) \subset [0, 1]^k$ denotes the set of $(B_1, \ldots, B_k)$ induced by any test-function of schema $Y_{\{(y_1^i, y_2^i)\}_{i \in [m]}}$. Finally, note that the number of basis schema with respect to $\{s_i\}_{i \in [k]}$ is $N_k = 2^{k-1}$.*

Refer to Figure 8 to get a sense for what the basis schema look like. Note that the y axis of the figure shows the *normalized* prefix sum. Also, Figure 9 is a depiction of the Completeness result for the case where $k = 2$.

*Proof of Corollary E.14.* Recall that $\mathbb{A}(\{s_i\}_{i \in [k]})$ and $A(Y_{\{(y_1^i, y_2^i)\}_{i \in [m]}})$ for schema $Y_{\{(y_1^i, y_2^i)\}_{i \in [m]}}$ are defined follows.

$$A(Y_{\{(y_1^i, y_2^i)\}_{i \in [m]}}) := \{(B_1(\mathcal{Y}), \ldots, B_k(\mathcal{Y})) : \mathcal{Y} \text{ continuous test-function of schema } Y_{\{(y_1^i, y_2^i)\}_{i \in [m]}} \text{ w.r.t. } \{s_i\}_{i \in [k]}\}$$
$$\mathbb{A}(\{s_i\}_{i \in [k]}) := \{(B_1(\mathcal{Y}), \ldots, B_k(\mathcal{Y})) : \mathcal{Y} \text{ continuous test-function w.r.t. } \{s_i\}_{i \in [k]}\}$$

First, because every test-function of a basis schema is a test-function,

$$\mathbb{A}(\{s_i\}_{i \in [k]}) \supset \bigcup_{m \in [k], valid \; \{(y_1^i, y_2^i)\}_{i \in [m]}} A(Y_{\{(y_1^i, y_2^i)\}_{i \in [m]}})$$

It suffices to show that the converse inclusion holds.

$$\mathbb{A}(\{s_i\}_{i \in [k]}) \subset \bigcup_{m \in [k], valid \; \{(y_1^i, y_2^i)\}_{i \in [m]}} A(Y_{\{(y_1^i, y_2^i)\}_{i \in [m]}})$$

To do this, we will prove that any arbitrary continuous test-function can be converted into an equivalent test-function (in the sense of Definition E.12), but which follows one of the basis schema. This conversion process is done via Algorithm 5, which partitions the input test-function into pieces, and then uses Lemma E.13 to convert each piece into a monotone curve, yielding a test-function of a basis schema.

More precisely, given an arbitrary test-function $\mathcal{Y}$, the main idea is to partition $\mathcal{Y}$ into pieces, such that each piece is either a partial Type (I, $\beta$) or Type (II, $\beta$) test-function for some $1 \leq \beta \leq k$. Each piece can then be rearranged into a monotone curve via an application of Lemma E.14. We use Algorithm 5 to attain $\{(y_1^\alpha, y_2^\alpha)\}_{\alpha \in [m-1]}$ which characterize the appropriate basis-schema for which there exists an equivalent test-function to $\mathcal{Y}$.

Recall that the span of a partial test-function is the set of lines $\{l_1, \ldots, l_k\}$ that it crosses at some point in its domain (where the lines are indexed from $1, \ldots, k$). The span of a test-function is a continuous subset of $[k]$ since we assume that

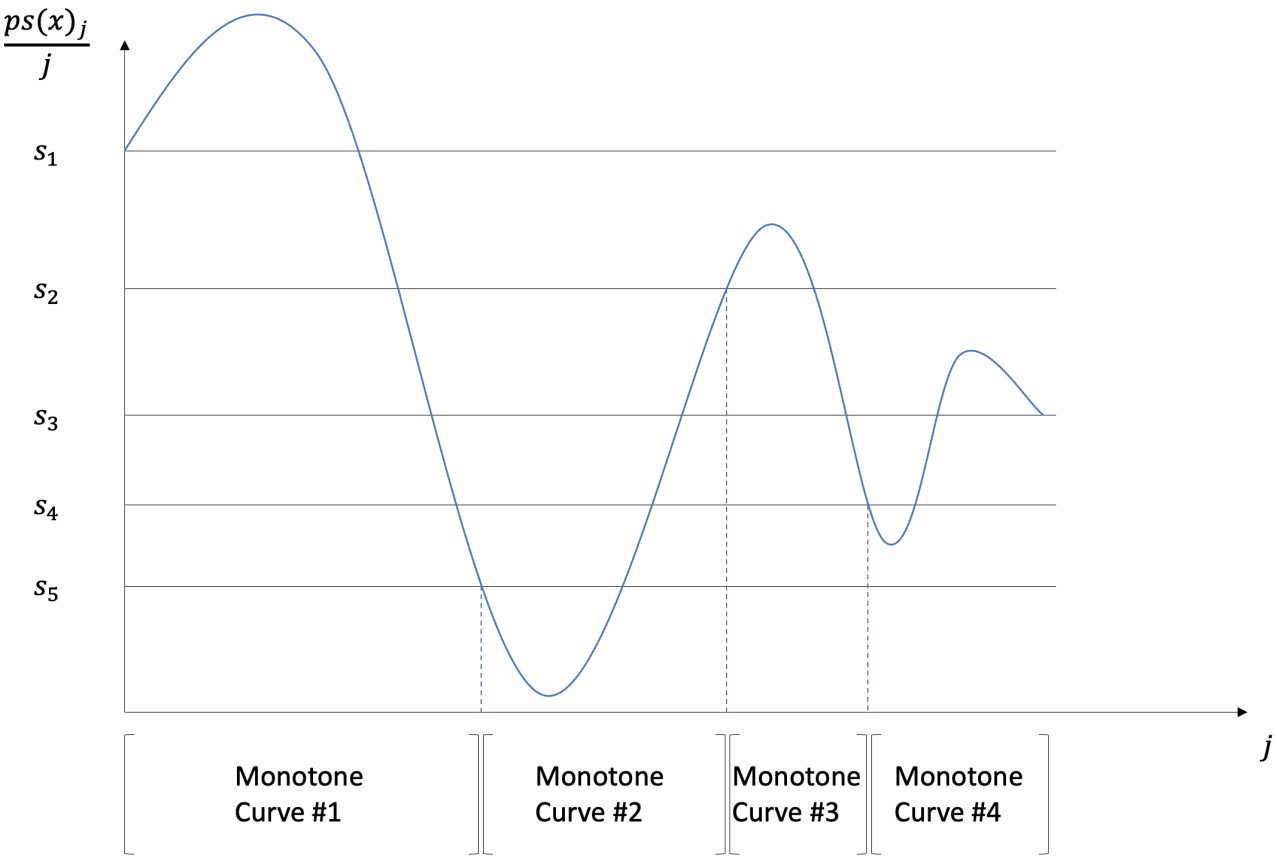

*Figure 8.* Depiction of a Basis Schema consisting of 4 monotone curves. y-axis shows the normalized prefix sum of input string $x$. x-axis shows the length of the prefix of input string $x$. The five horizontal lines correspond to five lines with slopes $\{s_1, \ldots, s_5\}$. Curve corresponds to Basis Schema with $m = 5$, $(y_1^1, y_2^1) = (1, 5), (y_1^2, y_2^2) = (5, 2), (y_1^3, y_2^3) = (2, 4), (y_1^4, y_2^4) = (4, 3)$. Finally, note that each monotone curve consists of multiple segments. The first one has 5 segments, the second one has 4 segments, the third one has 3 segments, and the fourth one has 3 segments.

$l_i$ has the highest slope $s_i$ and $\{s_i\}_{i \in [k]}$ is sorted descending. Regarding notation in Algorithm 5, $\mathrm{Span}_i$ is a tuple of two integers in $[k]^2$, which represent the smallest and largest index of lines in the span of some partial continuous test-function. $\min(\mathrm{Span}_i)$ is the smaller element in the tuple, and $\max(\mathrm{Span}_i)$ is the larger element. $\{s_i\}_{i \in [k]}$ are the $k$ slopes we fixed at the beginning, and $\mathcal{Y}(j) = s_i j$ indicates that test-function $\mathcal{Y}$ intersects the line $\mathsf{y} = s_i \mathsf{x}$ at $(j, s_i j)$. Also, when we say that $\mathcal{Y}$ intersects the line $l_i$ on $[t, 1]$ to mean that there is some $j \in [t, 1]$ where $\mathcal{Y}(j) = s_i j$, for $t \leq 1$.

Towards proving completeness of the basis schema, we will prove the following three claims. At the end, we will use these claims to argue that $\mathbb{A}(\{s_i\}_{i \in [k]}) \subset \bigcup_{m \in [k], \text{valid } \{(y_1^i, y_2^i)\}_{i \in [m]}} A(Y_{\{(y_1^i, y_2^i)\}_{i \in [m]}})$.

1. Algorithm 5 terminates.

2. Algorithm 5 returns valid YPairs $:= \{(y_1^i, y_2^i)_{i \in [m-1]}\}$ where $m := |\text{YPairs}| + 1$ with $m \leq k$, and where the valid predicate is defined in the statement of Corollary E.14.

3. MaxTB $:= \{T_i\}_{0 \leq i \leq m-1} \subset [0, 1]$ is a set of real numbers where for each $i \in \{0, 1, , \ldots, m-2\}$, $\mathcal{Y}$ restricted to the interval $[T_i, T_{i+1}]$ can be rearranged into an equivalent monotone curve by Lemma E.13.

**Proving Termination.** At each iteration $\alpha$, Algorithm 5 maintains variables $\mathrm{Span}_\alpha$, $t_{\alpha-1}$, and $b_{\alpha-1}$, which we claim satisfy the property that $\mathrm{Span}_\alpha$ holds the smallest and largest index of lines $\{1, \ldots, k\}$ which $\mathcal{Y}$ spans when $\mathcal{Y}$ is restricted to interval $[\max(t_{\alpha-1}, b_{\alpha-1}), 1]$.

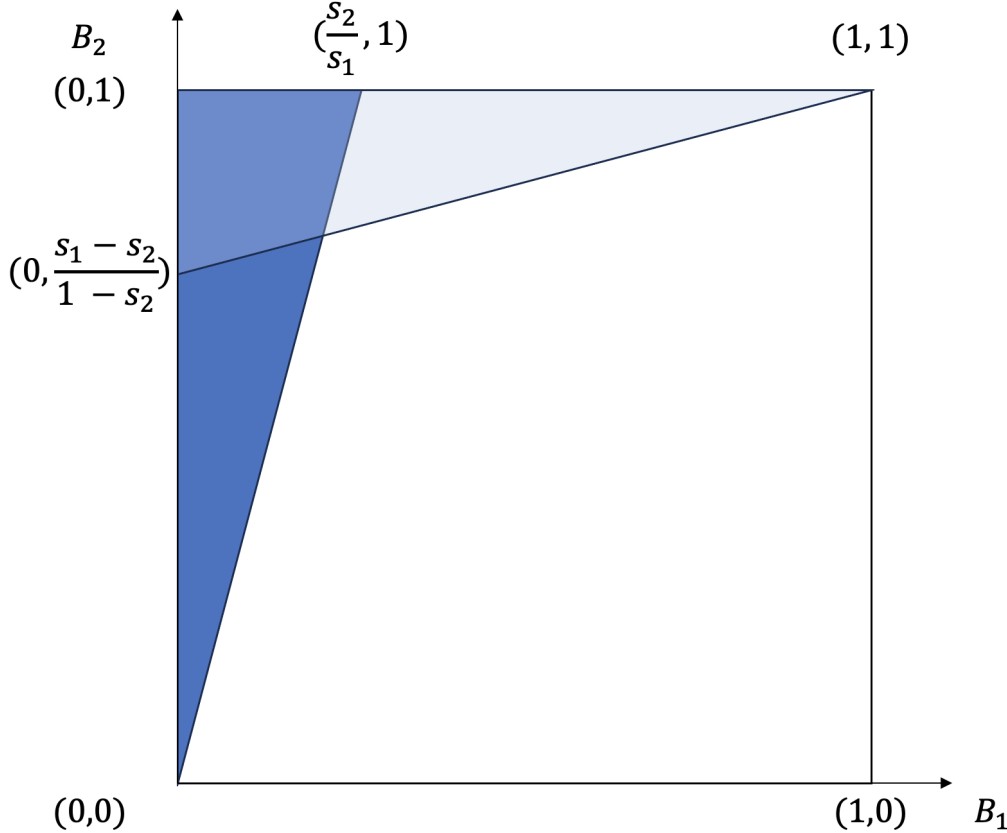

*Figure 9.* Depiction of $\mathbb{A}(\{s_i\}_{i \in [k]})$ for $k = 2$ with two slopes $s_1 > s_2$. $B_1$ is on the horizontal axis while $B_2$ is on the vertical axis. When $k = 2$, there are only two basis schema: a single monotone (up) curve and a single monotone (down) curve. The dark blue triangle with vertices $\{(0, 0), (0, 1), (\frac{s_2}{s_1}, 1)\}$ is the set of activations induced by test-functions of the monotone (down) curve basis schema. The light blue triangle with vertices $\{(1, 1), (0, 1), (0, \frac{s_1 - s_2}{1 - s_2})\}$ is the set of activations induced by test-functions of the monotone (up) curve. The completeness result says that the union of these two triangles equals $\mathbb{A}(\{s_1, s_2\})$.

First, suppose the span of $\mathcal{Y}$ is a single line (recall that the span of $\mathcal{Y}$ is always at least one line, since by Lemma E.2, we let the endpoint of any test-function be on some line). Then, Algorithm 5 initializes $\text{Span}_1$ to be the span of $\mathcal{Y}$, when restricted to interval $[0, 1]$. Algorithm 5 computes $y_1^1, y_2^1$, then terminates. Our claim holds.

For $\mathcal{Y}$ that span 2 or more lines, we will prove via induction that for each iteration $\alpha$, $\text{Span}_\alpha$, $t_{\alpha-1}$, and $b_{\alpha-1}$ satisfy our claim. As a base case, $\text{Span}_1 = \text{Span}(\mathcal{Y})$ satisfies our claim. As the inductive step, suppose $\text{Span}_\alpha$ holds the smallest and largest index of lines $\{1, \ldots, k\}$ which $\mathcal{Y}$ spans, when restricted to interval $[\max(t_{\alpha-1}, b_{\alpha-1}), 1]$. Then, in iteration $\alpha$, the variables $t_\alpha$ and $b_\alpha$ exist (recall that a continuous test-function only intersects the lines at finitely many points, so the max is well defined). We can never have that $t_\alpha = b_\alpha$ as long as $\min(\text{Span}_{\alpha-1}) < \max(\text{Span}_{\alpha-1})$, which must be true at the start of iteration $\alpha$, otherwise the algorithm would have terminated in iteration $\alpha - 1$. Thus, either $t_\alpha > b_\alpha$ or $t_\alpha < b_\alpha$. If $t_\alpha > b_\alpha$, then the last point where $\mathcal{Y}$ intersects line $\min(\text{Span}_\alpha)$ is later than the last point where $\mathcal{Y}$ intersects line $\max(\text{Span}_\alpha)$. On the interval $[\max(t_\alpha, b_\alpha), 1]$, $\mathcal{Y}$ never intersects line $\max(\text{Span}_\alpha)$ again. Thus, the smallest index of any line which $\mathcal{Y}$ intersects on the interval $[\max(t_\alpha, b_\alpha), 1]$ is $\min(\text{Span}_\alpha)$. The largest index of any line which $\mathcal{Y}$ intersects on the interval $[\max(t_\alpha, b_\alpha), 1]$ must be less than $\max(\text{Span}_\alpha)$ as $t_\alpha > b_\alpha$. Thus, the largest index line which $\mathcal{Y}$ intersects on the interval $[\max(t_\alpha, b_\alpha), 1]$ is given by the expression, $\max\{i < \max(\text{Span}_\alpha) : i \in [k], \mathcal{Y} \text{ intersects line } l_i \text{ on } [\max(t_\alpha, b_\alpha), 1]\}$. This shows that line 9 correctly sets $\text{Span}_{\alpha+1}$, completing the induction. A similar correctness argument can be made for the case where $t_\alpha < b_\alpha$. This proves that for all $\alpha$, $\text{Span}_\alpha$ correctly holds the smallest and largest index of lines $\{1, \ldots, k\}$ which $\mathcal{Y}$ spans when it is restricted to interval $[\max(t_{\alpha-1}, b_{\alpha-1}), 1]$.

---

**Algorithm 5** Decompose-Into-Monotone-Curves

---

1: **Initialize:** $\text{Span}_1 \leftarrow \text{Span}(\mathcal{Y})$, $t_0 \leftarrow 0$, $b_0 \leftarrow 0$
2: $\text{YPairs} \leftarrow [\,]$     % *List of all* $(y_1^\alpha, y_2^\alpha)$ *pairs*
3: $\text{MaxTB} \leftarrow [0\,]$     % *List of all* $\max(t_\alpha, b_\alpha)$ *values*
4: **for** $\alpha \in \{1, 2, \ldots, k-1\}$ **do**
5:     $t_\alpha \leftarrow \max\Big\{ j \geq \max(t_{\alpha-1}, b_{\alpha-1}) : \mathcal{Y}(j) = s_{\min(\text{Span}_\alpha)}\, j \Big\}$
6:     $b_\alpha \leftarrow \max\Big\{ j \geq \max(t_{\alpha-1}, b_{\alpha-1}) : \mathcal{Y}(j) = s_{\max(\text{Span}_\alpha)}\, j \Big\}$
7:     **if** $t_\alpha > b_\alpha$ **then**
8:        $y_1^\alpha \leftarrow \max(\text{Span}_\alpha)$     and     $y_2^\alpha \leftarrow \min(\text{Span}_\alpha)$
9:        $\text{Span}_{\alpha+1} \leftarrow \Big( \min(\text{Span}_\alpha),\ \max\{ i < \max(\text{Span}_\alpha) : i \in [k],\ \mathcal{Y} \text{ intersects line } l_i \text{ on } [\max(t_\alpha, b_\alpha), 1] \} \Big)$
10:     **else**
11:        $y_1^\alpha \leftarrow \min(\text{Span}_\alpha)$     and     $y_2^\alpha \leftarrow \max(\text{Span}_\alpha)$
12:        $\text{Span}_{\alpha+1} \leftarrow \Big( \min\{ i > \min(\text{Span}_\alpha) : i \in [k],\ \mathcal{Y} \text{ intersects line } l_i \text{ on } [\max(t_\alpha, b_\alpha), 1] \},\ \max(\text{Span}_\alpha) \Big)$
13:     **end if**
14:     $\text{YPairs} \leftarrow \text{YPairs} \cup \{(y_1^\alpha, y_2^\alpha)\}$
15:     $\text{MaxTB} \leftarrow \text{MaxTB} \cup \{\max(t_\alpha, b_\alpha)\}$
16:     **if** $\max(\text{Span}_\alpha) = \min(\text{Span}_\alpha)$ **then**
17:        **break**
18:     **end if**
19: **end for**
20: **return** $(\text{YPairs}, \text{MaxTB})$

---

For all $\alpha$, we have that $\max(\text{Span}_\alpha) - \min(\text{Span}_\alpha) \leq \max(\text{Span}_{\alpha-1}) - \min(\text{Span}_{\alpha-1}) - 1$. Thus, the Algorithm terminates after at most $|\text{Span}(\mathcal{Y})| - 1 \leq k - 1$ iterations since the largest that $|\text{Span}(\mathcal{Y})|$ can be is $k$, when $\mathcal{Y}$ spans all $k$ lines (by $|\text{Span}(\mathcal{Y})|$, we mean the number of lines which $\mathcal{Y}$ spans). In particular, if we let $m := |\text{YPairs}| + 1$ returned by Algorithm 5, then $m \leq |\text{Span}(\mathcal{Y})| \leq k$.

**Proving Validity of YPairs.** Towards the second claim, first note that if $t_\alpha > b_\alpha$ in iteration $\alpha$ of Algorithm 5 and the minimum and maximum of $\text{Span}_{\alpha+1}$ are not equal, then in the next iteration $\alpha + 1$, it will be the case that $b_{\alpha+1} > t_{\alpha+1}$. This is because the last point where $\mathcal{Y}$ crosses line $\min(\text{Span}_\alpha)$ has x-coordinate $t_\alpha$, by line 5 in iteration $\alpha$. However, by lines 8 and 9 in iteration $\alpha$, it must also be true that in the next iteration $\alpha + 1$, $t_{\alpha+1}$, defined in line 5 of iteration $\alpha + 1$, equals $\max(t_\alpha, b_\alpha) = t_\alpha$. Since we cannot have $t_{\alpha+1} = b_{\alpha+1}$ (argued previously) and $t_{\alpha+1}, b_{\alpha+1}$ are both at least $\max(t_\alpha, b_\alpha)$, we have that $b_{\alpha+1} > t_{\alpha+1} = t_\alpha$. By lines 11 and 12 in iteration $\alpha + 1$, Algorithm 5 will set $y_1^\alpha > y_2^{\alpha+1} > y_1^{\alpha+1} = y_2^\alpha$. Analogously, if $t_\alpha < b_\alpha$ and $\text{Span}_{\alpha+1}$ has a distinct minimum and maximum element, then $b_{\alpha+1} < t_{\alpha+1}$, and $y_1^\alpha < y_2^{\alpha+1} < y_1^{\alpha+1} = y_2^\alpha$. This proves that the algorithm returns YPairs which is a set of at most $m - 1 \leq |\text{Span}(\mathcal{Y})| - 1 \leq k - 1$ pairs $\{(y_1^\alpha, y_2^\alpha)\}_{\alpha \in [m-1]}$ where each pair has a distinct minimum and maximum and satisfies the properties in Corollary E.14, except potentially for the property that $(y_1^1, y_2^1) = (1, k)$ or $(y_1^1, y_2^1) = (k, 1)$.

The last issue about ensuring YPairs satisfies the property that $(y_1^1, y_2^1) = (1, k)$ or $(y_1^1, y_2^1) = (k, 1)$ is simple to deal with. The issue arises when $\text{Span}(\mathcal{Y}) \subsetneq [k]$, so that $\mathcal{Y}$ does not span all $k$ lines. However, we can note that any test-function which does not span all $k$ lines can be thought of as part of a schema which does span all $k$ lines, except that the segments of the schema which cross lines in $[k] - \text{Span}(\mathcal{Y})$ are the first segments of the schema, and their lengths are *set equal to 0* for the particular case of $\mathcal{Y}$ (which is a valid setting of the lengths of the first segments, per Lemma E.6). Thus, though $\mathcal{Y}$ does not span $k$ lines, it can be thought of as belonging to a schema which does. We will discuss this more at the end.

**Converting Each Part into a Monotone Curve, to Convert $\mathcal{Y}$ into a Continuous Test-function of a Basis Schema.** Third, with $\text{MaxTB} := \{T_i\}_{0 \leq i \leq m-1}$, for each $\alpha \in [m-1]$, the pair $(y_1^\alpha, y_2^\alpha)$ generated by an iteration of Algorithm 5 corresponds to the interval $[T_{\alpha-1}, T_\alpha]$, in the sense that

$$(T_{\alpha-1}, \mathcal{Y}(T_{\alpha-1})) = (T_{\alpha-1}, T_{\alpha-1} \cdot s_{y_1^\alpha})$$
$$(T_\alpha, \mathcal{Y}(T_\alpha)) = (T_\alpha, T_\alpha \cdot s_{y_2^\alpha})$$

We have already argued that the span of the partial test-function given by restricting $\mathcal{Y}$ to $[T_\alpha, 1]$ is given exactly by $\mathrm{Span}_\alpha$, where either $y_1^\alpha = \min(\mathrm{Span}_\alpha)$ and $y_2^\alpha = \max(\mathrm{Span}_\alpha)$; or $y_1^\alpha = \max(\mathrm{Span}_\alpha)$ and $y_2^\alpha = \min(\mathrm{Span}_\alpha)$. Thus, the restriction of $\mathcal{Y}$ on interval $[T_{\alpha-1}, T_\alpha]$ is a Type (I, $\max(\mathrm{Span}_\alpha) - \min(\mathrm{Span}_\alpha) + 1$) or Type (II, $\max(\mathrm{Span}_\alpha) - \min(\mathrm{Span}_\alpha) + 1$) partial test-function, for which Lemma E.13 applies.

By Lemma E.13, the restriction of $\mathcal{Y}$ on interval $[T_{\alpha-1}, T_\alpha]$ is equivalent to a monotone curve of span $\mathrm{Span}_\alpha$, which has start-point $(T_{\alpha-1}, s_{y_1^\alpha} \cdot T_{\alpha-1})$ and end-point $(T_\alpha, s_{y_2^\alpha} \cdot T_\alpha)$. The monotone curve will have the same start-point, end-point, length, and induced activations as $\mathcal{Y}$ restricted[1] to $[T_{\alpha-1}, T_\alpha]$. Thus, we can construct such a monotone curve for each $\alpha \in [m-1]$ and concatenate these monotone curves together. The resulting test-function is equivalent to $\mathcal{Y}$, as each individual monotone curve is equivalent to the corresponding restriction of $\mathcal{Y}$.

We claim the resulting test-function is of one of the basis schema described in Corollary E.14. This essentially follows from the validity of YPairs outputted by Algorithm 5, which we justified in the previous section. To reiterate a key point, if $\mathrm{Span}(\mathcal{Y}) = [k]$, then YPairs satisfies the conditions of Corollary E.14 exactly. On the other hand, if $\mathrm{Span}(\mathcal{Y}) \subsetneq [k]$ is a strict (continuous) subset of $[k]$, then still we can view the concatenation of monotone curves above as a test-function of a basis schema in Corollary E.14. To see this, we observe that for any such $\mathcal{Y}$, there is a basis schema $Y$ such that if we set the lengths of the segments of the monotone curves in schema $Y$ whose span is a strict superset of $\mathrm{Span}(\mathcal{Y})$ to $0$, then the remaining monotone curves are given by YPairs outputted by Algorithm 5 on input $\mathcal{Y}$. This basis schema $Y$ would be given as follows. Given YPairs outputted by Algorithm 5 on input $\mathcal{Y}$, we pre-pend either one or two pairs to YPairs. If $\min(\mathrm{Span}(\mathcal{Y})) = 1$ or $\max(\mathrm{Span}(\mathcal{Y})) = k$, we pre-pend a single pair to YPairs: $(k, 1)$ in the first case and $(1, k)$ in the second case. If $\min(\mathrm{Span}(\mathcal{Y})) > 1$ and $\max(\mathrm{Span}(\mathcal{Y})) < k$, we pre-pend two pairs $(1, k), (k, \min(\mathrm{Span}(\mathcal{Y})))$ to YPairs. The resulting list of pairs with one or two pairs pre-pended, which we call $L$, will have at most $k-1$ pairs total. $L$ will satisfy all requirements in Corollary E.14. As we argued that we can simply set the lengths of the monotone curves corresponding to the pre-pended pairs in $L$ to $0$, the final test-function we attained by concatenating all the monotone curves together in the previous paragraph is of the basis schema corresponding to the list of pairs $L$.

Thus, the concatenation of monotone curves described above, based on the output by Algorithm 5, is of some basis schema specified in Corollary E.14. This proves that any continuous test-function $\mathcal{Y}$ is equivalent (in particular, induces the same activations) to some test-function of a basis schema specified in Corollary E.14. This implies that $\mathbb{A}(\{s_i\}_{i \in [k]}) \subset \bigcup_{m \in [k], \text{valid } \{(y_1^i, y_2^i)\}_{i \in [m]}} A(Y_{\{(y_1^i, y_2^i)\}_{i \in [m]}})$, as desired.

Finally, regarding the number of basis schemas, we can first partition up the schema based on which of the $k$ lines the end-point is on. Let $f(k, i)$ be the number of basis schema w.r.t. a configuration of $k$ slopes whose end-point is on line $i \in [k]$. We claim that $f(k, i) = \binom{k-1}{i-1}$, from which it follows that the total number of basis schema is $\sum_{i \in [k]} \binom{k-1}{i-1} = 2^{k-1}$.

For $i \in [k]$, to see that $f(k, i) = \binom{k-1}{i-1}$, let $I_i := [k+1] - \{i, i+1\}$. Consider all permutations $\sigma(I_i)$ which contain $\{1, 2, \ldots, i-1\}$ and $\{k+1, \ldots, i+2\}$ as a subsequence. We claim there is a one-to-one correspondence between any such $\sigma(I_i)$ and a basis schema $Y_{\{(y_1^i, y_2^i)\}_{i \in [m]}}$. Given $Y_{\{(y_1^i, y_2^i)\}_{i \in [m]}}$, let $g(\{(y_1^i, y_2^i)\}_{i \in [m]})$ be a permutation of $I_i$ where the ordering of element $j \in I_i$ in the permutation is the relative ordering of sector $j$ based on the last time where schema $Y_{\{(y_1^i, y_2^i)\}_{i \in [m]}}$ passed through sector $j$. The mapping is injective because for any two $\{(y_1^i, y_2^i)\}_{i \in [m]} \neq \{((y_1^i)', (y_2^i)')\}_{i \in [m']}$, the first pair such that $(y_1^i, y_2^i) \neq ((y_1^i)', (y_2^i)')$ will be such that $y_2^i \neq (y_2^i)'$, and one of these schema will have visited $\mathrm{Sector}_{y_2^i}$ or $\mathrm{Sector}_{(y_2^i)'}$ for the last time while the other will return to it later. The surjectivity of the mapping can be checked easily.

There are $i-1$ elements in $\{1, 2, \ldots, i-1\}$ and $k-1$ elements in $I_i$, so there are $\binom{k-1}{i-1}$ such permutations. ■

**Corollary E.15.** *Given a $(k, T)$-configuration $\{s_i\}_{i \in [k]}$, each basis test-function schema specified in Corollary E.14 crosses the $k$ lines of slopes $\{s_i\}_{i \in [k]}$ a total of at most $k^2$ times.*

*Proof.* By Corollary E.14, each basis schema consists of at most $k$ monotone curves. The $i$th monotone curve has a span at most the span of that of the previous monotone curve minus 1, so since the first monotone curve spans $k$ lines, there will be at most $k$ monotone curves. Each monotone curve intersects the $k$ lines each once, except for the two lines at the top and bottom of its span. However, these can be reduced to one intersection when you concatenate alternating monotone lines

---

[1] Here we are applying Lemma E.13 on restrictions of test-functions where the end-point is not at 1, but we can do this due to the homogeneity of the setup

together as such.

Thus, the number of times the test-functions of any basis schema crosses the $k$ lines is $\sum_{i=1}^{k} i = \frac{k(k+1)}{2} \leq k^2$. $\blacksquare$

**Convexity of Schema.**

**Lemma E.16.** *(Convexity of activation set of test-functions of a schema) Given a $(k, T)$-configuration, $\{s_i\}_{i \in [k]}$, consider any schema continuous test-functions. Suppose the schema specifies $M$ segments. Let $A^{(M)}$ be the set of valid $(n_1, \ldots, n_M)$ of segment lengths with $\sum_{i=1}^{M} n_i = 1$, where a valid setting of $(n_1, \ldots, n_M)$ satisfies the constraints described in Lemma E.6. Then $A^{(M)}$ is a convex set of dimension $M - 1$. Moreover, $A^{(M)}$ is a simplex.*

*Proof.* $A^{(M)}$ is the intersection of the linear subspace over $(n_1, \ldots, n_M)$ given by $\sum_{i=1}^{M} n_i = 1$ along with $M$ halfspaces of the form in Lemma E.6:

$$\forall i \in [M], n_i \geq \frac{p_i}{q_i} \sum_{j=1}^{i-1} n_j$$

Each $p_i, q_i$ will be determined from the slopes of the lines that segment $i$ first intersects and last intersects. By Lemma E.6, there are three cases:

**Case 1: Segment $i$ Crosses From Line $j$ to Line $j + 1$.** Then, $n_i \geq (\frac{s_j}{s_{j+1}} - 1) \sum_{j=1}^{i-1} n_j$.

**Case 2: Segment $i$ Crosses From Line $j + 1$ to Line $j$.** Then, $n_i \geq (\frac{1 - s_{j+1}}{1 - s_j} - 1) \sum_{j=1}^{i-1} n_j$.

**Case 3: Segment $i$ Crosses From Line $j$ to Line $j$.** Then, $\frac{p_i}{q_i} = 0$.

First, since $A^{(M)}$ is the intersection of convex sets, it is convex.

Second, regarding dimension of $A^{(M)}$, since all elements $\{s_i\}_{i \in [k]}$ of the configuration are distinct and in $(0, 1)$, each of the $M$ halfspaces acts on a different subset of the variables $\{n_i\}_{i \in [M]}$:

$$\{n_1\}, \{n_1, n_2\}, \{n_1, n_2, n_3\}, \ldots, \{n_1, \ldots, n_{M-1}\}, \{n_1, \ldots, n_M\}$$

In particular, no combination of inequality constraint can form an equality constraint which implies that the intersection of the $M$ halfspaces has dimension $M$. Thus, $A^{(M)}$ has dimension $M - 1$ due to the additional constraint $\sum_{i=1}^{M} n_i = 1$.

Third, $A^{(M)}$ can be thought of as a polytope over $M - 1$ variables, once we substitute in $n_M = 1 - \sum_{i=1}^{M-1} n_i$, while it has exactly $M$ faces given by the $M$ halfspaces, with $n_M = 1 - \sum_{i=1}^{M-1} n_i$ substituted into the inequalities. The $M$ faces of $A^{(M)}$ remain distinct even after the substitution since the only face whose linear inequality included $n_M$ will have a bias term of 1 after the substitution, while the other faces' linear inequality remain homogeneous. Since $A^{(M)}$ has $M$ faces and has dimension $M - 1$, it is a simplex.

$\blacksquare$

### E.4. Lemmas for Margin of Point in a Polytope.

**Average of Vertices of a Polytope.**  As a motivation for the following definition, recall that the centroid of a $d$-dimensional simplex is the average of its $d + 1$ vertices. Informally speaking, the centroid has the nice property that it is far from each face of the simplex. This property is useful for our proof, though we will need to extend this average of vertices notion to general convex polytopes.

**Definition E.17.** (Average of Vertices) Given a convex polytope $P$ of $N$ vertices $v_1, \ldots, v_N$, we will define the average-of-vertices of $P$ as $c = \frac{1}{N} \sum_{i \in [N]} v_i \in P$.

Note that for a general convex polytope $P$, the average of its vertices will not, in general, be its centroid.

**Margin of Average of Vertices of Polytope with $\mathrm{poly}(T) \cdot \mathrm{poly}(K)$ Precision Faces.**

**Definition 6.10** (Margin). Given a linear inequality $L$ over $M$ variables and a point $x \in \mathbb{R}^M$, define $L(x)$ as the difference between the left-hand-side and right-hand-side of the inequality when the coordinates of $x$ are plugged into $L$. Let $L(x) = 0 \iff x$ satisfies the inequalities with equality. We say $L(x)$ is the margin of $x$ for $L$.

Often, we will mention the notion of margin in the context of a polytope $P$, where we are interested in the margin of a point $x \in P$ onto a face of $P$. For any face $F$ of $P$, $F$ will be defined as the boundary of the halfspace given by some linear inequality $L$. In this case, WLOG, we will define the margin of any point $x$ in $P$ to $F$ to be $L(x)$. Moreover, we will define $L$ so that $L(x)$ is non-negative for $x \in P$. As such, we will sometimes refer to the margin of a point $x \in P$ onto a face of $P$ as the "positive" margin to emphasize this point.

We start with the following Lemma.

**Lemma E.18.** *Consider a nonempty $(M-1)$-dimensional convex polytope $P \subset \mathbb{R}^{M-1}$. Suppose the faces of $P$ consists of $N$ halfspaces over variables $\{n_i\}_{i \in [M-1]}$, where each halfspace is given by a linear inequality with integer coefficients of magnitude at most $p$. For $j \in [N]$, define $L_j$ as the linear inequality for the jth face.*

*Then, for any $j \in [N]$, for any vertex $x$ of $P$ which does not lie on the jth face of $P$, then the positive margin $L_j(x)$ is lower bounded as follows.*

$$L_j(x) \geq \frac{1}{(\sqrt{M}p)^M}$$

*Proof.* For each $j \in [N]$, represent the linear inequality $L_j$ as a vector in $v_j \in [-p,p]^M$ such that $\forall x' \in \mathbb{R}^{M-1}, v_j^\top \begin{pmatrix} x' \\ -1 \end{pmatrix}$ is the margin of $x$ on the constraint given by $L_j$. In addition, in our definition of the vector representations $v_j$, we want to ensure that the vectors $v_j$ are such that $\forall j \in [N], v_j^\top \begin{pmatrix} x \\ -1 \end{pmatrix} > 0$ for all vertices $x$ which do not lie on the $j$th face of $P$. This is always possible due to the convexity of the polytope, which is contained in the intersection of the halfspaces defined by each of its faces.

As an example of vector representations of inequalities,

$$n_1 + 2n_2 + 3n_3 > 4 \iff (1,2,3,4)$$
$$2n_1 - 3n_2 + 10n_3 \leq -9 \iff (-2,3,-10,9)$$

WLOG, we will prove the statement for the 1st face of $P$. Pick any vertex $x$ which does not lie on the 1st face of $P$ (i.e. $|L_1(x)| > 0$). $x$ is the intersection of $M-1$ distinct faces of $P$, whose indices we denote as $\{j_i\}_{i \in [M-1]} \subset [N] - \{1\}$. Let $\overline{A} \in [-p,p]^{M \times M}$ be a matrix such that for all $i \in [M-1]$, the $i$th row of $\overline{A}$ is $v_{j_i}$ the vector representation for $L_{j_i}$. Let the $M$th row be $v_1$, the vector representation of $L_1$. Write $\overline{A}$ in block form as:

$$\overline{A} = \begin{bmatrix} A & b \\ c^\top & d \end{bmatrix}$$

where $A \in [-p,p]^{(M-1) \times (M-1)}$ and $b, c \in [-p,p]^{M-1}$ and $d \in [-p,p]$, so that $\begin{pmatrix} c \\ d \end{pmatrix}$ is the vector representation of $L_1$.

We have that $\forall i \in [M-1], L_{j_i}(x) = (\overline{A}_i)^\top \begin{pmatrix} x \\ -1 \end{pmatrix}$, where $\overline{A}_i$ denotes the $i$th row of $\overline{A}$. Since $x$ lies on all the faces whose indices are in $\{j_i\}_{i \in [M-1]}$,

$$\forall i \in [M-1], L_{j_i}(x) = 0$$
$$\implies Ax = b$$
$$\implies x = A^{-1}b$$
$$L_1(x) = (\overline{A}_M)^\top \begin{pmatrix} A^{-1}b \\ -1 \end{pmatrix}$$
$$= \begin{pmatrix} c \\ d \end{pmatrix}^\top \begin{pmatrix} A^{-1}b \\ -1 \end{pmatrix}$$
$$= c^\top A^{-1}b - d$$

Next, we note that

$$|\overline{A}| = \left| \begin{bmatrix} A & b \\ 0 & d - c^\top A^{-1}b \end{bmatrix} \right|$$
$$= |A|(d - c^\top A^{-1}b)$$

Thus,

$$L_1(x) = \left| \frac{|\overline{A}|}{|A|} \right| > 0$$

where the margin is strictly positive since $x_1$ does not lie on face 1. Its margin has a minimum value of $\left| \frac{1}{|A|} \right| \geq \frac{1}{(\sqrt{M}p)^M}$ since the numerator $|\overline{A}|$ is an integer while the denominator is upper bounded by $(\sqrt{M}p)^M$. The upper bound on the determinant of $A$ is by the Geometric-Mean-Quadratic-Mean inequality on the eigenvalues of $A$, where $|A| \leq [\frac{1}{M-1}||A||_F^2]^{\frac{M-1}{2}} \leq (\sqrt{M} \cdot p)^M$.

Finally, this lower bound holds for any other face of $P$ and any vertex of $P$ which does not lie on that face by an analogous argument. ∎

We now provide a Lemma which can improve the margin lower bound in the case that we know some additional information about the aforementioned matrix $A$ defined in the setting of Lemma E.18.

**Lemma E.19.** *Suppose* $(T, K), (T', K') \in \mathbb{N}^2$ *are such that* $K \leq T^2$ *and* $K' \leq (T')^2$. *WLOG, suppose* $T \geq T'$. *Suppose* $M \in \mathbb{N}$ *such that* $\max(K, K') \leq M \leq 2(K + K')$, *and suppose constants* $c, d = O(1)$. *Suppose matrix* $A \in \mathbb{Z}^{M-1 \times M-1}$ *such that:*

1. $\forall 1 \leq i \leq \min(cK, M-1), A_i \in \{-O(T^d), \ldots, O(T^d)\}^{M-1}$

2. $\forall \min(cK, M-1) \leq i \leq M-1, A_i \in \{-O((T')^d), \ldots, O((T')^d)\}^{M-1}$

*Then* $|A| \leq O(T^{O(K)} \cdot (T')^{O(K')})$.

*Proof.* If $T, T' \geq 2$, by the homogeneity of the determinant, we can factor out a factor of $O(T^d)$ from each of the first $\min(cK, M-1)$ rows of $A$ and a factor of $O((T')^d)$ from the remaining rows of $A$.

$$|A| \leq (O(T^d))^{\min(cK,M-1)} \cdot (O((T')^d))^{M-1-\min(cK,M-1)}|\tilde{A}|$$

Where $\tilde{A}$ is an $(M-1) \times (M-1)$ matrix such that the largest magnitude of any entry in $\tilde{A}$ is 1. Since $T, T' \geq 2$, then there is a universal constant $1 < \epsilon = O(1)$ such that:

$$|A| \leq (T^{\epsilon \cdot d})^{\min(cK, M-1)} \cdot ((T')^{\epsilon \cdot d})^{M-1-\min(cK, M-1)} |\tilde{A}|$$

By the same Quadratic-Mean-Geometric-Mean argument in Lemma E.18,

$$|\tilde{A}| \leq \left(\frac{1}{M-1}||\tilde{A}||_F^2\right)^{\frac{M-1}{2}}$$
$$\leq M^{\frac{M}{2}}$$

Suppose that $K \leq K'$. Then $M \leq 4K'$ so that $(T^{\epsilon d})^{\min(cK, M-1)} \cdot ((T')^{\epsilon d})^{M-1-\min(cK, M-1)} \leq T^{\epsilon cdK} \cdot (T')^{4\epsilon dK'}$. Meanwhile, since $M \leq 4K'$ and $K' \leq (T')^2$, then $M^{\frac{M}{2}} \leq (4(T')^2)^{2K'}$, so that $|A| \leq (T^{\epsilon d})^{\min(cK, M-1)} \cdot ((T')^{\epsilon d})^{M-1-\min(cK, M-1)} M^{\frac{M}{2}} \leq T^{\epsilon cdK} \cdot (T')^{4\epsilon dK'} \cdot (4(T')^2)^{2K'} \leq O(T^{O(K)} \cdot (T')^{O(K')})$.

Now suppose that $K \geq K'$. Then $M \leq 4K$ so that $(T^{\epsilon d})^{\min(cK, M-1)} \cdot ((T')^{\epsilon d})^{M-1-\min(cK, M-1)} \leq T^{\epsilon cdK} \cdot (T')^{4\epsilon dK} \leq T^{\epsilon cdK} \cdot T^{4\epsilon dK}$. Meanwhile, since $M \leq 4K$ and $K \leq T^2$, then $M^{\frac{M}{2}} \leq (4T^2)^{2K}$, so that $|A| \leq (T^{\epsilon d})^{\min(cK, M-1)} \cdot ((T')^{\epsilon d})^{M-1-\min(cK, M-1)} M^{\frac{M}{2}} \leq T^{\epsilon cdK} \cdot T^{4\epsilon dK} \cdot (4T^2)^{2K} \leq O(T^{O(K)} \cdot (T')^{O(K')})$.

In the case $T = 1$ and $T' = 1$, then $M = O(1)$, so $|A| \leq O(1)$. In the case $T \geq 2$ but $T' = 1$, then $K' = (T')^2 = 1$, and following a similar argument as the case where $T, T' \geq 2$ and $K \geq K'$ yields the desired bound. ■

As a consequence, we have the following finer-grained version of Lemma E.18.

**Lemma E.20.** *Suppose $\alpha \in \mathbb{N}$, $(T, K), (T', K') \in \mathbb{N}^2$ are such that $T^K \leq \alpha$ and $(T')^{K'} \leq \alpha$ and $K \leq T^2$ and $K' \leq (T')^2$. WLOG, suppose $T \geq T'$. Suppose $M \in \mathbb{N}$ such that $\max(K, K') \leq M \leq 2(K + K')$. Suppose $d_1, d_2 = O(1)$.*

*Consider a nonempty $(M-1)$-dimensional convex polytope $P \subset \mathbb{R}^{M-1}$. Suppose the faces of $P$ consists of $N$ halfspaces over variables $\{n_i\}_{i \in [M-1]}$. Suppose that there is a subset of at most $d_1 \cdot K$ faces of $P$ such that each face of the subset is given by a linear inequality whose coefficients have precision at most $O(T^{d_2})$ while the faces not in the subset all have precision at most $O((T')^{d_2})$. For $j \in [N]$, define $L_j$ as the linear constraint for the jth face.*

*Then, for any $j \in [N]$, for any vertex $x$ of $P$ which does not lie on the jth face of $P$, then the positive margin $L_j(x)$ is lower bounded as follows.*

$$L_j(x) \geq \Omega\left(\frac{1}{\alpha^{O(1)}}\right)$$

*Proof.* We follow the proof of Lemma E.18 up to the step where we've defined $A$ and $\overline{A}$ and where we deduce that for any vertex $x$ of $P$ which does not lie on the first face of $P$,

$$L_1(x) = |\frac{|\overline{A}|}{|A|}| > 0$$

Because $A$ satisfies the pre-conditions Lemma E.19 once we swap of rows appropriately, then $|A| \leq O(T^{O(K)}(T')^{O(K')}) \leq O(\alpha^{O(1)})$. Thus, $L_1(x) \geq \Omega(\frac{1}{\alpha^{O(1)}})$. An analogous argument holds for all the other faces of $P$. ■

The following corollary applies Lemma E.20 to our setting of interest.

**Corollary E.21.** *Suppose $\alpha \in \mathbb{N}$, $(T, K), (T', K') \in \mathbb{N}^2$ are such that $T^K \leq \alpha$ and $(T')^{K'} \leq \alpha$ and $K \leq T^2$ and $K' \leq (T')^2$. WLOG, suppose $T \geq T'$. Suppose $M \in \mathbb{N}$ such that $\max(K, K') \leq M \leq 2(K + K')$. Suppose $d_1, d_2 = O(1)$.*

*Denote the coordinates in an $M$-dimensional Euclidean space $E$ as $(n_1, \ldots, n_M)$. Consider a nonempty $(M-1)$-dimensional polytope $P \subset E$, such that $\mathrm{cl}(P)$ has vertices $V$, with $|V| \geq M$, and $N$ faces. Suppose $P$ is contained in the*

$(M-1)$-*dimensional subspace given by* $\sum_{i\in[M]} n_i = 1$. *Suppose there is a subset of at most* $d_1 \cdot K$ *faces of* $P$ *such that each face of the subset is given by a linear inequality whose coefficients have precision at most* $O(T^{d_2})$ *while the faces not in the subset all have precision at most* $O((T')^{d_2})$. *For* $j \in [N]$, *define* $L_j$ *as the linear constraint for the* $j$th *face.*

*Then the average of the* $|V|$ *vertices of* $P$ *will have margin at least* $\Omega(\frac{1}{|V|}\frac{1}{\alpha^{O(1)}})$ *for all* $(M-2)$-*dimensional faces of* $P$.

*Proof.* To reduce this to the setting of Lemma E.20, first substitute $n_M = 1 - \sum_{i=1}^{M-1} n_i$ in the inequalities for all $N$ faces of $P$ so that each inequality is an equivalent inequality over $(n_1, \ldots, n_{M-1})$ with integer coefficients increased by a factor of at most 2 (which will ultimately be absorbed by the Big-O notation).

Applying Lemma E.20 on $P$, where $n_M = 1 - \sum_{i=1}^{M-1} n_i$ was substituted into the inequalities defining faces of $P$, implies that for every $(M-2)$-dimensional face $F$ of $\mathrm{cl}(P)$ and every vertex $x$ of $\mathrm{cl}(P)$ which does not lie on $F$, then the margin of $x$ on $F$ is at least $\Omega(\frac{1}{\alpha^{O(1)}})$. All vertices which lie on $F$ have margin 0.

Consider $c$, the average of vertices $\{x_1, \ldots, x_{|V|}\}$ of $P$.

$$c = \frac{1}{|V|} \sum_{i=1}^{|V|} x_i$$

For any face $F$ of $P$ with constraint given by $L_F(\cdot)$, at least one of the $|V|$ vertices $\{x_1, \ldots, x_{|V|}\}$ does not lie on $F$, as $F$ is $(M-2)$-dimensional while $P$ is $(M-1)$-dimensional. By linearity of the margin,

$$L_F(c) = \frac{1}{|V|} \sum_{i=1}^{|V|} L_F(x_i)$$
$$\geq \Omega(\frac{1}{|V|}\frac{1}{\alpha^{O(1)}})$$

∎

Finally, we need one more Lemma to bound the number of vertices of the polytope of interest in the main proof.

**Lemma E.22.** *Denote the coordinates in an* $M$-*dimensional Euclidean space* $E$ *as* $(n_1, \ldots, n_M)$. *Consider a nonempty* $(M-1)$-*dimensional polytope* $P \subset E$, *such that* $\mathrm{cl}(P)$ *has vertices* $V$, *with* $|V| \geq M$, *and* $N$ *faces. Suppose* $P$ *is contained in the* $(M-1)$-*dimensional subspace given by* $\sum_{i\in[M]} n_i = 1$, *and* $P$ *is the intersection of an* $(M-1)$-*dimensional simplex and two halfspaces. Then, the number of vertices of* $P$ *is at most* $3M^2$.

*Proof.* Let $P = A \cap H_1 \cap H_2$, where $A$ is the $(M-1)$-dimensional simplex, and $H_1, H_2$ are the halfspaces. First, $A$ will have $M$ vertices and $M$ faces.

The new vertices formed by intersecting $H_1$ with $A$ can be upper bounded by counting the number of tuples of $M-1$ faces of $A$, as each vertex is formed by the intersection of $M$ faces. $H_1$, when intersecting $A$ will be able to form at most $\binom{M}{M-1} = M$ new vertices. $H_2$, when intersecting $A \cap H_1$, will be able to form at most $\binom{M+1}{M-1} = \frac{M(M+1)}{2}$ new vertices. The total vertices of $A \cap H_1 \cap H_2$ is at most $2M + \frac{M(M+1)}{2} \leq 3M^2$. ∎

### E.5. Lemmas for Discrete Approximation.

In this section, we are interested in showing that an activation $(B_1, \ldots, B_k)$ realized by some continuous test-function $\mathcal{Y}$ can also be approximately realized by some discrete test-function $\mathcal{X}$. The difficulty is that discrete test-functions consist of a sequence of lattice points $\{(j, ps(x)_j)\}_{j\in[|x|]}$, where successive lattice points differ by a step of $(+1, 0)$ (corresponding to $x_j = 0$) or $(+1, +1)$ (corresponding to $x_j = 1$).

First, here is an auxiliary Lemma that provides a strategy to construct a trajectory of lattice points where successive lattice points differ by a step of $(+1, 0)$ or $(+1, +1)$ and where the trajectory stays between two lines $y = \frac{b}{a}x$ and $y = \frac{d}{c}x$. The strategy offers a way to iteratively find the next lattice point in the trajectory.

**Lemma E.23.** *For any two 2D lines with slopes $1 > \frac{b}{a} > \frac{d}{c} > 0$ where $a, b, c, d \leq T$, there exists an infinite sequence of bits $x \in \{0, 1\}^\infty$ such that for all $n \geq T^2$ we have that $\frac{b}{a}n \geq \mathtt{ps}(x)_n > \frac{d}{c}n$. That is, the discrete test-function given by $x$ is above the lower line and below the upper line.*

*Proof.* $\min_{0 \leq a, b, c, d \leq T, \frac{b}{a} \neq \frac{d}{c}} (\frac{b}{a} - \frac{d}{c}) \geq \frac{1}{T^2}$, so for $n \geq T^2$, $(\frac{b}{a} - \frac{d}{c})n \geq 1$. Thus, for every $n \geq T^2$, there is at least one lattice point $(n, y_n) \in \mathbb{Z}^2$ with $\frac{b}{a}n \geq y_n > \frac{d}{c}n$.

Now let's consider a strategy to jump from $(n, y_n)$ to $(n + 1, y_{n+1})$. For all $n \geq T^2$, from any such lattice point, $(n, y_n)$, either $(n + 1, y_n)$ or $(n + 1, y_n + 1)$ will be in $(\frac{d}{c}(n + 1), \frac{b}{a}(n + 1)]$. Thus, there is always a continuation of the trajectory that remains between the two lines; one either takes a step along $(+1, 0)$ (corresponding to $x_{n+1} = 0$) or along $(+1, +1)$ (corresponding to $x_{n+1} = 1$) to reach $(n + 1, y_n)$ or $(n + 1, y_n + 1)$, respectively.

Finally, the trajectory from $(0, 0)$ to $(T^2, y_{T^2})$ can always be made by some trajectory using increments $(+1, +1)$ and $(+1, 0)$ since $y_{T^2} \leq T^2 \frac{b}{a} < T^2$. $\blacksquare$

The following Lemma says that we can approximate any continuous test-function $\mathcal{Y}$ with a length $n$ discrete test-function, for $n$ above some threshold.

**Lemma E.24.** *(Discrete Approximation to Continuous Test-Function) Suppose $\in \mathbb{N}$, $(T, K), (T', K') \in \mathbb{N}^2$ are such that $T^K \leq \alpha$ and $(T')^{K'} \leq \alpha$ and $K \leq T^2$ and $K' \leq (T')^2$. WLOG, suppose $T \geq T'$. Suppose we are given a $(k, T)$-configuration $\{s_i\}_{i \in [k]}$, where a subset of $\{s_i\}_{i \in [k]}$ of size at most $K$ have precision at most $T$, while the remaining entries of $\{s_i\}_{i \in [k]}$ have precision at most $T'$. For any schema $Y$ of $M$ segments, suppose $\mathcal{Y}$ is any continuous test-function of schema $Y$, with segment lengths $(\overline{n}_1(\mathcal{Y}), \ldots, \overline{n}_M(\mathcal{Y})) \in [0, 1]^M$ (where we assumed WLOG that $\sum_j \overline{n}_j(\mathcal{Y}) = 1$). Suppose every $\overline{n}_j(\mathcal{Y})$ is a rational number and that the common denominator of all $(\overline{n}_j(\mathcal{Y}))_{j \in [M]}$ is $p$.*

*Then there exists an $n_0 \leq O(p \cdot \alpha^2)$ so that for any positive integer multiple $n$ of $n_0$, there exists a discrete test-function $\mathcal{X}$ of length $n$ so that*

$$\forall i \in [k], |B_i(\mathcal{Y}) - B_i(\mathcal{X})| \leq \frac{T^2 + M}{n}$$

*Proof.* Denote $\mathrm{idx}(i) \in [k]$ as the line which the end-point of segment $i$ of the schema $Y$ (the schema of $\mathcal{Y}$) lies on.

First, given the $k$ slopes $s_1, s_2, \ldots s_k \in (0, 1)$, suppose that the denominators of these slopes are $a^{(1)}, \ldots, a^{(k)}$, where $\forall i \in [k], s_i = \frac{b^{(i)}}{a^{(i)}}$. There is a subset $U \subset [k]$ of size at most $K$ such that $\{a^{(i)} : i \in U\}$ are all positive integers of precision at most $T$. Further, $\{a^{(i)} : i \notin U\}$ is a set of at most $K'$ positive integers of precision at most $T'$. Let $d$ be the least common multiple of $\{a^{(1)}, \ldots, a^{(k)}\}$. Note that $d \leq T^K \cdot (T')^{K'} \leq \alpha^2$.

Now, set $n_0 = d \cdot p$, and consider any multiple $n$ of $n_0$. Consider the quantities, $\{n_i(\mathcal{Y})\}_{i \in [M]} := \{n \cdot \overline{n}_i(\mathcal{Y})\}_{i \in [M]}$, which are the segment lengths $\mathcal{Y}$, rescaled by a factor of $n$. We will refer to this rescaled $\mathcal{Y}$ instead of $\mathcal{Y}$ for convenience of notation in the proof; like a discrete test-function of length $n$, the rescaled $\mathcal{Y}$ will be a map from $[0, n] \to [0, n]$. Denote $\delta_i := \sum_{j=1}^i n_j(\mathcal{Y})$ for all $i \in [M]$ and $\delta_0 = 0$.

We will construct a discrete test-function which follows the same schema as the rescaled $\mathcal{Y}$ on the interval $[T^2, n]$, though on the interval $[0, T^2]$, $\mathcal{X}$ will have no guarantees.

First, we will define $M + 1$ lattice points where the discrete test-function will cross over the $k$ lines ("crossing points"). Second, we show how to connect those crossing points with a sequence of lattice points, where consecutive lattice points differ by increments of the 2D vectors, $(+1, +1)$ and $(+1, 0)$ (so that the sequence of lattice points is like a discrete test-function). At the end, we will argue that the constructed discrete test-function $\mathcal{X}$ and the rescaled $\mathcal{Y}$ will satisfy the guarantee that $\forall i \in [k], |B_i(\mathcal{Y}) - B_i(\mathcal{X})| \leq \frac{M + T^2}{n}$.

**Step 1: Defining the Crossing Points of the Discrete Test-Function on the $k$ Lines.** The rescaled continuous test-function $\mathcal{Y}$, where the input domain and the output is scaled by $n$, will cross the $k$ lines $l_1, \ldots, l_k$ at the $M + 1$ points,

$$\{(\delta_i, s_{\mathrm{idx}(i)}\delta_i)\}_{i \in [M]} \cup \{(0, 0)\}$$

Due to the way we have set $n$, these are all lattice points. Since $n = d \cdot p$, and all $\overline{n}_i(\mathcal{Y})$ have common denominator $p$, then $\overline{n}_i(\mathcal{Y})$ is an integer for all $i \in [M]$, so $\delta_i$ is an integer for all $i \in [M]$. Second, because of the factor of $d$, $s_{\text{idx}(i)}\delta_i$ is an integer. They are therefore valid points for the discrete test-function to go through, and we will design a discrete test-function that crosses the $k$ lines at $\{(\delta_i, s_{\text{idx}(i)}\delta_i)\}_{i \in [M]} \cup \{(0,0)\}$ for all $\delta_i \geq T^2$.

**Step 2: Connecting the Crossing Points.** We'll describe a strategy for constructing the discrete test-function that connects these crossing points. For any segment in the continuous schema, it will either cross between two consecutive lines, or it will cross a line and then recross the same line. We need only show how to do these types of crossings in a way that only crosses the appropriate lines at the start and endpoint of the segment $\{(\delta_i, s_{\text{idx}(i)}\delta_i)\}_{i \in [M]}$ and at no point in between. Consider the $(i+1)$th segment of $\mathcal{Y}$, which has start-point $(\delta_i, s_{\text{idx}(i)}\delta_i)$ and end-point $(\delta_{i+1}, s_{\text{idx}(i+1)}\delta_{i+1})$. There are three cases:

- The segment crosses down (segment start-point on line $l_j$ and end-point on line $l_{j+1}$ for some $j \in [k-1]$)

- The segment crosses up (segment start-point on line $l_{j+1}$ and end-point on $l_j$ for some $j \in [k-1]$)

- The segment crosses and re-crosses the same line (segment start and end-point on $l_j$ for some $j \in [k]$)

The strategies below will apply when the segment's start-point's x-coordinate is at least $T^2$ (i.e. $\delta_i \geq T^2$). After describing these, we will handle the corner case where a segment's start-point's x-coordinate is less than $T^2$ but its end-point's x-coordinate is larger than $T^2$.

**Case 1: Segment Crosses Down.** Suppose that we must connect the following start and end-point with a sequence of lattice points which lie between lines $l_j, l_{j+1}$:

$$(\delta_i, s_j \cdot \delta_i), (\delta_{i+1}, s_{j+1} \cdot \delta_{i+1}) \in \mathbb{Z}^2$$

The corresponding segment of the rescaled continuous test-function is the $(i+1)$th segment with length $n_{i+1}(\mathcal{Y})) \in \mathbb{Z}$.

First, there exists a sequence (trajectory) of lattice points $\{(x_t, y_t)\}_{0 \leq t \leq n_{i+1}(\mathcal{Y})}$ where $(x_0, y_0) = (\delta_i, s_j \cdot \delta_i)$, $(x_{n_{i+1}(\mathcal{Y})}, y_{n_{i+1}(\mathcal{Y})}) = (\delta_{i+1}, s_{j+1} \cdot \delta_{i+1})$ where for all $0 \leq t < n_{i+1}(\mathcal{Y}), x_{t+1} = x_t + 1$; and $y_{t+1} = y_t + 1$ or $y_{t+1} = y_t$. This is because the rescaled $\mathcal{Y}$ is a continuous test-function which connects these two lattice points and which is 1-Lipschitz and non-decreasing. We want to further show that there exists such a trajectory of lattice points $\{(x_t, y_t)\}$ such that $\forall 1 \leq t \leq n_{i+1}(\mathcal{Y}) - 1, s_j \cdot x_t \geq y_t > s_{j+1} \cdot x_t$.

The strategy to do so constructs a trajectory from the start-point $(\delta_i, s_j \cdot \delta_i)$ to the end-point $(\delta_{i+1}, s_{j+1} \cdot \delta_{i+1})$ and consists of two phases:

- (Phase 1) First, the following iterative procedure is applied, starting with $(x_0, y_0) = (\delta_i, s_j \cdot \delta_i)$ and $t = 0$:

    From the current lattice point $(x_t, y_t)$, choose $(x_{t+1}, y_{t+1}) = (x_t + 1, y_t)$ or $(x_t + 1, y_t + 1)$ depending on which one satisfies $s_j \cdot x_{t+1} \geq y_{t+1} > s_{j+1} \cdot x_{t+1}$. At least one of the two will always satisfy this condition if $x_0 \geq T^2$, as argued in Lemma E.23. If both are possible, then choose either.

    This is repeated until, $y_t$, the y-coordinate of the current lattice point equals $s_{j+1}\delta_{i+1}$.

- (Phase 2) The trajectory only takes steps along the direction $(+1, 0)$ so that $(x_{t+1}, y_{t+1}) = (x_t + 1, y_t)$ until the end-point is reached.

In Phase 1, we will always be able to find a next point $(x_{t+1}, y_{t+1})$ between $l_j, l_{j+1}$ from a current point $(x_t, y_t)$ between $l_j, l_{j+1}$, if $x_0 \geq T^2$ as argued in Lemma E.23. By induction, all such lattice points will lie between the two lines $l_j, l_{j+1}$. Phase 1 will terminate because $y_t$ is non-decreasing in $t$ and cannot remain bounded by any constant as $t$ increases to $\infty$, in order to stay between $l_j, l_{j+1}$, since $s_{j+1} > 0$. Thus, there exists some minimal $t_*$ such that $y_{t_*} = s_{j+1}\delta_{i+1}$ for the first time. Because the lattice points always stay between $l_j, l_{j+1}$, then at time $t_*$, we must have that $x_{t_*} < \delta_{i+1}$. Thus, Phase 1 terminates and Phase 2 can commence.

It is clear that Phase 2 will yield a set of lattice points that are between $l_j, l_{j+1}$ with the last lattice point being $(\delta_{i+1}, s_{j+1} \cdot \delta_{i+1})$ as desired.

This connects the two points with a trajectory that does not intersect $l_j$ and $l_{j+1}$ at any points besides the start and endpoint, as neither phase crosses the two lines.

**Case 2: Segment Crosses Up.**

Suppose that we must connect the following start and end-point with a sequence of lattice points which lie between lines $l_j, l_{j+1}$:

$$(\delta_i, s_{j+1} \cdot \delta_i), (\delta_{i+1}, s_j \cdot \delta_{i+1})$$

The argument is analogous. First, there exists a sequence of lattice points $\{(x_t, y_t)\}$ where $(x_0, y_0) = (\delta_i, s_{j+1} \cdot \delta_i)$, $(x_{n_{i+1}(\mathcal{Y})}, y_{n_{i+1}(\mathcal{Y})}) = (\delta_{i+1}, s_j \cdot \delta_{i+1})$ where for all $0 \leq t < n_{i+1}(\mathcal{Y}), x_{t+1} = x_t + 1$ and $y_{t+1} = y_t + 1$ or $y_{t+1} = y_t$. This is because the rescaled $\mathcal{Y}$ is a continuous test-function which connects these two lattice points and which is 1-Lipschitz and non-decreasing. We want to further show that there exists such a trajectory of lattice points $\{(x_t, y_t)\}$ such that $\forall 1 \leq t \leq n_{i+1}(\mathcal{Y}) - 1, s_j \cdot x_t \geq y_t > s_{j+1} \cdot x_t$.

The strategy to do so consists of two phases:

- (Phase 1) First, the following iterative procedure is applied, starting with $(x_0, y_0) = (\delta_i, s_{j+1} \cdot \delta_i)$ and $t = 0$:

   From the current lattice point $(x_t, y_t)$, choose $(x_{t+1}, y_{t+1}) = (x_t + 1, y_t)$ or $(x_t + 1, y_t + 1)$ depending on which one satisfies $s_j \cdot x_{t+1} \geq y_{t+1} > s_{j+1} \cdot x_{t+1}$. At least one of the two will always satisfy this condition if $x_0 \geq T^2$, as argued in Lemma E.23. If both are possible, then chose either.

   This is repeated until the current lattice point $(x_t, y_t)$ and the end-point $(\delta_{i+1}, s_j \cdot \delta_{i+1})$ can be connected with a line of slope 1. That is, $s_j \cdot \delta_{i+1} - y_t = \delta_{i+1} - x_t$.

- (Phase 2) The trajectory only takes steps along the direction $(+1, +1)$ so that $(x_{t+1}, y_{t+1}) = (x_t + 1, y_t + 1)$ until the end-point is reached.

The analysis of this strategy is analogous to that in Case 1.

**Case 3: Segment Crosses and Recrosses the Same Line, $l_j$.**

Suppose that we must connect the following start and end-point with a sequence of lattice points.

$$(\delta_i, s_j \cdot \delta_i), (\delta_{i+1}, s_j \cdot \delta_{i+1})$$

This can always be accomplished by a discrete test-function if $\delta_i \geq T^2$. There are two subcases: (1) if the trajectory needs to cross above line $j$ then recross line $j$, and (2) if the trajectory needs to cross below line $j$ and then recross line $j$. In case (1), we require the sequence of lattice points to lie between lines $l_{j-1}, l_j$ while in case (2), we require the sequence of lattice points to lie between lines $l_j, l_{j+1}$.

A simple strategy for subcase (1) is to first set $(x_1, y_1) = (\delta_i + 1, s_j \delta_i + 1)$ (i.e. move one step along to $(+1, +1)$ from the start-point). Note that $(\delta_i + 1, s_j \delta_i + 1)$ will not lie above line $j - 1$ since $\delta_i \geq T^2$, so $(s_{j-1} - s_j)\delta_i \geq 1$ and $s_j \delta_i + 1 \leq s_{j-1}(\delta_i + 1)$. The trajectory will now be above line $j$ and below line $j - 1$. Next, use Lemma E.23 to iteratively find the next lattice point $(x_{t+1}, y_{t+1})$ from the current lattice point $(x_t, y_t)$ by taking steps that are either $(+1, +1)$ or $(+1, 0)$ while staying between $l_j, l_{j-1}$ until $y_t = s_j \cdot \delta_{i+1}$. Then, take the remaining steps along $(+1, 0)$ until the end-point is reached.

An analogous strategy exists for subcase (2). Move one step according to $(+1, 0)$ so that $(x_1, y_1)$ is below line $j$ and above line $j+1$. Use Lemma E.23 to move to the next lattice point strictly between line $j$ and $j+1$ until $(\delta_{i+1} - x_t = s_j \cdot \delta_{i+1} - y_t)$.

Notice that this strategy is similar to those in Cases 1 and 2, and the analysis of them is the same.

**Step 3: Constructing the Final Discrete Test-Function.** We have proved that for each $i \in [M]$, the two crossing points $(\delta_{i-1}, s_{\text{idx}(i-1)}\delta_{i-1})$ and $(\delta_i, s_{\text{idx}(i)}\delta_i)$ can be connected by some sequence of lattice points that only crosses the $k$ lines $\{l_1, \ldots, l_k\}$ at the crossing points, as long as $\delta_{i-1} \geq T^2$. The lattice points differ from each other by increments of $(+1, 0)$ and $(+1, +1)$, like a discrete test-function. Each sequence of lattice points can be identified with one segment of the rescaled $\mathcal{Y}$ so that the number of lattice points, excluding $(\delta_{i-1}, s_{\text{idx}(i-1)}\delta_{i-1})$, equals $n_i(\mathcal{Y})$.

Before constructing the final discrete test-function, we will need to construct the rest of the lattice points of the discrete test-function corresponding to segments of $\mathcal{Y}$ whose start-point is not larger than $T^2$.

Suppose that the $M_0$th segment of $\mathcal{Y}$ is such that $0 \leq \delta_{M_0-1} \leq T^2 - 1 < \delta_{M_0}$, where $M_0 \in [M]$. We split the $M_0$th segment up into two pieces. Consider the restriction of the $M_0$th segment to the following domains: $[\delta_{M_0-1}, T^2]$ and $[T^2, \delta_{M_0}]$. We will construct a sequence of lattice points for the restriction of the $M_0$th segment on $[T^2, \delta_{M_0}]$ using similar techniques as in Step 2. We will then construct a sequence of lattice points from $(0, 0)$ to $(T^2, y_*)$ for some $y_*$, to complete the discrete test-function. There are three cases to consider.

**Case 1:** $\text{idx}(M_0 - 1) = j, \text{idx}(M_0) = j + 1$ for some $j \in [k - 1]$.

We have that $(s_j - s_{j+1})(T^2) \geq 1$, so there exists a lattice point $(T^2, y_*)$ such that $y_* \in (s_{j+1}T^2, s_jT^2]$, with $y_* \in \mathbb{Z}$. We then construct a sequence of lattice points from $(T^2, y_*)$ to $(\delta_{M_0}, s_{\text{idx}(M_0)}\delta_{M_0})$ using the same strategy as in Step 2, Case 1. This sequence of lattice points will be such that consecutive lattice points differ by an increment of $(+1, +1)$ or $(+1, 0)$ and all lattice points are between $l_j$ and $l_{j+1}$.

We also construct a sequence of lattice points from $(0, 0)$ to $(T^2, y_*)$. The only requirement of this sequence of lattice points is that consecutive lattice points differ by an increment of $(+1, +1)$ or $(+1, 0)$. Since $y_* \leq s_jT^2 < T^2$, this will be possible.

**Case 2:** $\text{idx}(M_0 - 1) = j + 1, \text{idx}(M_0) = j$ for some $j \in [k - 1]$.

Similar to Case 1. We have that $(s_j - s_{j+1})(T^2) \geq 1$, so there exists a lattice point $(T^2, y_*)$ such that $y_* \in (s_{j+1}T^2, s_jT^2]$, with $y_* \in \mathbb{Z}$. We then construct a sequence of lattice points from $(T^2, y_*)$ to $(\delta_{M_0}, s_{\text{idx}(M_0)}\delta_{M_0})$ using the same strategy as in Step 2, Case 2. This sequence of lattice points will be such that consecutive lattice points differ by an increment of $(+1, +1)$ or $(+1, 0)$ and all lattice points are between $l_j$ and $l_{j+1}$.

We also construct a sequence of lattice points from $(0, 0)$ to $(T^2, y_*)$. The only requirement of this sequence of lattice points is that consecutive lattice points differ by an increment of $(+1, +1)$ or $(+1, 0)$. Since $y_* \leq s_jT^2 < T^2$, this will be possible.

**Case 3:** $\text{idx}(M_0 - 1) = j, \text{idx}(M_0) = j$ for some $j \in [k]$.

There are two subcases: either segment $M_0$ is contained in $\text{Sector}_j$ or in $\text{Sector}_{j+1}$ (Sectors are defined in Definition E.4). We will just consider the first subcase, as the second is analogous. Define $s_0 = 1$ and $l_0$ as the homogeneous, 2D line with slope $s_0$.

We have that $(s_{j-1} - s_j)(T^2) \geq 1$, so there exists a lattice point $(T^2, y_*)$ such that $y_* \in (s_jT^2, s_{j-1}T^2]$, with $y_* \in \mathbb{Z}$. We then construct a sequence of lattice points from $(T^2, y_*)$ to $(\delta_{M_0}, s_{\text{idx}(M_0)}\delta_{M_0})$ using the same strategy as in Step 2, Case 1. This sequence of lattice points will be such that consecutive lattice points differ by an increment of $(+1, +1)$ or $(+1, 0)$ and all lattice points are between $l_{j-1}$ and $l_j$.

We also construct a sequence of lattice points from $(0, 0)$ to $(T^2, y_*)$. The only requirement of this sequence of lattice points is that consecutive lattice points differ by an increment of $(+1, +1)$ or $(+1, 0)$. Since $y_* \leq s_{j-1}T^2 \leq T^2$, this will be possible.

We now construct the full discrete test-function by concatenating all these sequences of lattice points into a single sequence of lattice points. For each $i \in [M_0, M]$, use the aforementioned constructions to construct a sequence of lattice points with start-point $(\max(\delta_{i-1}, T^2), s_{\text{idx}(i-1)} \cdot \max(\delta_{i-1}, T^2))$ and end-point $(\delta_i, s_{\text{idx}(i)}\delta_i)$. Remove the start-point from each of these sequences of lattice points, and concatenate all the sequences of lattice points for $i \in [M]$. Finally, concatenate the sequence of lattice points from $(0, 0)$ to $(T^2, y_*)$, for $y_*$ as described in Case 3 above. This sequence of lattice points is a valid discrete test-function since consecutive lattice points differ by an increment of $(+1, +1)$ or $(+1, 0)$. We will refer to the constructed discrete test-function as $\mathcal{X}$.

**Properties of the Final Construction.** For $i \in [M_0, M]$, there is a correspondence between the $i$th segment of $\mathcal{Y}$ and the sequence of lattice points with start-point $(\max(\delta_{i-1}, T^2), s_{\text{idx}(i-1)} \cdot \max(\delta_{i-1}, T^2))$ and end-point $(\delta_i, s_{\text{idx}(i)}\delta_i)$. The difference between the number of lattice points in the latter and the length of the former is at most $T^2$. In addition, all except possibly one of those lattice points (the end-point) lie in the sector in which the $i$th segment of $\mathcal{Y}$ lies in.

Define $(B_i(\mathcal{Y}))_{i \in [k]}$ and $(B_i(\mathcal{X}))_{i \in [k]}$ as follows.

$$\forall i \in [k], B_i(\mathcal{Y}) := \int_0^1 \mathbb{1}[\mathcal{Y}(j) > s_i \cdot j]dj$$

$$\forall i \in [k], B_i(\mathcal{X}) := \frac{1}{n}\sum_{j=1}^n \mathbb{1}[\mathcal{X}(j) > s_i \cdot j]$$

For any $i \in [k]$, $B_i(\mathcal{X})$ depends on the total number of lattice points (out of $n$) which are strictly above line $i$, while $B_i(\mathcal{Y})$ depends on the total length of segments which lie above line $i$. The maximum deviation between $B_i(\mathcal{X})$ and $B_i(\mathcal{Y})$ is upper bounded by $\frac{1}{n}$ multiplied by the number of lattice points $(j, \mathcal{X}(j)), j \in [n]$, which are not in the Sector which their corresponding segment is in, which is upper bounded by $\frac{T^2}{n} + \frac{\# \text{ times } \mathcal{Y} \text{ crosses line } i}{n} = \frac{T^2+M}{n}$. More precisely, for any $i \in [k]$,

$$
\begin{aligned}
|B_i(\mathcal{X}) - B_i(\mathcal{Y})| &\leq \frac{1}{n} \sum_{m \in [M]} \sum_{\delta_{m-1} < j \leq \delta_m} \mathbb{1}[(j, \mathcal{X}(j)) \text{ is not in the same Sector that segment } m \text{ of } \mathcal{Y} \text{ is in}] \\
&\leq \frac{1}{n}[T^2 + \#(j \in [0,1] : \mathcal{Y}(j) = s_i \cdot j)] \\
&\leq \frac{T^2 + M}{n}
\end{aligned}
$$

The $\frac{T^2}{n}$ term arises from the lattice points $(j, \mathcal{X}(j))$ for $j \in [T^2]$, which have no guarantee of being in the Sector of the corresponding segment of $\mathcal{Y}$. The $\frac{\# \text{ times } \mathcal{Y} \text{ crosses line } i}{n}$ term arises from the fact that for $j > T^2$, only $(j, \mathcal{X}(j))$ which lie on line $i$ may potentially lie in a different Sector than that of the corresponding segment of $\mathcal{Y}$. This is upper bounded by $M$ since $\mathcal{Y}$ has only $M$ segments and so can cross line $i$ at most $M$ times. ∎

### E.6. Auxiliary Lemmas

**Convex Geometry Lemmas.**

**Lemma E.25.** *(Theorem 6.2 of (Rockafellar, 1970)) Let $C$ be a nonempty convex set. Then its relative interior, $\mathrm{ri}(C)$, is also nonempty.*

**Lemma E.26.** *(Theorem 6.1 of (Rockafellar, 1970)) Let $C$ be a convex set in $\mathbb{R}^n$. Let $x \in \mathrm{ri}(C)$ and $z \in \mathrm{cl}(C)$. Then for all $0 \leq \lambda < 1$, $(1 - \lambda)x + \lambda z \in \mathrm{ri}(C)$.*

**Lemma E.27.** *Let $C$ be a nonempty convex set in $\mathbb{R}^n$. Consider an open halfspace $H := \{x \in \mathbb{R}^n : \sum_{i \in [n]} \lambda_i x_i > \alpha\}$. If $C \cap H \neq \emptyset$, then $\dim(C \cap H) = \dim(C)$.*

*Proof.* Suppose that $p \in C \cap H$. If $p \in \mathrm{ri}(C)$, then since $H$ is an open set, we are done.

If not, $p \in \partial C := \mathrm{cl}(C) - \mathrm{ri}(C)$. By Lemma E.25, since $C \neq \emptyset$, there exists $x \in \mathrm{ri}(C)$. By Lemma E.26, $\forall 0 \leq \lambda < 1$, $p_\lambda := \lambda p + (1 - \lambda)x \in \mathrm{ri}(C)$.

We claim there exists some setting of $\lambda \approx 1$ such that $p_\lambda \in H$. Since $H$ is open, there exists $\delta > 0$ such that $\mathrm{Ball}_\delta(p) := \{x' \in \mathbb{R}^n : ||x' - p||_2 < \delta\} \subset H$. Taking $\lambda > 1 - \frac{\delta}{||x-p||_2}$ ensures that $||p_\lambda - p||_2 < \delta \implies p_\lambda \in H$. Thus, $p_\lambda \in H \cap \mathrm{ri}(C)$. ∎

**Lemmas Regarding Existence of Low Precision Element in $L_\infty$ Ball.**

**Lemma E.28.** *Let $N > 0$ be an integer. For any $x \in \mathbb{R}$, the interval $[x, x + \frac{1}{N}]$ contains at least one element of the form $\frac{m}{N}$ for integer $m$.*

*Proof.* For $N > 0$, the set $S = \{\frac{m}{N} : m \in \mathbb{Z}\}$ is such that the minimum distance between any two consecutive elements is $\frac{1}{N}$. Since the length of the interval $[x, x + \frac{1}{N}]$ is $\frac{1}{N}$, it must contain some element of $S$, or else this would imply that there is some pair of consecutive elements in $S$ such that the distance between them is greater than $\frac{1}{N}$. ∎

**Lemma E.29.** *Let $N > 0$ be an integer. For any $x = (x_1, \ldots, x_d) \in \mathbb{R}^d$ with $\sum_{i \in [d]} x_i = 1$, the $L_\infty$ ball $\{x' \in \mathbb{R}^d : ||x - x'||_\infty \leq \frac{1}{N}\} \cap \{x' : \sum_{i \in [d]} x'_i = 1\}$ contains a point $x'' \in \mathbb{Q}^d$ whose fractional coordinates have a least common denominator of at most $d \cdot N$.*

*Proof.* Consider $\mathrm{Ball}^\infty_{\frac{1}{dN}}(x) := \{x' \in \mathbb{R}^d : ||x - x'||_\infty \leq \frac{1}{dN}\} \cap \{x' \in \mathbb{R}^d : \sum_{i \in [d]} x'_i = 1\}$.

Along each dimension $i \in \{1, 2, \ldots, d-1\}$, there must exist a number in the interval $[x_i - \frac{1}{dN}, x_i + \frac{1}{dN}]$ of the form $\frac{m_i}{dN}$ for integer $m_i$ by Lemma E.28. We can set $x''_i = \frac{m_i}{dN}$ for all $i \in [d-1]$. The last coordinate $x''_d$ will need to be set to $1 - \sum_{i \in [d-1]} \frac{m_i}{dN}$ so that $x'' \in \{x' : \sum_{i \in [d]} x'_i = 1\}$. Finally, $|x''_d - x_d| \leq \sum_{i \in [d-1]} |x''_i - x_i| \leq \frac{d-1}{dN} \leq \frac{1}{N}$. Hence, we have that $x'' \in \mathrm{Ball}^\infty_{\frac{1}{N}}(x)$ and that each coordinate is of the form of $\frac{m_i}{dN}$ for some integer $m_i$. ∎

### Relating to Technical Lemma E.30.

**Lemma E.30.** *If for every $f \in$ C-RASP$^2$ with $K(f)$ heads, we have that $z > 0$ and $\sum_{i \in [K(f)]} \lambda_i > z$, then for any $f, f' \in$ C-RASP$^2$ that are not equal, with $K$ and $K'$ heads respectively, then $f$ and $f'$ will differ on some continuous test-function, $\mathcal{Y}$, which strictly satisfies the second layer inequalities. That is, with $\max(K, K') \leq k \leq K + K'$ distinct heads between $f$ and $f'$, let $(B_1(\mathcal{Y}), \ldots, B_k(\mathcal{Y}))$ be the activations induced by $\mathcal{Y}$. Then, either:*

$$\sum_{i=1}^{K} \lambda_i B_{\mathrm{ord}(1,i)}(\mathcal{Y}) > z$$

$$\sum_{i=1}^{K'} \lambda'_i B_{\mathrm{ord}(2,i)}(\mathcal{Y}) < z'$$

*Or,*

$$\sum_{i=1}^{K} \lambda_i B_{\mathrm{ord}(1,i)}(\mathcal{Y}) < z$$

$$\sum_{i=1}^{K'} \lambda'_i B_{\mathrm{ord}(2,i)}(\mathcal{Y}) > z'$$

To prove this, we will need Lemma E.33 and E.34. First, define the following notion of connectedness for subsets of Euclidean space.

**Definition E.31.** (Definition 2.4 of (Freiwald, 2014)) Two subsets $A$ and $B$ of a metric space $X$ are said to be separated if both $\mathrm{cl}(A) \cap B = \emptyset$ and $A \cap \mathrm{cl}(B) = \emptyset$. A set $E \subset X$ is connected if $E$ is not the union of two nonempty, separated sets.

We will utilize the following Lemma (implicitly) in the proof below to show that a union of sets $\bigcup_{j \in [N]} A_j$ is connected, by iteratively arguing that for each $j \in [N] - \{1\}$, $A_j$ is not separated from $\bigcup_{k<j} A_k$ (i.e. they "share an edge").

**Lemma E.32.** *(Theorem 2.9 in (Freiwald, 2014)) Suppose $C$ and $\{C_\alpha\}_{\alpha \in I}$ are connected subsets of $X$ and that for each $\alpha$, $C_\alpha$ and $C$ are not separated. Then $C \cup \bigcup_{\alpha \in I} C_\alpha$ is connected*

Next, fix $f, f'$ and the $k$ distinct slopes $\{s_i\}_{i \in [k]}$ across the first layers of both programs. Let's consider the $k$-dimensional set $\mathbb{A}(\{s_i\}_{i \in [k]}) \subset [0, 1]^k$. $\mathbb{A}(\{s_i\}_i)$ is equal to the union of a finite number of convex polytopes by Corollary E.14 and Lemma E.16.

$$\mathbb{A}(\{s_i\}_{i \in [k]}) = \bigcup_{m \in [k], \text{valid } \{(y_1^i, y_2^i)\}_{i \in [m]} \subset \{1, \ldots, k\}^m} A(Y_{\{(y_1^i, y_2^i)\}_{i \in [m]}})$$

Here are two key Lemmas we will need.

**Lemma E.33.** *(Dimension of Polytopes) Given a $(k, T)$-configuration $\{s_i\}_{i \in [k]}$, then for each basis schema $Y_{\{(y_1^i, y_2^i)\}_{i \in [m]}}$, the set of activations $A(Y_{\{(y_1^i, y_2^i)\}_{i \in [m]}})$ has dimension $k$.*

*Proof.* Consider any basis schema $Y$ that spans $k$ lines with $M'$ segments, as described in Corollary E.14. It suffices to show $k$ independent degrees of freedom in $A(Y)$.

Given $(B_1, \ldots B_k)$ consider the full-rank linear map, $Q : \mathbb{R}^k \to \mathbb{R}^k$

$$Q(B_1, \ldots, B_k) = (B_1, B_2 - B_1, B_3 - B_2, \ldots, B_k - B_{k-1})$$

Denote $Q \circ A(Y)$ as the set consisting of points attained by applying $Q$ on all points in $A(Y)$. We have $\dim(Q \circ A(Y)) = \dim(A(Y)) - \dim(\ker(Q) \cap A(Y))$ so since $Q$ has rank $k$, $\dim(\ker(Q) \cap A(Y)) \leq \dim(\ker(Q)) = 0$. So to prove that $\dim(A(Y)) = k$, it suffices to prove $\dim(Q \circ A(Y))$ is $k$.

By Lemma E.16, schema $Y$ will span all $k$ lines and will have at least one segment in each of the $k + 1$ sectors. First, set the lengths of all the segments that are in sector $k + 1$ to 0. This can always be done because by Lemma E.6, there is no constraint on any segment in $\text{Sector}_{k+1}$ since such a segment would not be crossing two distinct lines. Say that there are $M$ non-zero-length segments left that are each in the sectors $\{1, 2, \ldots, k\}$. Let $(n_1, \ldots, n_M)$ be the lengths of these segments.

There is a linear mapping $QL : \mathbb{R}^M \to \mathbb{R}^k$ that maps $(n_1, \ldots, n_M)$ to a vector $v \in \mathbb{R}^k$, where for all $i \in [k]$, $v_i$ is the sum of the length of segments that are contained in in $\text{Sector}_i$. This map is the multiplication of $Q$ and $L$, where $L$ is a linear map $L : \mathbb{R}^M \to \mathbb{R}^k$ which maps segment lengths of $Y$, $(n_1, \ldots, n_M)$, to activations $(B_1, \ldots, B_k)$. In matrix form, $L \in \{0, 1\}^{k \times M}$ is such that $L_{ij} = 1 \iff$ segment $j$ in schema $Y$ lies above line $i$ and hence contributes to activation $B_i$.

Now, for each sector $i \in [k]$, let:

$$j_i := \max\{j \in [M] : n_j \in \text{Sector}_i\}$$

be the index of the last segment in the schema that is in Sector $i$.

Now, for each $i \in [k]$, consider a setting of $(n_1, \ldots, n_M)$ where for all $j < j_i$, $n_j = 0$, but $n_{j_i} > 0$. Denote the setting for $i \in [k]$ as $N_i := (n_1^{(i)}, \ldots, n_M^{(i)})$. We claim that the $k$ vectors, $\{QL(N_i)\}_{i \in [k]}$ are linearly independent. Since $\{QL(N_i)\}_{i \in [k]} \subset Q \circ A(Y)$, this would imply the first point of the Lemma. The claim that $\{QL(N_i)\}_{i \in [k]}$ are linearly independent can be argued by induction.

As a base case, suppose that $i_0 \in [k]$ is the sector where the $M$th segment of the schema lies in. Then $QL(N_{i_0}) = e_{i_0}$. The singleton set is linearly independent.

For each $1 \leq r \leq k$, denote $I_r := \{i \in [k] : j_i \text{ is one of the } r\text{-largest of the set } \{j_i\}_{i \in [k]}\}$, so $I_1 = \{i_0\}$. Suppose by the inductive hypothesis that for $1 \leq r \leq k$, the $r$ vectors $\{QL(N_i) : i \in I_r\}$ are linearly independent. Now, let $i_*$ be the sector such that $j_{i_*}$ is the $(r + 1)$th largest of the set $\{j_i\}_{i \in [k]}$. $(n_1^{(i_*)}, \ldots, n_M^{(i_*)})$ will be such that $n_{j_{i_*}}^{(i_*)}$ is nonzero, which implies that the $i_*$th entry of $QL(N_{i_*})$ is nonzero. In contrast, for all $i \in I_r$, the $i_*$th entry of $QL(N_i)$ is zero, because for all $j > j_{i_*}$, there is no segment $n_j$ that lies in sector $i_*$, by definition of $j_{i_*}$, so by the construction of $N_i$ where all segments of index less than $j_i$ are set to 0, then the $i_*$th entry of $QL(N_i)$ is 0 for all $i \in I_r$. Thus, the $r + 1$ vectors $\{QL(N_i) : i \in I_{r+1}\}$ are linearly independent. It follows that $I_k$ is a set of $k$ linearly independent vectors in $Q \circ A(Y)$, so $\dim(Q \circ A(Y)) = k$. ∎

**Lemma E.34.** *(Connectedness of Interiors of Polytopes) Given a $(k, T)$-configuration $\{s_i\}_{i \in [k]}$, the set $\bigcup_{m \in [k], \text{valid } \{(y_1^i, y_2^i)\}_{i \in [m]} \subset \{1, \ldots, k\}^m} \text{int}(A(Y_{\{(y_1^i, y_2^i)\}_{i \in [m]}}))$ is connected.*

*Proof.* Let us first recall the set of basis schemas in Corollary E.14. For any $1 \leq m \leq k$ and $\{(y_1^i, y_2^i)\}_{i \in [m]} \subset \{1, \ldots, k\}^m$ such that

$$y_1^m = y_2^m$$
$$\forall i \in [m-1], y_1^i \neq y_2^i$$
$$\forall i \in [m-2], y_1^i < y_2^i \implies y_2^i = y_1^{i+1} > y_2^{i+1} > y_1^i$$
$$\forall i \in [m-2], y_1^i > y_2^i \implies y_2^i = y_1^{i+1} < y_2^{i+1} < y_1^i$$
$$(y_1^1, y_2^1) = (1, k) \text{ or } (y_1^1, y_2^1) = (k, 1)$$

We can the test-function schema, $Y_{\{(y_1^i, y_2^i)\}_{i \in [m]}}$ as the concatenation of $m-1$ monotone curves, where the $i$th monotone curve goes from lines $y_1^i$ to $y_2^i$ for $i \in [m-1]$. The set of these test-function schemas over all valid $\{(y_1^i, y_2^i)\}_{i \in [m]}$ satisfying the above is complete by Corollary E.14.

Let $A(Y_{\{(y_1^i, y_2^i)\}_{i \in [m]}})$ denote the set of activations induced by test-functions of schema $Y_{\{(y_1^i, y_2^i)\}_{i \in [m]}}$. $A(Y_{\{(y_1^i, y_2^i)\}_{i \in [m]}})$ is a convex set by Lemma E.16. Thus, each set $\text{int}(A(Y_{\{(y_1^i, y_2^i)\}_{i \in [m]}}))$, by itself, is connected.

Now, imagine a graph where each basis schema $Y_{\{(y_1^i, y_2^i)\}_{i \in [m]}}$ is a node. Create an edge between basis schema $Y$ and $Y'$ if $\text{int}(A(Y)) \cap \text{int}(A(Y')) \neq \emptyset$. By Lemma E.32, to prove our desired conclusion, it is sufficient to show that this graph is connected. For this, we will need to prove which edges exist and that these edges suffice to connect the graph.

Categorize the basis schema into levels, where schema of the same levels have the same number of monotone curves (i.e. the parameter $m \in [k]$).

**Edges Within Same Level.** Consider any two basis test-function schemas, $Y_{\{(y_1^i, y_2^i)\}_{i \in [m]}}$ and $Y_{\{((y_1^i)', (y_2^i)')\}_{i \in [m]}}$, with $m$ monotone curves. Suppose the two schemas' 1st through $(m-2)$-th monotone curves are the same, but the $(m-1)$th (i.e. their last) monotone curve's endpoint is different by 1 index. That is,

$$\forall i \in [m-2], y_1^i = (y_1^i)' \text{ and } y_2^i = (y_2^i)'$$
$$y_1^{m-1} = (y_1^{m-1})'$$
$$y_2^{m-1} = (y_2^{m-1})' - 1$$

We claim that the set of activations of these two "adjacent" schema must share an interior point:

$$\text{int}(A(Y_{\{(y_1^i, y_2^i)\}_{i \in [m]}})) \cap \text{int}(A(Y_{\{((y_1^i)', (y_2^i)')\}_{i \in [m]}})) \neq \emptyset$$

Suppose $Y_{\{(y_1^i, y_2^i)\}_{i \in [m]}}$ has $M$ segments and $Y_{\{((y_1^i)', (y_2^i)')\}_{i \in [m]}}$ has $M'$ segments[2]. Because the two schema have the same monotone curves except the last monotone curve, then there are two possibilities:

- Either $y_1^{m-1} < y_2^{m-1}$ and $(y_1^{m-1})' < (y_2^{m-1})'$. One can interpret these monotone curves as "moving downward", since the $(m-1)$th monotone curve of $Y_{\{(y_1^i, y_2^i)\}_{i \in [m]}}$ (resp. $Y_{\{((y_1^i)', (y_2^i)')\}_{i \in [m]}}$ has $M'$) has start-point on line $y_1^{m-1}$ (resp. $(y_1^{m-1})'$) and end-point on line $y_2^{m-1}$ (resp. $(y_2^{m-1})'$). Note the lines with lower index have higher slope by convention (i.e. $s_1$ is largest).

- Or $y_1^{m-1} > y_2^{m-1}$ and $(y_1^{m-1})' > (y_2^{m-1})'$. One can interpret these monotone curves as "moving upward".

WLOG, consider just the first case (the second one utilizes an analogous argument). Then, since $y_1^{m-1} < y_2^{m-1}$, the last monotone curve is moving downward. Suppose WLOG that the monotone curve which starts at line $(y_1^{m-1})'$ and ends at line $(y_2^{m-1})'$ moves down one more line than the monotone curve which starts at line $y_1^{m-1}$ and ends at line $y_2^{m-1}$.

---

[2]Note: the two schema have the same number of monotone curves, but different numbers of segments

Thus, $M' = M + 1$. Moreover, the set of segments which make up the two schema are similar: for all $j \in [M]$, segment $j$ of $Y_{\{(y_1^i, y_2^i)\}_{i \in [m]}}$ lies in the same sector as segment $j$ of $Y_{\{((y_1^i)', (y_2^i)')\}_{i \in [m]}}$. Meanwhile, segment $M + 1$ of $Y_{\{((y_1^i)', (y_2^i)')\}_{i \in [m]}}$ has no counterpart in $Y_{\{(y_1^i, y_2^i)\}_{i \in [m]}}$.

To show connectedness, we need to exhibit a point $(B_1, \ldots, B_k) \in [0, 1]^k$ that is in the interior of $A(Y_{\{(y_1^i, y_2^i)\}_{i \in [m]}})$ and $A(Y_{\{((y_1^i)', (y_2^i)')\}_{i \in [m]}})$.

For schema $Y_{\{(y_1^i, y_2^i)\}_{i \in [m]}}$ (resp. $Y_{\{((y_1^i)', (y_2^i)')\}_{i \in [m]}}$), there is a linear transformation $L$ (resp. $L'$) that maps the set of valid segment lengths $A^{(M)}(Y_{\{(y_1^i, y_2^i)\}_{i \in [m]}}) \in [0, 1]^M$ (resp. $A^{(M')}(Y_{\{((y_1^i)', (y_2^i)')\}_{i \in [m]}}) \in [0, 1]^{M'}$) of schema $Y_{\{(y_1^i, y_2^i)\}_{i \in [m]}}$ (resp. $Y_{\{((y_1^i)', (y_2^i)')\}_{i \in [m]}}$) to the set of activations $A(Y_{\{(y_1^i, y_2^i)\}_{i \in [m]}}) \in [0, 1]^k$ (resp. $A(Y_{\{((y_1^i)', (y_2^i)')\}_{i \in [m]}}) \in [0, 1]^k$). Being rank $k$ (see bullet point 1 of Lemma E.33), $L$ (resp. $L'$) will map interior points in $A^{(M)}(Y_{\{(y_1^i, y_2^i)\}_{i \in [m]}})$ (resp. $A^{(M')}(Y_{\{((y_1^i)', (y_2^i)')\}_{i \in [m]}})$) to interior points of $A(Y_{\{(y_1^i, y_2^i)\}_{i \in [m]}})$ (resp. $A(Y_{\{((y_1^i)', (y_2^i)')\}_{i \in [m]}})$). Thus, it suffices to exhibit two settings of the lengths of the segments of the two schema $(a_1, \ldots, a_M)$ and $(b_1, \ldots, b_{M'}) = (b_1, \ldots, b_M, b_{M+1})$, that are interior points in their respective polytopes $A^{(M)}(Y_{\{(y_1^i, y_2^i)\}_{i \in [m]}})$ (resp. $A^{(M')}(Y_{\{((y_1^i)', (y_2^i)')\}_{i \in [m]}})$), that give rise to the same $(B_1, \ldots, B_k) \in [0, 1]^k$.

Towards this goal, we are interested in the regime where $b_{M+1} = \delta \ll 1$ and $\forall j \in [M], a_j \approx b_j$, and where each segment length has slack $\epsilon$ with respect to its constraint defined in Lemma E.6. Suppose by Lemma E.6 that the constraints for the first $M$ segments for schema $Y_{\{((y_1^i)', (y_2^i)')\}_{i \in [m]}}$ are:

$$b_j \geq \frac{p_j}{q_j} \sum_{i=1}^{j-1} b_i, \forall j \in [M]$$

We'll now describe how to set $\{a_j\}_{j \in [M]}$ and $\{b_j\}_{j \in [M']}$. We will largely ignore the normalization conditions that $\sum_{j \in [M]} a_j = 1$ and $\sum_{j \in [M']} b_j = 1$, but only require that the total length is the same: $\sum_{j \in [M]} a_j = \sum_{j \in [M']} b_j > 0$.

For $\epsilon, \delta > 0$, set $b_1 = 1$ and for $j \in \{2, \ldots, M\}$, sequentially assign $b_j = \frac{p_j}{q_j} \sum_{i=1}^{j-1} b_i + \epsilon$. Set $b_{M+1} = \delta$ (where the only constraint for $b_{M+1}$ is that it is nonnegative, being the last segment). $\{b_j\}_{j \in [M+1]}$ is a valid set of segment lengths respecting the constraints imposed by schema $Y_{\{((y_1^i)', (y_2^i)')\}_{i \in [m]}}$ as per Lemma E.6. $\{b_j\}_{j \in [M+1]}$ is also an interior point of $A^{(M')}(Y_{\{((y_1^i)', (y_2^i)')\}_{i \in [m]}})$, since each constraint has slack $\epsilon > 0$ or $\delta > 0$.

Now, suppose that the $(M + 1)$th segment of $Y_{\{((y_1^i)', (y_2^i)')\}_{i \in [m]}}$ lies in sector $i_*$. Because each basis schema spans all $k$ lines, then there exists some $j_* \in [M]$ such that segment $j_*$ in $Y_{\{(y_1^i, y_2^i)\}_{i \in [m]}}$ is also in sector $i_*$. Set $\{a_i\}$ as follows.

$$\forall j \in [M] - \{j_*\}, \text{ set } a_j = b_j$$
$$\text{Finally, set } a_{j_*} = b_{j_*} + \delta$$

We now need to argue this is a valid assignment of $\{a_i\}$, respecting the constraints on segments imposed by the schema $Y_{\{(y_1^i, y_2^i)\}_{i \in [m]}}$. Because the first $M$ segments of $Y_{\{((y_1^i)', (y_2^i)')\}_{i \in [m]}}$ are the same as the $M$ segments of $Y_{\{(y_1^i, y_2^i)\}_{i \in [m]}}$, then these segments have the same constraints.

It follows that for all $j \leq j_*$, $a_j$ meets its constraint. For subsequent segments, due to the extra $\delta$ contribution on $a_{j_*}$, we will need to argue that their constraint is still met with some nonzero slack. The $t$th constraint after $j^*$ will have its slack lessened by at most $\delta \cdot [\max_{p,q \in \mathbb{Z}, |p|, |q| \leq T}(\frac{p}{q})^t] \leq \delta T^M \leq \delta T^{k^2}$, where we used that $t \leq M \leq k^2$ by Corollary E.15.

Thus, as long as $\delta < \frac{\epsilon}{T^{k^2}}$, then this setting of $\{a_j\}_{j \in [M]}$ will still satisfy each constraint of $Y_{\{(y_1^i, y_2^i)\}_{i \in [m]}}$ with nonzero slack.

$$a_j \geq \frac{p_j}{q_j} \sum_{i=1}^{j-1} a_i, \forall j \in [M]$$

Thus, $\{a_j\}_{j\in[M]}$ is an interior point of $A^{(M)}(Y_{\{(y_1^i,y_2^i)\}_{i\in[m]}})$.

Finally, the segments $\{a_j\}_{j\in[M]}$ and $\{b_j\}_{j\in[M+1]}$ induce the same $\{B_i\}_{i\in[k]}$ since the sum of the segment lengths in each sector is the same. This proves that:

$$\text{int}(A(Y_{\{(y_1^i,y_2^i)\}_{i\in[m]}})) \cap \text{int}(A(Y_{\{((y_1^i)',(y_2^i)')\}_{i\in[m]}})) \neq \emptyset$$

**Edges Between Levels $L$ and $L+1$.** Consider any two schema $Y_{\{(y_1^i,y_2^i)\}_{i\in[m-1]}}, Y_{\{((y_1^i)',(y_2^i)')\}_{i\in[m]}}$ with $m-2$ and $m-1$ monotone curves, respectively, where $Y_{\{((y_1^i)',(y_2^i)')\}_{i\in[m]}}$'s first $m-2$ monotone curves are the same as $Y_{\{(y_1^i,y_2^i)\}_{i\in[m-1]}}$'s $m-2$ monotone curves. Moreover, let schema $Y_{\{((y_1^i)',(y_2^i)')\}_{i\in[m]}}$ be such that:

- In the case that $(y_1^{m-1})' < (y_2^{m-1})'$, then $(y_1^{m-1})' + 1 = (y_2^{m-1})'$.

- In the case that $(y_1^{m-1})' > (y_2^{m-1})'$, then $(y_1^{m-1})' - 1 = (y_2^{m-1})'$

Instead of a schema of $m-2$ monotone curves, schema $Y_{\{(y_1^i,y_2^i)\}_{i\in[m]}}$ can also be thought of as a schema of $m-1$ monotone curves, where the last monotone curves will only cross the line with index $y_1^{m-2} = y_2^{m-2}$. Thus, these two schema can be thought of as having their final $(m-1)$-th monotone curves' endpoints differ by one line, which reduces to the case above with two schema in the same level. Using exactly the same argument as in the previous section with two schema in the same level, we can conclude that

$$\text{int}(A(Y_{\{(y_1^i,y_2^i)\}_{i\in[m-1]}})) \cap \text{int}(A(Y_{\{((y_1^i)',(y_2^i)')\}_{i\in[m]}})) \neq \emptyset$$

**Edges Between Two Connected Components.** In our graph where basis schema are nodes, we now have two connected components:

- Schemas where the first monotone curve is a monotone (up) curve from lines $k$ to $1$

- Schemas where the first monotone curve is a monotone (down) curve from lines $1$ to $k$.

Within each of these connected components, we have proved connectedness with the previous two cases, by connecting schemas of the same prefix (of monotone curves) and the same level, and schemas of the same prefix and different levels.

The nodes in the first connected component will all connect in a tree to the schema consisting of a single monotone curve from lines $k$ to $1$. The nodes in the second connected component will all connect in a tree to the schema consisting of a single monotone curve from lines $1$ to $k$.

Let $Y$ be the schema for $m=2$ with a single monotone (up) curve from line $k$ to $1$, and $Z$ be the schema for $m=3$ with two monotone curves: one monotone (down) curve from line $1$ to $k$ and then a monotone (up) curve from $k$ to $2$. These are two representatives from the two aforementioned connected components. We claim that:

$$\text{int}(A(Y)) \cap \text{int}(A(Z)) \neq \emptyset$$

We will start with an interior point for $Z$. Suppose it has segment lengths $z_1, \ldots, z_{k+1}, \ldots, z_{2k}$, where $z_1$ is above line $1$, $z_{k+1}$ is below line $k$, and segment $z_{2k}$ lies above line $2$ and below line $1$, with end-point on line $2$.

Again, ignore the normalization condition that $\sum_{i\in[2k]} z_i = 1$. For $\epsilon > 0$, we assign values to the segments as follows.

$$z_1 = 1$$
$$\forall 2 \leq i \leq 2k-1, z_i = \frac{p_i}{q_i}\sum_{j=1}^{i-1} z_j + \epsilon$$

Note that since $z_{2k}$ is the last segment of the schema, it need not cross a sector and need only cross and re-cross line 2. Therefore, it has no constraints under schema $Z$ except that $z_{2k} \geq 0$. However, the key idea is that we can choose for $z_{2k}$ to be sufficiently large so as to satisfy the constraint needed to cross to line 1 from line 2.

$$z_{2k} = \frac{p_{2k}}{q_{2k}} \sum_{j=1}^{2k-1} z_j + \epsilon$$

Such a $\{z_i\}_{i \in [2k]}$, after normalization, exists in the (relative) interior of $A^{(M)}(Z)$.

We claim that rearranging the segments $\{z_i\}_{i \in [2k]}$ into a monotone (up) curve $(y_1, \ldots, y_{k+1})$ with slack $\epsilon$ will always be possible for any $\epsilon > 0$. This essentially follows from Lemma E.13 as we have intentionally set $z_{2k} = \frac{p_{2k}}{q_{2k}} \sum_{j=1}^{2k-1} z_j + \epsilon$ large enough to cross from line 2 to line 1 (and have its end-point on line 1). More precisely, the following constraints are met by $\{z_i\}_{i \in [2k]}$.

$$\forall k+2 \leq i \leq 2k, z_i = \frac{p_i}{q_i} \sum_{j=1}^{i-1} z_j + \epsilon$$
$$\geq \frac{p_i}{q_i} \sum_{j \leq i-1 : z_j \text{ in sector} \geq (2k+3-i)} z_j + \epsilon$$

Then, we obtain a valid monotone curve with segment lengths $(y_1, \ldots, y_{k+1})$ by setting:

$$y_1 = z_{k+1}$$
$$y_{k+1} = z_1$$
$$y_i = z_{i+k} + z_{k+2-i}, 2 \leq i \leq k$$

Thus, the large value of $z_{2k}$ enables the re-arranged test-function of monotone schema to cross from line 2 to line 1. Finally, there is no constraint on $y_{k+1}$, so all constraints of the schema $Y$ are satisfied. This re-arrangement will have slack $\min(\epsilon, y_1, y_{k+1}) \geq \epsilon$ because each of the original constraints in $Z$ on the segments $z_1, \ldots, z_{2k-1}$ as well as the "pseudo-constraint" we imposed on $z_{2k}$ only got looser in this rearrangement, and there is no constraint on $y_{k+1}$. Thus, the rearranged monotone curve, after normalization of the segment lengths, is in the interior of $A^{(M)}(Y)$.

$\{y_i\}$ and $\{z_i\}$ are both interior in their respective polytopes and induce the same activations $(B_1, \ldots, B_k)$ since the segments of $\{y_i\}$ are just a re-arrangement of the segments of $\{z_i\}$, and $\sum_{i \in [2k]} z_i = \sum_{i \in [k+1]} y_i$. Thus, they give rise to the same interior point $(B_1, \ldots, B_k) \in \text{int}(A(Y)), (B_1, \ldots, B_k) \in \text{int}(A(Z))$, so $\text{int}(A(Y)) \cap \text{int}(A(Z)) \neq \emptyset$.

These edges demonstrate that the graph is connected, proving the Lemma. ∎

**Proof of Lemma E.30.**

**Lemma E.35.** *(Lemma E.30, restated) If for every $f \in$ C-RASP$^2$ with $K(f)$ heads, we have that $z > 0$ and $\sum_{i \in [K(f)]} \lambda_i > z$, then for any $f, f' \in$ C-RASP$^2$ that are not equal, with $K$ and $K'$ heads respectively, then $f$ and $f'$ will differ on some continuous test-function, $\mathcal{Y}$, which strictly satisfies the second layer inequalities. That is, with $\max(K, K') \leq k \leq K + K'$ distinct heads between $f$ and $f'$, let $(B_1(\mathcal{Y}), \ldots, B_k(\mathcal{Y}))$ be the activations induced by $\mathcal{Y}$. Then, either:*

$$\sum_{i=1}^{K} \lambda_i B_{\text{ord}(1,i)}(\mathcal{Y}) > z$$
$$\sum_{i=1}^{K'} \lambda_i' B_{\text{ord}(2,i)}(\mathcal{Y}) < z'$$

*Or,*

$$\sum_{i=1}^{K} \lambda_i B_{\text{ord}(1,i)}(\mathcal{Y}) < z$$

$$\sum_{i=1}^{K'} \lambda_i' B_{\text{ord}(2,i)}(\mathcal{Y}) > z'$$

*Proof of Lemma E.30.* Fix $f, f'$ and their $k$ distinct slopes $\{s_i\}_{i \in [k]}$. Suppose $f, f'$ differ on some discrete test-function $\mathcal{X}$. We want to show that there must be a *continuous* test-function that strictly satisfy the two inequalities strictly and distinguishes $f$ and $f'$.

Let's consider the $k$-dimensional set $\mathbb{A}(\{s_i\}_i) \subset [0, 1]^k$. $\mathbb{A}(\{s_i\}_i)$ is equal to the union of a finite number of convex sets by Corollary E.14 and Lemma E.16.

$$\mathbb{A}(\{s_i\}_{i \in [k]}) = \bigcup_{m \in [k], \text{valid } \{(y_1^i, y_2^i)\}_{i \in [m]} \subset \{1, \dots, k\}^m} A(Y_{\{(y_1^i, y_2^i)\}_{i \in [m]}})$$

Now consider two halfspaces $H_1, H_2$, induced by the second layers of $f$ and $f'$.

$$H_1 := \{B \in \mathbb{R}^k : \sum_{i=1}^{K} \lambda_i B_{\text{ord}(1,i)} > z\}$$

$$H_2 := \{B \in \mathbb{R}^k : \sum_{i=1}^{K'} \lambda_i' B_{\text{ord}(2,i)} > z'\}$$

First, we cannot have that $H_1 = H_2$, as then $f$ and $f'$ must share the same heads in their first layer and also have the same coefficients in the second layer. Thus, $f = f'$, contradicting the original assumption that $f$ and $f'$ are not equal (as witnessed by $\mathcal{X}$).

Thus, $H_1 \neq H_2$. Denote

$$L_1 := \{B \in \mathbb{R}^k : \sum_{i=1}^{K} \lambda_i B_{\text{ord}(1,i)} = z\}$$

$$L_2 := \{B \in \mathbb{R}^k : \sum_{i=1}^{K'} \lambda_i' B_{\text{ord}(2,i)} = z'\}$$

$H_1 \neq H_2$ implies $L_1 \neq L_2$. Now, there are two cases, given by whether $L_1$ and $L_2$ have a nonzero intersection or not.

**Case 1:** $L_1 \cap L_2 \neq \emptyset$

$H_1, H_2$ divide $\mathbb{R}^k$ into 4 quadrants. Denote the $(+, +)$ quadrant as $H_1 \cap H_2$, $(+, -)$ quadrant as $H_1 \cap H_2^c$, $(-, +)$ quadrant as $H_1^c \cap H_2$ and $(-, -)$ quadrant as $H_1^c \cap H_2^c$.

Towards contradiction, let us consider all possible $\mathbb{A}(\{s_i\}_i)$ such that the following Condition I holds.

- (Condition I) there is no $B \in \mathbb{A}(\{s_i\}_i)$ in the $(-, +)$ and $(+, -)$ quadrants such that:

$$\sum_{i=1}^{K} \lambda_i B_{\text{ord}(1,i)} > z$$

$$\sum_{i=1}^{K'} \lambda_i' B_{\text{ord}(2,i)} < z'$$

or

$$\sum_{i=1}^{K} \lambda_i B_{\text{ord}(1,i)} < z$$

$$\sum_{i=1}^{K'} \lambda_i' B_{\text{ord}(2,i)} > z'$$

For ease of notation, index each basis schema of Corollary E.14 with $j \in [N_k]$, where $N_k \ (= 2^{k-1})$ is the total number of basis schema, so that $\mathbb{A}(\{s_i\}_i) = \bigcup_{j=1}^{N_k} A_j$. Thus, Condition I must also apply to each $A_j, \forall j \in [N_k]$.

In order that $A_j$ satisfies Condition I, there is only three convex possibilities:

1. (Type I) $A_j$ is a convex subset of a $(k-1)$-dimensional plane that has nonzero intersection with the $(k-2)$-dimensional intersection of $O = L_1 \cap L_2$ and that is a subset of

$$\big[\text{cl}(H_1) \cap \text{cl}(H_2)\big] \cup \big[H_1^c \cap H_2^c\big]$$

2. (Type II) $A_j$ is a convex subset of the quadrant $\text{cl}(H_1) \cap \text{cl}(H_2)$, where:

$$\sum_{i=1}^{K} \lambda_i B_{\text{ord}(1,i)} \geq z$$

$$\sum_{i=1}^{K'} \lambda_i' B_{\text{ord}(2,i)} \geq z'$$

3. (Type III) $A_j$ is a convex subset of the quadrant, $H_1^c \cap H_2^c$, where:

$$\sum_{i=1}^{K} \lambda_i B_{\text{ord}(1,i)} \leq z$$

$$\sum_{i=1}^{K'} \lambda_i' B_{\text{ord}(2,i)} \leq z'$$

Lemma E.33 implies that it is impossible for any $A_j$ to be Type I since the dimension of Type I sets is $k-1$ but $A_j$ is dimension $k$.

Lemma E.34 implies that either all $A_j$ are type II or all $A_j$ are type III, which can be seen as follows. Suppose towards contradiction that there was an $A_j$ of type II and a separate $A_{j'}$ of type III. First, no point $O = L_1 \cap L_2$ can be an interior point of any $A_j$, because $O$ is a $k-2$ dimensional subspace, and it is impossible for an $L_2$-ball centered at any point in $O$ of any nonzero radius and dimension $k$ to be completely contained in $[\text{cl}(H_1) \cap \text{cl}(H_2)] \cup [H_1^c \cap H_2^c] \supset \bigcup_{j=1}^{N_k} A_j$, as such a ball would always have a non-empty intersection with $H_1 \cap \text{int}(H_2^c)$ and $H_2 \cap \text{int}(H_1^c)$. Thus, $O$ is disjoint from $\bigcup_{j \in [N_k]} \text{int}(A_j)$. Because there is an $A_j$ of type II and a separate $A_{j'}$ of type III, part of $\bigcup_{j=1}^{N_k} \text{int}(A_j)$ is contained in $\text{cl}(H_1) \cap \text{cl}(H_2)$ and part of it is contained in $H_1^c \cap H_2^c$. However, since no point in $O$ can be in $\bigcup_{j=1}^{N_k} \text{int}(A_j)$, then $\bigcup_{j=1}^{N_k} \text{int}(A_j)$ is not connected (i.e. it is the union of two non-empty, separated sets). This contradicts Lemma E.34. Thus, all $A_j$ must be type II or they must all be type III.

Finally, observe that $(0, \ldots, 0) \in \mathbb{A}(\{s_i\}_i) = \bigcup_{j=1}^{N_k} A_j$, realized by a test-function which has slope 0 everywhere. Also, $(1, \ldots, 1) \in \mathbb{A}(\{s_i\}_i) = \bigcup_{j=1}^{N_k} A_j$, realized by a test-function which has slope 1 everywhere.

Suppose all $A_j$ were type II. At least one $A_j$ must contain $(0, \ldots, 0)$, but if $(0, \ldots, 0)$ is in the quadrant, $\mathrm{cl}(H_1) \cap \mathrm{cl}(H_2)$, this implies:

$$0 = \sum_{i=1}^{K'} (\lambda_i' \cdot 0) \geq z'$$

$$0 = \sum_{i=1}^{K} (\lambda_i \cdot 0) \geq z$$

This contradicts the assumption that $z, z' > 0$.

Now, suppose all $A_j$ were type III. At least one $A_j$ must contain $(1, \ldots, 1)$, but if $(1, \ldots, 1)$ is in the quadrant, $H_1^c \cap H_2^c$, this implies:

$$\sum_{i=1}^{K'} (\lambda_i' \cdot 1) \leq z'$$

$$\sum_{i=1}^{K} (\lambda_i \cdot 1) \leq z$$

This contradicts the assumption that $\sum_{i=1}^{K} \lambda_i > z$ and $\sum_{i=1}^{K'} \lambda_i' > z'$.

In short, $z > 0$ and $\sum_{i \in [K(f)]} \lambda_i > z$ for all functions in $f \in \text{C-RASP}^2 \implies \mathbb{A}(\{s_i\}_i)$ does not satisfy Condition I.

Thus, there exists $B \in \mathbb{A}(\{s_i\}_i)$, realized by a continuous test-function, in the $(-, +)$ and $(+, -)$ quadrants such that:

$$\sum_{i=1}^{K} \lambda_i B_{\mathrm{ord}(1,i)} > z$$

$$\sum_{i=1}^{K'} \lambda_i' B_{\mathrm{ord}(2,i)} < z'$$

or

$$\sum_{i=1}^{K} \lambda_i B_{\mathrm{ord}(1,i)} < z$$

$$\sum_{i=1}^{K'} \lambda_i' B_{\mathrm{ord}(2,i)} > z'$$

**Case 2:** $L_1 \cap L_2 = \emptyset$.

The argument here is similar. We know $L_1, L_2$ are parallel hyperplanes which never intersect. We will argue that it is still true that $\{A_j\}_{j \in [N_k]}$ are either all type II or all type III, and then reach a contradiction with the assumption that $z > 0$ and $\sum_{i \in [K(f)]} \lambda_i > z$ for all functions in C-RASP$^2$. Given that $L_1, L_2$ are parallel, consider two subcases.

Let $(+, -), (+, -)$ denote the subcase where halfspace $H_1, H_2$ have the same parity, but their intercept term is different. An example of this are two halfspaces: "$x + y > 2, x + y > 1$".

Let $(-,+),(+,-)$ be the subcase where $H_1, H_2$ have opposite parities. An example of this are the two halfspaces:"$x+y < 2, x+y > 1$".

**Subcase 1:** $(+,-),(+,-)$ In this case, in order that the planes are not identical, there is a nonzero difference in the intercept term between $L_1, L_2$. By Lemma E.34, since $\bigcup_{j\in[N_k]} \text{int}(A_j)$ is connected, then $\{A_j\}_{j\in[N_k]}$ are either all type II or all type III, via a similar argument as Case 1, which contradicts that $z, z' > 0$, and $\sum_{i=1}^{K} \lambda_i > z$ and $\sum_{i=1}^{K'} \lambda_i' > z'$ respectively. Thus, Condition I cannot hold.

**Subcase 2:** $(+,-),(-,+)$ In this case, either $\text{cl}(H_1) \cap \text{cl}(H_2) \neq \emptyset$ or $\text{int}(H_1^c) \cap \text{int}(H_2^c) \neq \emptyset$. Once again, $\{A_j\}_{j\in[N_k]}$ are either all type II or all type III, contradicting that $z, z' > 0$, and $\sum_{i=1}^{K} \lambda_i > z$ and $\sum_{i=1}^{K'} \lambda_i' > z'$ respectively. Thus, Condition I cannot hold. ∎

Here is a Corollary of Lemma E.34 which will be useful in strengthening Lemma E.30, which will let us improve the final bound from $O(T^{O(K^2)})(\alpha^{O(\log \alpha)})$ to $O(T^{O(K)})(\alpha^{O(1)})$.

**Corollary E.36.** *(Connectedness of $\text{int}(\mathbb{A}_{2,3}(\{s_i\}_{i\in[k]})))$ Define $\mathbb{A}_{2,3}(\{s_i\}_{i\in[k]})$ as the set of activations by basis schema where $m \in \{2,3\}$. That is, there is only either one or two monotone curves in the schema.*

$$\mathbb{A}_{2,3}(\{s_i\}_{i\in[k]}) := \bigcup_{m\in\{2,3\},valid \ \{(y_1^i,y_2^i)\}_{i\in[m]} \subset \{1,\ldots,k\}^m} A(Y_{\{(y_1^i,y_2^i)\}_{i\in[m]}})$$

*Then, we claim that $\bigcup_{m\in\{2,3\},valid \ \{(y_1^i,y_2^i)\}_{i\in[m]} \subset \{1,\ldots,k\}^m} \text{int}(A(Y_{\{(y_1^i,y_2^i)\}_{i\in[m]}}))$ is connected.*

*Proof.* We have that $\mathbb{A}_{2,3}(\{s_i\}_{i\in[k]}) := A(Y_{\{(1,k),(k,k)\}}) \cup \bigcup_{j\in\{2,\ldots,k-1\}} A(Y_{\{(1,k),(k,j),(j,j)\}}) \cup A(Y_{\{(k,1),(1,1)\}}) \cup \bigcup_{j\in\{2,\ldots,k-1\}} A(Y_{\{(k,1),(1,j),(j,j)\}})$. The connectedness of $\text{int}(A(Y_{\{(1,k),(k,k)\}})) \cup \bigcup_{j\in\{2,\ldots,k-1\}} \text{int}(A(Y_{\{(1,k),(k,j),(j,j)\}})) \cup \text{int}(A(Y_{\{(k,1),(1,1)\}})) \cup \bigcup_{j\in\{2,\ldots,k-1\}} \text{int}(A(Y_{\{(k,1),(1,j),(j,j)\}}))$ follows from inspecting the edges in the graph construction in Lemma E.34.

In the sections where we analyzed edges between nodes in the same level and between consecutive levels, it follows that:

$$\forall j \in \{2,\ldots,k-1\}, \text{ node } Y_{\{(1,k),(k,k)\}} \text{ is connected to node } Y_{\{(1,k),(k,j),(j,j)\}}$$

And

$$\forall j \in \{2,\ldots,k-1\}, \text{ node } Y_{\{(k,1),(1,1)\}} \text{ is connected to node } Y_{\{(k,1),(1,j),(j,j)\}}$$

In the final section of the proof of Lemma E.34, we showed that

$$\text{node } Y_{\{(k,1),(1,1)\}} \text{ is connected to node } Y_{\{(1,k),(k,2),(2,2)\}}$$

∎

As a corollary, we can strengthen Lemma E.30 to the following.

**Lemma E.37.** *(Stronger version of Lemma E.30) If for every $f \in$ C-RASP$^2$ with $K(f)$ heads, we have that $z > 0$ and $\sum_{i\in[K(f)]} \lambda_i > z$, then for any $f, f' \in$ C-RASP$^2$ that are not equal, with $K$ and $K'$ heads respectively, then $f$ and $f'$ will differ on some continuous test-function, $\mathcal{Y}$, which strictly satisfies the second layer inequalities. That is, with $\max(K, K') \leq k \leq K + K'$ distinct heads between $f$ and $f'$, let $(B_1(\mathcal{Y}),\ldots,B_k(\mathcal{Y}))$ be the activations induced by $\mathcal{Y}$. Then, either:*

$$\sum_{i=1}^{K} \lambda_i B_{\text{ord}(1,i)}(\mathcal{Y}) > z$$

$$\sum_{i=1}^{K'} \lambda_i' B_{\text{ord}(2,i)}(\mathcal{Y}) < z'$$

*Or,*

$$\sum_{i=1}^{K} \lambda_i B_{\mathrm{ord}(1,i)}(\mathcal{Y}) < z$$

$$\sum_{i=1}^{K'} \lambda_i' B_{\mathrm{ord}(2,i)}(\mathcal{Y}) > z'$$

*In addition, $\mathcal{Y}$ will be of a basis schema $Y$ with either one or two monotone curves: $Y \in \{Y_{\{(y_1^i, y_2^i)\}_{i \in [m]}} : m \in \{2, 3\}, valid \{(y_1^i, y_2^i)\}_{i \in [m]} \subset \{1, \ldots, k\}^m\}$. In particular, the number of segments $M$ in the schema of $\mathcal{Y}$ will be at most $2 \cdot k$ instead of $k^2$ for general basis schema.*

*Proof.* We repeat the proof of Lemma E.30, but with $\mathrm{int}(\mathbb{A}_{2,3}(\{s_i\}_{i \in [k]}))$ substituted for $\mathbb{A}(\{s_i\}_{i \in [k]})$ and Corollary E.36 substituted for Lemma E.34. Note that $\mathrm{int}(\mathbb{A}_{2,3}(\{s_i\}_{i \in [k]}))$ still contains the points $(0, \ldots, 0) \in [0, 1]^k$ and $(1, \ldots, 1) \in [0, 1]^k$ as well, as both of these are realized by a test-function of where the slope everywhere is equal to $0$ everywhere and $1$ everywhere, respectively, which are both of schemas with just one monotone curve. ∎

