# OpenReview forum: "Non-Asymptotic Length Generalization"
_ICML.cc/2025/Conference — ICML 2025 poster_

### Official Review · Reviewer_YXKG · 2025-03-11

**Overall Recommendation:** 3

**Summary:**

Summary:
The paper analyzes the minimum input-length required for learning length generalizable DFA, CFG and C-RASP programs and provides non-asymptotic bounds for the minimal length inputs to differentiate different DFA and C-RASP programs.

Strengths:
The proof idea is quite interesting: the authors design a finite set of schemas that corresponds to the prefix behaviors of all possible inputs and maps them to convex polytopes. The analysis on binary sequences are then converted into geometric analysis in high-dimensional space, which provides an interesting perspective on how C-RASP programs process sequential inputs.
The proven C-RASP length generalization theorem and its proof idea can potentially serve as a critical step in showing how Transformers achieve length-generalization on certain classes of computational problems. This further strengthens the importance of the RASP-L conjecture.

Weaknesses:

The proof sketch in the main content may be quite confusing without reading the supplementary materials. In particular, the author can provide more details on the “basis schemas” in the main content, which is a core contribution but quite confusing in the current way it’s written on page 7. In particular, it’s unclear what the $$Y_{\{(y_1^i,y_2^i)\}}$$ or $$m$$ refer to on page 7 since they are not properly explained. Figure 2 is also currently quite confusing since 1) the average prefix sums should be discrete instead of continuous for finite input lengths and 2) the “monotone” parts of the function are apparently not monotone. It is also not immediately clear why the number of basis schemas is finite. Although the exact meaning of the authors can be clarified by looking at the details of page 19 and 20, it would be better for readers if these confusions can be resolved by looking at the main paper and Figure 2 alone since these are critical parts of the paper’s contribution and should be made as explicit as possible. As such, I would recommend the authors spend more space explaining this portion of the paper.

The following concerns are regarding scope limitations rather than technical weaknesses:
Although the authors showed an interesting perspective on length-generalization and C-RASP and provide a novel proof for a formal language theory result, there’s still a huge gap when viewed from the perspective of length-generalization of Transformers, which C-RASP is designed to model. Therefore, it is unclear whether the author’s main contributions can indeed lead to fundamental progress regarding Transformer models. In particular, according to my current understanding:
It is yet unclear whether there’s a training setting (i.e. loss function) for Transformers such that the optima correspond to the minimum C-RASP program interpolator on the training dataset.
For any given number of layers and heads, the C-RASP program have very restricted expressive capability compared to RASP or the full Transformer architecture
The proven results implies that, if ALL inputs of length at most O((KT)^{(O(K^2)}) are given to a minimum length C-RASP interpolator, then the resulting C-RASP program must be length-generalizable. However, since this require the dataset size to be exponential, this is highly unrealistic, so a more interesting result for the ML community might be probabilistic (i.e., the probability of a dataset of size D drawn according to some distribution over {0,1}^* to result in a length-generalizable C-RASP interpolator).
The exponential dependence on O(K^2) is also quite unrealistic for Transformer models, which would cause the length to explode even for K=5, for example.
As such, it is unclear whether the proven theorem can indeed be part of an a larger theorem regarding Transformer length generalization, or whether it would be easier to pursue a path without C-RASP as the intermediate language.

Despite these limitations, the perspective introduced by the theorem and proof is quite interesting and may lead to further advancement in the general field of formal language theory and Transformers, and thus I recommend accepting the paper, especially after the proof clarification suggested earlier.

**Claims And Evidence:**

There are 3 main theoretical claims of the paper, each of which is supported by corresponding theorems and proofs:
non-asymptomatic upper bound for length generalization for DFAs
Impossibility result for non-asymptomatic upper bound for length generalization for CFGs
non-asymptomatic upper bound for length generalization for C-RASP

**Essential References Not Discussed:**

I’m not aware of closely related works not mentioned in the paper

**Experimental Designs Or Analyses:**

N/A

**Methods And Evaluation Criteria:**

The paper uses the minimum length differentiating 2 C-RASP programs to study length-generalization of Transformer-based LLMs. There are apparent limitations in this approach, since it requires all inputs up to a given length to be given as training data for length generalization to occur. As a result, the training data size is exponential w.r.t. the length bound, which is already exponential w.r.t. the number of heads in the C-RASP program $K$. There paper also considers C-RASP programs of depth 2, which is a very limited setting. The paper also assumes that the learning process is a minimum complexity interpolator, but whether this indeed represents Transformer learning is unknown both theoretically or empirically. As such, there’s still a long way from this work towards solving the overarching problem.
Having the above said, since it’s the first work in this direction AFAIK, I believe the paper still provides decent contribution for the field.
For the proof, the authors design a finite set of schemas that corresponds to the prefix behaviors of all possible inputs and maps them to convex polytopes. The analysis on binary sequences are then converted into geometric analysis in high-dimensional space, which provides an interesting perspective on how C-RASP programs process sequential inputs.

**Other Comments Or Suggestions:**

I would suggest to drastically shorten the abstract to at most 1/2 the size. It currently sounds like an intro rather than a concise summary of results. I recommend only one sentence for motivation, once sentence for DFA+CFG results (since they are not the main contribution), 2 sentences for the C-RASP result, and once sentences on the proof method.
Similarly, the introduction section also contains too much motivation that are very far from the actual results shown in the paper.
The saved spaces can be use to provide a better description of the proof sketch.
I believe that the paper would be good for acceptance with better organization and proof sketch

**Other Strengths And Weaknesses:**

The proof sketch in the main content may be quite confusing without reading the supplementary materials. In particular, the author can provide more details on the “basis schemas” in the main content, which is a core contribution but quite confusing in the current way it’s written on page 7. In particular, it’s unclear what the $$Y_{\{(y_1^i,y_2^i)\}}$$ or $$m$$ refer to on page 7 since they are not properly explained. Figure 2 is also currently quite confusing since 1) the average prefix sums should be discrete instead of continuous for finite input lengths and 2) the “monotone” parts of the function are apparently not monotone. It is also not immediately clear why the number of basis schemas is finite. Although the exact meaning of the authors can be clarified by looking at the details of page 19 and 20, it would be better for readers if these confusions can be resolved by looking at the main paper and Figure 2 alone since these are critical parts of the paper’s contribution and should be made as explicit as possible. As such, I would recommend the authors spend more space explaining this portion of the paper.

**Questions For Authors:**

Are my following understanding of the current state of the problem and the practicality of the paper true?
It is yet unclear whether there’s a training setting (i.e. loss function) for Transformers such that the optima correspond to the minimum C-RASP program interpolator on the training dataset.
For any given number of layers and heads, the C-RASP program have very restricted expressive capability compared to RASP or the full Transformer architecture. Even though C-RASP is an upper bound on the expressiveness, it results in drastically larger number of heads and layers
The proven results implies that, if ALL inputs of length at most O((KT)^{(O(K^2)}) are given to a minimum length C-RASP interpolator, then the resulting C-RASP program must be length-generalizable. However, since this require the dataset size to be exponential, i.e. 2^O((KT)^{(O(K^2)})
The exponential dependence on O(K^2) is would cause the length to explode even for a Transformers with 5 heads, for example.

**Relation To Broader Scientific Literature:**

The paper shows a condition for length-generalization of Transformer LLMs on simple computational problems (expressible by 2-layer C-RASP) assuming that the RASP-L conjecture is also true for C-RASP.
Previous works have mostly focused on expressiveness upper bounds and lower bounds, while this paper leverages the ideas and results (e.g., C-RASP) from these previous works work to approach the problem of length generalization.
Although there any many limitations in the current work as discussed in “Methods”, the paper’s proof idea and perspective can be contributive for future work in the direction.

**Theoretical Claims:**

The full proof (40 pages long) is apparently to long to be fully checked. I checked the main novel idea from page 19 to page 21, which makes sense assuming the correctness of lemmas in the first 8 pages. However, I would greatly appreciate if the authors can have a better proof sketch regarding the definition of test-functions and the proof of convexity in the main content.

---

> ### Author Rebuttal · Authors · 2025-03-29
>
> We sincerely thank reviewer YXKG for their time, detailed comments, and constructive criticism.
>
> **Author statement:** The submitted paper contained an error in the proofs, which we discovered only after submission. We’ve revised the proofs and sent the revised draft to the AC. Here are how the revised draft’s results differ from the submitted draft’s.
> - The main result (Theorem 5.1) is unchanged except for the addition of a weak assumption to the function class $C-RASP^{2,K,T}$, that $\sum_{i \in [K]} \lambda_i > z$ for each function in $C-RASP^{2,K,T}$ (analogous to the existing assumption $z > 0$; see line 220, left col). Theorem 5.1 claims the same upper bound of $N_{A_{si}}(K,T) \leq O((KT)^{O(K^2)})$ for $C-RASP^{2,K,T}$. The proof is almost exactly the same. The significance of the result is unchanged.
> - The revised draft no longer contains Theorems 5.4 and 5.6, which together were intended to extend Theorem 5.1 to prove length generalization guarantees for a class of functions which contains Dyck1 and its higher-precision variants. The absence of these side-results doesn’t affect the significance of the paper.
> - Results for DFAs and CFGs unchanged.
>
> Source of error:
> - Lemma A.24 in line 1557
>
> We apologize for the error and hope that either the reviewer can view the revised draft; or ignore Theorems 5.4 and 5.6 in their review. Thank you.
>
> **Addressing Reviewer Concerns:**
>
> > The proof sketch in the main content may be quite confusing
>
> We apologize for the lack of clarity of the proof sketch and thank the reviewer for their patience in reading through it. We will re-write it and include more details of the basis schemas in the main content.
>
> > It is yet unclear whether there’s a training setting for Transformers such that the optima correspond to the minimum C-RASP program interpolator on the training dataset.
>
> This is true. However, there is circumstantial evidence which lends some credence to the idea of studying transformer training with the min-complexity interpolator, for some complexity measure.
>
> - Abbe et. al. find empirically that transformers are biased towards learning functions of low degree-profile [1].
>
> - Bhattamishra et. al. find empirically that transformers are biased towards learning functions of low sensitivity [2].
>
> If one assumes that there is some complexity measure $C$ such that training transformers with SGD roughly corresponds to the min-complexity interpolator with $C$ (which is strong, granted), then the RASP-L paper [3] suggests that $C$ should be RASP-L program length, as they show empirically that when the ground-truth is also a short-RASP program, then the trained model achieves far better length generalization.
>
> [1] Abbe et. al. arxiv.org/pdf/2301.13105
>
> [2] Bhattamishra et. al. arxiv.org/pdf/2211.12316
>
> [3] Zhou et. al. arxiv.org/abs/2310.16028
>
> > The C-RASP program have very restricted expressive capability
>
> We agree.
>
> > The proven results imply that … since this require the dataset size to be exponential, this is highly unrealistic
>
> Having all inputs of length at most O((KT)^{(O(K^2)}) is a sufficient condition for length generalization, and indeed unrealistic. A weaker sufficient condition is that for every pair of unequal functions in the hypothesis class, there is at least one (short) input which distinguishes them. The cardinality of the training set need only be at most the square of the number of hypotheses (which is $O(T^{O(K)})$ instead of $2^{ O(T^{O(K^2)})}$).
>
> > A more interesting result might be probabilistic
>
> This is an important question. If one introduces a test or train distribution (or both) and relax the definition of length generalization, one can perhaps get more palatable sample complexity.
>
> If one keeps the current notion of length generalization (i.e. identification) but require that the training set be drawn i.i.d. from some training distribution, we suspect that the sample complexity will be no better than our doubly-exponential sample complexity, because perfect length generalization requires one to distinguish every pair of functions in the hypothesis class. Given two very similar functions $f, f’$, we suspect the strings of length $n$ which distinguish $f$ and $f’$ may have density which is order $1/\exp(n)$. Random sampling is no better than including all strings of length n.
>
> One could also relax the definition of length generalization from identification. One could say that length generalization is achieved if the hypothesis achieves 99% accuracy on a particular test-distribution. However, it is unclear what is a realistic test distribution is, so that the theoretical setup models practical scenarios. Also, the 99%-accuracy length generalization notion may be too weak. The 1% of examples which a learned hypothesis gets wrong could be exclusively on the long/interesting examples that we care about. If the test-distribution is supported on {0,1}$^*$, the weight of all strings length $N$ or larger will be less than 1% for N suff. large.

---

### Official Review · Reviewer_CEK9 · 2025-03-13

**Overall Recommendation:** 4

**Summary:**

The paper provides new theoretical results related to criteria on a training set such that an idealized learner can learn a function that exhibits length generalization.

The paper considers an idealized learner, termed the Minimum-Complexity Interpolator. The learner is defined with respect to a hypothesis class (i.e. set of functions) and a complexity function defined over this set. The learner learns the lowest complexity function in the set that perfectly fits the training data.

The paper then studies, for several different tasks, what is the maximum input length required in the training set such that for any data generating function in the hypothesis class, the learner is guaranteed to learn a function that exhibits length generalization.

The paper derives results related to the upper bound on this maximum input length for three settings:

* n-state Deterministic Finite Automata (DFA) – deriving a bound of 2n-2
* Context-Free Grammars (CFGs) – deriving a negative result
* C-RASP: a class of functions mapping binary strings to binary labels, using the RASP derivative defined by Yang & Chiang, 2024. Prior work has shown that this class of functions is related to those expressible by Transformers (with some caveats related to whether the Transformer is assumed to be finite vs. infinite precision). These results apply to the Dyck-1 language studied by prior work.

## Update after rebuttal

Thank you for your response and for being transparent about the issue identified with theorems 5.4 and 5.6. As the main result of 5.1 still holds, this does not significantly affect my judgement.

Despite the limitations with respect to the potential gap between the theory and empirically observed behavior with real models and optimizers (which are clearly acknowledged by the authors and noted by the other reviewers) I still recommend acceptance for this paper. I think it brings not only a novel perspective, but also non-trivial theoretical results. I can see this work having implications for not only future theory work in this area, but also influencing how we design "fair" (i.e. that, minimally, there exists a plausible minimum complexity interpolator that exhibits the desired generalization behavior) splits to assess length generalization and other OOD generalizations, as well as potentially inspiring new methods that seek to add inductive biases to bridge the gap between the minimum complexity interpolator and empirically observed behavior.

**Claims And Evidence:**

Yes, the limitations of the key claims are discussed, i.e. that they do not consider expressibility of real architectures (e.g. finite-precision Transformers) and the idealized learner may not closely relate to the functions learned in practice (e.g. by SGD).

**Essential References Not Discussed:**

As far as I'm aware, there are no essential references missing that directly relate to the theoretical results.

However, I do think that the paper could be improved and made relevant to a wider audience by discussing a broader range of prior work for the camera ready, e.g.:

* The connection with the RASP-L conjecture was good to see discussed. (Concurrent work) Shaw et al. 2024 (https://arxiv.org/abs/2410.18077) also discuss the RASP-L conjecture and define the notion of program minimality with respect to a training set, which might be relevant in this context. They connect this notion to the existence of a local optimum for a Universal Transformer that exhibits length generalization. They then similarly compute the minimum input length required for such an optimum to exist, e.g. for addition. While clearly not the same as the current work, there are similarities in the ambition to characterize necessary conditions on a training set that enable length generalization.
* Empirical work that studies length generalization in Transformers related to Chomsky hierarchy complexity might be good to mention to provide context for the theoretical results in this paper: https://arxiv.org/abs/2207.02098, https://arxiv.org/abs/2305.16843, https://arxiv.org/abs/2009.11264
* Other theoretical work specific to Transformers: Hahn et al. 2020 (https://aclanthology.org/2020.tacl-1.11/) has negative theoretical findings for Transformers ability to express length invariant algorithms for parity and Dyck languages. Some of these results were subsequently discussed and extended in Chiang & Cholak, 2022 (https://arxiv.org/abs/2202.12172), providing a more positive result.
* To provide more context for the negative results for CFG induction, it would be useful to understand how this relates to positive empirical results for inducing grammars that exhibit length generalization (using description length priors). For example, https://arxiv.org/abs/2010.12725 induced synchronous CFGs that exhibit length generalization on the SCAN benchmark using a description length prior over the grammars that appear similar to the complexity measure of CFGs studied in this paper. I assume the apparent contradiction is resolved because while the paper result holds for CFGs in general, many specific instances of CFGs may have a finite upper bound for identifying the underlying grammar? Or perhaps the difference between inducing CFGs and synchronous CFGs for transduction is critical here. Perhaps some discussion could help connect the results to the literature on linguistically-motivated models for length generalization.

**Experimental Designs Or Analyses:**

N/A - Purely theoretical results.

**Methods And Evaluation Criteria:**

N/A - Purely theoretical results.

**Other Comments Or Suggestions:**

Nits:

* Section 1 - “(Zhou et al., 2023) make” > Use \citet? This nit also applies to many other citations in the paper.
* Section 1 - “A theoretical guarantee is particularly crucial here, as empirically verifying a model’s ability to generalize to arbitrarily long sequences is impractical, if not impossible.” > Nit: This claim seemed a bit strong. If such guarantees are “crucial”, then it seems problematic that the theoretical results do not hold for models trained in practice, e.g. with SGD. In practice, testing models on inputs that are arbitrarily longer than those seen during training seems sufficient for most applications.
* It might be useful to clearly define what a “non-asymptotic” upper bound vs. “asymptomatic” upper bound in this context, given that it is such a key claim. Some readers might be familiar, but good to make explicit. Paragraph 2 of section 2 discusses this, but could maybe clarify this more directly and earlier.

**Other Strengths And Weaknesses:**

(Note that my overall impression and score assume the correctness of the proofs, which I did not carefully verify)

Strengths:

* The paper offers a new perspective on the feasibility of inducing functions that exhibit length generalization from finite length training sets. Length-based splits have been popular to empirically probe the generalization capabilities of various models and learning algorithms, so it is useful to build a stronger theoretical foundation of when such generalization is even theoretically feasible (for an idealized learner).
* The results on C-RASP are quite general, i.e. it is a very broad class of functions. It is also nice because it directly relates to the computational model of Transformers.
* The negative CFG results could be of interest to the community working on linguistically-motived methods for compositional generalization.

Weaknesses:

* Although clearly acknowledged, the results hold for a “minimum complexity interpolator”, and there is no theoretical or empirical evidence that this learner behaves similarly to, e.g. Transformers trained with SGD. However, while some empirical results could relate various complexity measures to the actual inductive biases of Transformers, I don't think this is necessary for this paper, which already makes theoretical contributions.

**Questions For Authors:**

None (see recommendations above)

**Relation To Broader Scientific Literature:**

As far as I am aware, the new results are novel.

The paper not only offers new theoretical results, but also offers a new perspective on assessing the feasibility (albeit under idealized assumptions) of length generalization from finite length training sets.

**Theoretical Claims:**

I did not check the correctness of the proofs in detail.

---

> ### Author Rebuttal · Authors · 2025-03-29
>
> We sincerely thank reviewer CEK9 for their time, detailed comments, and constructive criticism.
>
> **Author statement:** The submitted paper contained an error in the proofs, which we discovered only after submission. We’ve revised the proofs and sent the revised draft to the AC. Here are how the revised draft’s results differ from the submitted draft’s.
> - The main result (Theorem 5.1) is unchanged except for the addition of a weak assumption to the function class $C-RASP^{2,K,T}$, that $\sum_{i \in [K]} \lambda_i > z$ for each function in $C-RASP^{2,K,T}$ (analogous to the existing assumption $z > 0$; see line 220, left col). Theorem 5.1 claims the same upper bound of $N_{A_{si}}(K,T) \leq O((KT)^{O(K^2)})$ for $C-RASP^{2,K,T}$. The proof is almost exactly the same. The significance of the result is unchanged.
> - The revised draft no longer contains Theorems 5.4 and 5.6, which together were intended to extend Theorem 5.1 to prove length generalization guarantees for a class of functions which contains Dyck1 and its higher-precision variants. The absence of these side-results doesn’t affect the significance of the paper.
> - Results for DFAs and CFGs unchanged.
>
> Source of error:
> - Lemma A.24 in line 1557
>
> We apologize for the error and hope that either the reviewer can view the revised draft; or ignore Theorems 5.4 and 5.6 in their review. Thank you.
>
> **Addressing Reviewer Concerns:**
>
> > the paper could be made relevant to a wider audience by discussing a broader range of prior work
>
> We thank the reviewer for the very detailed list of references. We will add them in.
>
> > Shaw et. al. induced synchronous CFGs... I assume the apparent contradiction is resolved because ...
>
> Yes, specific instances of CFGs, $f \in \mathcal{F}$, have different ``identification times", $N_{A_{si}}(f)$ (line 126, left column). As such, some CFGs require much longer-length data than others in order to be identified by $A_{si}$. But this doesn’t contradict that there is a particular sequence of CFGs where the identification times as a function of the complexity of the CFGs (i.e. their description lengths) can grow faster than any computable function. All CFGs have finite $N_{A_{si}}(f)$, but this doesn’t preclude some CFGs from having very large $N_{A_{si}}(f)$ which is not bounded by any reasonable function of $C(f)$.
>
> In addition, the empirical studies of CFG induction evaluate the learned CFG on a test-set, with finite support. Achieving good accuracy on the test-set is an easier task than identification.
>
> > The results hold for a minimum complexity interpolator…
>
> We acknowledge that it is largely unclear whether the min-complexity interpolator behaves similarly to the realistic transformer training setting. However, there is circumstantial evidence which lends some credence to the idea of studying transformer training with the min-complexity interpolator.
>
> - Abbe et. al. find empirically that transformers are biased towards learning functions of low degree-profile [1].
>
> - Bhattamishra et. al. find empirically that transformers are biased towards learning functions of low sensitivity [2].
>
> - The RASP-L conjecture [3] suggests that transformers are biased towards learning functions of short RASP-L program length.
>
> [1] Abbe et. al. arxiv.org/pdf/2301.13105
>
> [2] Bhattamishra et. al. arxiv.org/pdf/2211.12316
>
> [3] Zhou et. al. arxiv.org/abs/2310.16028
>
> > (1) If such guarantees are crucial, then it is problematic that the theoretical results do not hold for models trained in practice. (2) In practice, testing models on inputs that are arbitrarily longer than those seen during training seems sufficient for most applications.
>
> (1) We agree
>
> (2) (i) It isn’t clear whether for most applications, one would have access to arbitrarily long data to test whether one’s model length generalized. Long proofs of difficult theorems, for instance, are very scarce.
>
> (ii) Even if one could generate arbitrarily long inputs to test one’s model for length generalization, for large integer $n > 0$, it would be infeasible to test one’s model on all $\exp(n)$ inputs of length $n$. One would have to choose a relatively-small representative set of examples of length $n$ which could certify that one's model has length generalized, but it isn’t clear how to do this systematically. I.i.d. samples may be too easy for the model to cheat/hack (see Fig. 6; page 9 of [4]).
>
> Due to (i) and (ii), we hope that theoretical guarantees could increase confidence that a model has length generalized, when checking length generalization empirically is hard.
>
> [4] Liu et. al. arxiv.org/abs/2210.10749
>
> > define “non-asymptotic” upper bound vs. “asymptotic” upper bound
>
> Will do. A non-asymptotic upper bound of $N_{A_{si}}(c)$ is a computable upper bound of $N_{A_{si}}(c)$ in $c$, the complexity of the ground truth, only omitting universal constants with Big-O notation. Asymptotic upper bounds omit some dependencies on the complexity or they only guarantee that $N_{A_{si}}(c) < \infty$.

---

### Official Review · Reviewer_5Dit · 2025-03-14

**Overall Recommendation:** 3

**Summary:**

The paper considers the question of the theory of length generalization for abstract classes of models, with in mind applications to extending the theory underlying important questions about prevalent models such as transformers.

Here length generalization denotes the property that if the model is perfectly accurate on length \le L inputs, then it will be perfectly accurate on inputs of any length. The questions addressed are
1) Does my model/hypothesis class allow for length-generalization?
2) If the answer to (1) is affirmative, then what is the value of L that guarantees this length-generalization?

For (1) the paper gives some negative results in line with work by Angluin on context-free grammars.
Importantly, the paper shows that (1) has positive answer for finite automata and for 2-layer C-RASP programs (modelling 2-layer transformers). The case of deeper (3 or more layer) C-RASP is left for future work.

When (1) has a positive answer, then the answer to (2) is formulated in terms of the complexity of models within the given class.

The guarantees are done in an abstract setting, using "minimum-complexity interpolator" algorithms, for which existence results are known, but which are in practice hard to find. Nevertheless, this is the first such result, and includes positive results for C-RASP, making this a possibly useful read for anyone interested in theoretical machine learning.

**Claims And Evidence:**

I think that the claims and limitations are well balanced.

It is sometimes hard to parse the actual presentation of the claims, and one has to read the paper at least twice in order to understand the claims.

**Essential References Not Discussed:**

I am not aware of any.

**Experimental Designs Or Analyses:**

There are no experiments in the paper.

**Methods And Evaluation Criteria:**

The paper does not have experiments.

**Other Comments Or Suggestions:**

Line 109 second column: "a complexity" is supposed to have what properties? can you give the reader an idea about what you have in mind?

Line 113 first column: "takes in a training set" would have to be "takes as an input a training set"

Line 119 and rest of the paper: two functions are equal if they take the same value and have the same domain, so why don't you use that notion and instead rename it by saying "they are equivalent"? For functions this is just the notion of equality. For algorithms, two algorithms are equivalent if they induce equal functions. But here you introduce a new name for the usual equality of functions, and this is very confusing. Please be precise about this wording?

Line 130-131 "Identification is synonymous with exactly learning or recovering the ground truth" -- it is not, the concept is more complicated than this summary.. perhaps try being precise with the descriptions, so that the reader doesn't have to guess what you were thinking there.

Line 139 "will allow you to correctly predict" -- I guess it's not "you" or "me" or a specific person, so I'd remove the word "you" from this sentence

Line 155, add "where c is a natural number"

About algorithm 1 and in general: can you maybe spend a few words regarding what the term "interpolator" refers to? what is it interpolating?

Line 119 second column "are very nontrivial" -- what does that mean exactly? I think you can be more specific/precise rather than handwaving it like that.

Line 124 second column "the learner" -- what is the difference between that and the algorithm itself? Or how is \mathcal A_{si} defined? In the paper this notion of "learner" is never defined, and to me it seems like there is no learning involved in your paper so I've been confused every time it talks about "learner" or "learning". Is it really necessary to use that kind of terminology?

Line 127 second column I'd add "in the following sense", since Lemma 3.1 is making precise the idea (the adjective "best") hinted at in the preceding paragraph.

 Line 189 - 190 first column: Is A_{si} the same as before? and what does it mean for it to "see inputs of length at least" some number?

About the definition of context-free grammars: Of course you know that this is not the correct definition. This is a definition of any grammar. For being context-free you have to put some restriction on the production rules, similarly to the case of linear CFG's that you wrote correctly.

Line 171-172 second column: "clean" means what? what detergent was it cleaned with?

In definition 3.6, it would be better to introduce the parameters before writing the formulas, not at the end of the statement.. So the reader doesn't have to read twice in order to parse the statements.

Line 268-269 first column "it undecidable" -- add the verb "is". Besides this typo, you should also be more precise with this statement.. do you mean that
a) there exist two CFGs for which it is undecidable whether they are equivalent, or
b) for any CFG there exists another one for which it is undecidable whether the two are equivalent or
c) there exists a pair of CFGs for which it is undecidable whether they are equivalent ?

Line 232-233 second column: the sentence about "Unfortunately [..] rather sudden" seems useless, consider removing it.

Line 311-316 second column: "This inability of the transformer architecture to even express the ground truth should be part of the reason why empirical length generalization results only show limited length generalization" -- what do you mean by "should be part of the reason why"? and why should it be part of that? this statement would be interesting if it were justified. But here it's stated a bit out of the blue, and without an actual argument it is empty. Please write the argument in some detail? (Also if you expand this a bit, it makes you careful about not mixing up cause with effect, a possible fallacy which is impossible to check with the current wording)

Definition 6.2 seems to be almost contradicting itself and a bit weird: the part about a tuple of rational numbers, when the tuple is a singleton, is a distinct definition than the case about single numbers. For example, 37/5 has 37-precision (as the larger between 37 and,5 is 37), whereas { 37/5 } has 5-precision (as the common denominator is 5)? And the fraction 370 / 50 has 370 precision or does it have 37 precision like 37/5 has?

Maybe check that this definition being weird does not have consequences in the places in which it is used?

**Other Strengths And Weaknesses:**

I think the main weakness is the areas with lack of clarity in the explanation in the paper.

Somehow the results seem presented without much care, and this is surprising as it should be simpler to do than to produce proofs for the results.

I will detail the above in the "other comments" section.

**Questions For Authors:**

A question I have is if the authors can discuss a bit what are extensions of the notion of length generalization to real-life cases in which we don't check for "perfect generalization" but rather for "generalization to accuracy 99%", i.e. how would the methods extend to probabilistical guarantees as in PAC learning? This is important to know for practical reasons.

**Relation To Broader Scientific Literature:**

I think that this is an important step forward in the theory of length-generalization.

**Theoretical Claims:**

I did not have a 100% read of the 50 pages of the paper, but I follow the proof sketches and they seem intuitively correct.

I have read at 100% only the proof of theorem 5.1 which seems well argued, it is actually much better written than the first 5-6 pages of the paper.
This gave me confidence that the remainder of the proofs should also be well written.

---

> ### Author Rebuttal · Authors · 2025-03-29
>
> We sincerely thank reviewer 5Dit for their time, detailed comments, and constructive criticism.
>
> **Author statement:** The submitted paper contained an error in the proofs, which we discovered only after submission. We’ve revised the proofs and sent the revised draft to the AC. Here are how the revised draft’s results differ from the submitted draft’s.
> - The main result (Theorem 5.1) is unchanged except for the addition of a weak assumption to the function class $C-RASP^{2,K,T}$, that $\sum_{i \in [K]} \lambda_i > z$ for each function in $C-RASP^{2,K,T}$ (analogous to the existing assumption $z > 0$; see line 220, left col). Theorem 5.1 claims the same upper bound of $N_{A_{si}}(K,T) \leq O((KT)^{O(K^2)})$ for $C-RASP^{2,K,T}$. The proof is almost exactly the same. The significance of the result is unchanged.
> - The revised draft no longer contains Theorems 5.4 and 5.6, which together were intended to extend Theorem 5.1 to prove length generalization guarantees for a class of functions which contains Dyck1 and its higher-precision variants. The absence of these side-results doesn’t affect the significance of the paper.
> - Results for DFAs and CFGs unchanged.
>
> Source of error:
>
> - Lemma A.24 in line 1557
>
> We apologize for the error and hope that either the reviewer can view the revised draft; or ignore Theorems 5.4 and 5.6 in their review. Thank you.
>
> **Addressing Reviewer Concerns:**
>
> > the main weakness is the areas with lack of clarity in the explanation in the paper.
>
> We apologize for the lack of clarity of the explanations and thank the reviewer for their patience in reading through the paper and for very detailed edits. We will improve it.
>
> > Line 109 second col: "a complexity" is supposed to have what properties?
>
> The only property we require of a complexity measure is that it is a mapping from function class to natural numbers. We pick complexity measures which are ``natural", such as the number of states in a DFA, or the description size of a CFG. For C-RASP, we chose $T$ and $K$ as two natural parameters.
>
> > Algorithm 1: what does "interpolator" refer to
>
> We meant that the learning algorithm picks a hypothesis $h \in \mathcal{F}$ such that each $(x,y)$ pair in the training set, $h(x) = y$.
>
> > Line 119 second col; Explain "are very nontrivial"
>
> We mean that it difficult to characterize the learned solution by SGD in non convex landscape. The learned solution is not necessarily local minima; even if it is a local minima, we don’t know which one it is due to non convexity.
>
> > Line 124 second col "the learner" - what is the difference
>
> “Learner” and Learning algorithm are synonymous. Learning is involved in the paper since the learning algorithm takes as input a training set and outputs a hypothesis.
>
> > Line 189 - 190 first col: (1) Is A_{si} the same as before? (2) what does it mean for it to "see inputs of length at least" some number?
>
> (1) Yes; $A_{si}$ always refers to Algorithm 1.
>
> (2) The sentence containing the phrase was meant as a prose introduction to definition 3.2 on line 192 (left column). The phrase “the learner sees inputs of length at most N” means that it takes as input dataset $D_N$ of (string, label) pairs for all strings of length at most $N$.
>
> > Line 268-269 first col: be more precise with this (undecidability) statement
>
> The computational problem which we’re saying is undecidable is: Given two CFGs $G_1$ and $G_2$, output whether or not $L(G_1) = L(G_2)$.
>
> > Line 311-316 second col: "This inability of the transformer … "
>
> If no setting of the transformers weights will result in the function expressed by the overall transformer being equal to the ground truth, then the transformer cannot learn a function which is correct on all inputs. What’s more, for many ground truths investigated empirically like PARITY, we believe that any function expressible by transformers will differ from the ground truth on infinitely many strings in {0,1}$^*$. Thus, transformer cannot length generalize perfectly to arbitrary lengths.
>
> > Definition 6.2 seems contradictory
>
> The definition of precision of rational numbers only applies in the context of rational numbers in [0,1], so the magnitude of the denominator is all that matters. Further, the precision of a rational number is calculated from the simplest form of the rational number, where its numerator and denominator are relatively prime.
>
> > Length generalization to accuracy 99%
>
> This is an important question. We can define train and test distributions, $D_{train}$ and $D_{test}$ over {0,1}$^*$ to make the setup more like PAC. E.g. for each natural number $N_{train}$ which is the maximum length of strings in the train distribution, let the test distribution be the uniform distribution over strings of length $N_{train} + 1$. However, it was unclear to us what choice of test and train distribution would make for a generally-realistic model for situations practitioners care about.
> More discussion in rebuttal to Reviewer YXKG at the bottom.

---

> > ### Comment · Reviewer_5Dit · 2025-04-02
> >
> > Thanks. I hope that if it gets accepted, you'll polish the presentation considerably. If you promise this, I'll raise the score to 4 (I don't think that the amendment on the theorems is ruining the paper, it's fine).
> >
> > For the question replies, they are along the lines that I was expecting, thanks for taking the time to reply, and i hope the best for this paper.

---

> > > ### Author Response · Authors · 2025-04-03
> > >
> > > We promise that we will polish the presentation considerably, especially the proof sketch. Thank you so much for your time and feedback.

---

### Official Review · Reviewer_qTut · 2025-03-14

**Overall Recommendation:** 2

**Summary:**

Taking up recent interest in length generalization of transformers and similar models, this paper studies length generalization for three formal models: DFAs, CFGs, and two-layer C-RASP, with the latter being the main technical contribution. C-RASP has recently been proposed as a formal model of some computations performed by transformers. The present paper shows that programs in two-layer sub-fragments of C-RASP can be identified from inputs up to an explicit computable bound (polynomial in precision, super-exponential in the number of heads). This is shown using ingenious arguments based on convex geometry. As such, it provides a potential path towards proving non-asymptotic quantitative length generalization guarantees.

## update after rebuttal

I maintain the stated concerns after the rebuttal. I believe that the authors and I are on the same page about the presence of these weaknesses.

**Claims And Evidence:**

The claims as described in Abstract, Introduction, and Conclusion are supported by convincing evidence.

**Essential References Not Discussed:**

I believe all essential references are discussed.

**Experimental Designs Or Analyses:**

N/A (no experiments).

**Methods And Evaluation Criteria:**

N/A. No empirical evaluation is conducted (nor is necessarily needed for this kind of work).

**Other Comments Or Suggestions:**

- around line 308 (right column): it is argued that PARITY is not expressible by transformers due to the AC0 bound. However, many C-RASP functions, such as Dyck-1, are also not in AC0. Relevant references re the difficulty of PARITY include [1,2]. I think state tracking problems believed to be outside of TC0 [3] might be a more apt example here.

[1] Hahn and Rofin, Why are Sensitive Functions Hard for Transformers?, ACL 2024
[2] Chiang and Cholak, Overcoming a theoretical limitation of self-attention, ACL 2022
[3] Merrill and Sabharwal, The Parallelism Tradeoff: Limitations of Log-Precision Transformers, TACL 2023

- what does the "si" in A_{si} in Lemma 3.1 stand for?

**Other Strengths And Weaknesses:**

Strengths

- The paper tackles a very important theoretical problem of current interest: providing non-asymptotic length generalization guarantees, and links to a recent line of work on RASP variants.

- The paper is upfront about some key limitations of the proposed setup.

Weaknesses

- I believe the biggest weakness of the paper is that the guarantees apply not to machine learning models themselves, but to the C-RASP formalism. As such, the paper proves an (interesting) result about that logical formalism, not about machine learning methods. It remains open (or at least isn't made explicit) how minimum-complexity inference based on C-RASP complexity can be linked at all to training transformers or other machine learning models. The authors are correct in stating that understanding SGD training is extremely intractable in this context, but at least linking to minimum-complexity inference based on transformers' complexities (e.g., parameter weight norms) would help make the relevance to machine learning more convincing. Besides proving theoretical links, experiments could also be helpful.

- Section 6 is very hard to read (some concrete questions below under "Questions for Authors"). I'd suggest a major rewrite of this section to reduce the technical burden on the reader, and minimally avoid any undefined formal objects.

**Questions For Authors:**

line 333, left column: why "s_k" not "s_N"?

line 341, left column: "they're in [0,1]": does this mean "both are subsets of [0,1]"?

line 356, right column: what does "B" range over?

line 361, right column: the completeness equation is hard to grasp at this point, e.g. "valid" has not been introduced. A prose statement could be more comprehensible.

line 413, left column: "the linear constraint for the i-th face" -- a "constraint" would be an equation, whereas here a number is referenced -- does "constraint" refer to the linear coefficient of the face?

line 427, right column: "and its higher-precision variants" -- what does this refer to?

**Relation To Broader Scientific Literature:**

The paper follows up on recent work arguing that length generalization in transformers can be predicted based on RASP fragments (RASP-L in Zhou et al 2024, C-RASP in Huang et al 2024, both cited in the paper). The key contribution is to show that, for some small RASP fragment (two-layer C-RASP), there are computable upper bounds on the lengths needed to allow full identification (hence, generalization) within the given hypothesis class.

I think the paper could acknowledge the link to Huang et al 2024 [4] a bit more. That work is mentioned once in passing in Section 2, but it is actually quite related in that (1) it also studies length-generalization for C-RASP, (2) also aims to formalize the RASP-L conjecture, (3) also uses a minimum-complexity interpolator, albeit with a different kind of complexity measure that applies to transformers instead of automata or C-RASP programs. There is of course a clear distinguishing property of the current paper, namely that it provides a non-asymptotic guarantee, and mentioning the relation to that prior work where relevant could help contextualize the paper better.

[4] Huang et al, A Formal Framework for Understanding Length Generalization in Transformers, ICLR 2025

**Theoretical Claims:**

I checked the proofs about C-RASP at a high level and they appear plausible to me. Within the reviewing timeframe, I could not check all details. While I did not check that the claimed scaling of input length is correct, I believe that a statement of this type can be obtained with the techniques used in this paper.

---

> ### Author Rebuttal · Authors · 2025-03-29
>
> We sincerely thank reviewer qTut for their time, detailed comments, and constructive criticism.
>
> **Author statement:**
> The submitted paper contained an error in the proofs, which we discovered only after submission. We’ve revised the proofs and sent the revised draft to the AC. Here are how the revised draft’s results differ from the submitted draft’s.
> - The main result (Theorem 5.1) is unchanged except for the addition of a weak assumption to the function class $C-RASP^{2,K,T}$, that $\sum_{i \in [K]} \lambda_i > z$ for each function in $C-RASP^{2,K,T}$ (analogous to the existing assumption $z > 0$; see line 220, left col). Theorem 5.1 claims the same upper bound of $N_{A_{si}}(K,T) \leq O((KT)^{O(K^2)})$ for $C-RASP^{2,K,T}$. The proof is almost exactly the same. The significance of the result is unchanged.
> - The revised draft no longer contains Theorems 5.4 and 5.6, which together were intended to extend Theorem 5.1 to prove length generalization guarantees for a class of functions which contains Dyck1 and its higher-precision variants. The absence of these side-results doesn’t affect the significance of the paper.
> - Results for DFAs and CFGs unchanged.
>
> Source of error:
> - Lemma A.24 in line 1557
>
> We apologize for the error and hope that either the reviewer can view the revised draft; or ignore Theorems 5.4 and 5.6 in their review. Thank you.
>
> **Addressing Reviewer Concerns:**
> > it remains open how minimum-complexity inference based on C-RASP complexity is linked to training transformers
>
> This is a valid point. However, there is some circumstantial evidence which lends some credence to the idea of studying transformer training with the min-complexity interpolator.
> - Abbe et. al. find empirically that transformers are biased towards learning functions of low degree-profile [1].
> - Bhattamishra et. al. find empirically that transformers are biased towards learning functions of low sensitivity [2].
> - The RASP-L conjecture [3] suggests that transformers are biased towards learning functions of short RASP-L program length.
>
> [1] Abbe et. al. arxiv.org/pdf/2301.13105
>
> [2] Bhattamishra et. al. arxiv.org/pdf/2211.12316
>
> [3] Zhou et. al. arxiv.org/abs/2310.16028
>
> > Section 6 is very hard to read.
>
> We apologize for the lack of clarity in Section 6 and thank the reviewer for their patience in reading through it. We will rewrite it.
>
> > line 333, left col: why "s_k" not "s_N"?
>
> $k$ is the number of 2D lines of the form $Y = s_i \cdot X$, with slope $s_i \in (0,1)$. The slope of each 2D line correspond to the parameters of a head in the first layer of either $f$ or $f’$.
>
> $N$ is the length of the discrete test-function (equivalently, of the string which corresponds to the discrete test-function).
> The interplay of $k$ and $N$ is in the definition of the activations induced by the discrete test-function, where for $i \in [k]$, the $i$th activation is the proportion of "time-steps" from 1 to N in which the test-function lies above line $i$.
>
> $\forall i \in[k], B_i := \frac{1}{N} \sum_{j = 1}^N 1[ps(x)_j > s_i \cdot j]$
>
> > line 341, left col: "they're in [0,1]": does this mean "both are subsets of [0,1]"?
>
> Yes
>
> > line 356, right col: what does "B" range over?
>
> $(B_1(x), \ldots, B_k(x))$ ranges over all possible binary strings $x \in$ { 0,1}$^*$. Each binary string corresponds to a discrete test-function, which induces a tuple of activations over the fixed $k$ lines of the form $Y = s_i \cdot X$.
>
> > line 413, left col: "the linear constraint for the i-th face" -- a "constraint" would be an equation, whereas here a number is referenced
>
> A constraint here is a linear inequality, e.g. $x_1 + ... + x_{M} \geq 1$. In line 413, we mean that the $i$th face of the polytope is parameterized by the boundary of the half-space given by some linear inequality $L_i$. Evaluating a point $x \in R^M$ on the constraint $L_i$ is denoted as $L_i(x)$ and returns a scalar, equal to the margin that the point $x$ has on the constraint. The variable $C$ in the Lemma 6.4 is the centroid defined in the preceding paragraph (line 405, left column), which is a point. $L_i(C)$ is the margin of the centroid on the $i$th face.
>
> > line 427, right col: what does "higher-precision variants" refer to
>
> This referred to Definition A.43 on line 2255 on page 42. However, the Dyck1 results are no longer valid.
>
> > what does the "si" in A_{si} stand for
>
> simplest interpolator
>
> > line 308 (right column): it is argued that PARITY is not expressible … I think state tracking problems believed to be outside of TC0 are a more apt example.
>
> We understand the reviewer’s point as that in order to argue that there are ground-truth functions which are experimentally tested for length generalization but that are theoretically intractable (due to not having a short C-RASP program), we should pick a function outside TC0 since C-RASP is upper bounded by TC0. We thank the reviewer for the insightful comment.
>
> > could acknowledge link to Huang et al more
>
> Will do.

---

> > ### Comment · Reviewer_qTut · 2025-04-02
> >
> > Thanks for the rebuttal. It appears the authors and I are generally on the same page about these questions (which is great), and I believe that the original assessment from my review continues to be valid. As my concern is largely about (subjective) significance, not rigor, I do not want to stand in the way of the paper being accepted.

---

> > > ### Author Response · Authors · 2025-04-03
> > >
> > > Thank you very much for your time and feedback.

---

### Decision · Program_Chairs · 2025-05-01

**Decision:**

Accept (poster)

**Comment:**

This paper develops an interesting theoretical perspective on length generalization in transformers using C-RASP models (a class of functions that is a superset of functions expressible by finite-precision transformers). All reviewers have commented on the technical novelty and rigour of the paper. There were some comments on improving the writing, which the authors are advised to incorporate in the camera-ready manuscript.